# FEDERATED Q-LEARNING: LINEAR REGRET SPEEDUP WITH LOW COMMUNICATION COST

**Zhong Zheng, Fengyu Gao, Lingzhou Xue & Jing Yang**[*]
The Pennsylvania State University
`{zvz5337,fzg5170,lzxue,yangjing}@psu.edu`

## ABSTRACT

In this paper, we consider federated reinforcement learning for tabular episodic Markov Decision Processes (MDP) where, under the coordination of a central server, multiple agents collaboratively explore the environment and learn an optimal policy without sharing their raw data. While linear speedup in the number of agents has been achieved for some metrics, such as convergence rate and sample complexity, in similar settings, it is unclear whether it is possible to design a *model-free* algorithm to achieve linear *regret* speedup with low communication cost. We propose two federated Q-Learning algorithms termed as FedQ-Hoeffding and FedQ-Bernstein, respectively, and show that the corresponding total regrets achieve a linear speedup compared with their single-agent counterparts when the time horizon is sufficiently large, while the communication cost scales logarithmically in the total number of time steps $T$. Those results rely on an event-triggered synchronization mechanism between the agents and the server, a novel step size selection when the server aggregates the local estimates of the state-action values to form the global estimates, and a set of new concentration inequalities to bound the sum of non-martingale differences. This is the first work showing that linear regret speedup and logarithmic communication cost can be achieved by model-free algorithms in federated reinforcement learning.

## 1 INTRODUCTION

Federated Learning (FL) (McMahan et al., 2017) is a distributed machine learning framework, where a large number of clients collectively engage in model training and accelerate the learning process, under the coordination of a central server. Notably, this approach keeps raw data confined to local devices and only communicates model updates between the clients and the server, thereby diminishing the potential for data exposure risks and reducing communication costs. As a result of these advantages, FL is gaining traction across various domains, including healthcare, telecommunications, retail, and personalized advertising.

On a different note, Reinforcement Learning (RL) (Sutton & Barto, 2018) is a subfield of machine learning focused on the intricate domain of sequential decision-making. Often modeled as a Markov Decision Process (MDP), the primary objective of RL is to obtain an optimal policy through sequential interactions with the previously unknown environment. RL has exhibited superhuman performances in various applications, such as games (Silver et al., 2016; 2017; 2018; Vinyals et al., 2019), robotics (Kober et al., 2013; Gu et al., 2017), and autonomous driving (Yurtsever et al., 2020), and garnered increasing attentions in different domains.

However, training an RL agent often requires large amounts of data, due to the inherent high dimensional state and action spaces (Akkaya et al., 2019; Kalashnikov et al., 2018), and sequentially generating such training data is very time-consuming (Nair et al., 2015). It thus has inspired a line of research that aims to extend the FL principle to the RL setting. The FL framework allows the agents to collaboratively train their decision-making models with limited information exchange between the agents, thereby accelerating the learning process and reducing communication costs. Among them, some model-based algorithms (e.g., Chen et al. (2023)) and policy-based algorithms (e.g.,

---

[*]L. Xue and J. Yang are co-corresponding authors.

Fan et al. (2021)) have already exhibited speedup with respect to the number of agents for learning regret or convergence rate.

Also, there is a collection of research focusing on model-free federated RL algorithms, which have shown encouraging results. Such algorithms build upon the classical value-based algorithms such as $Q$-learning (Watkins, 1989), which directly learns the optimal policy without estimating the underlying model. Among them, Jin et al. (2022) considered a heterogeneous setting where the agents interact with environments with different but known transition dynamics, and the objective is to obtain a policy that maximizes the overall performance in all environments. It proposes two federated RL algorithms, including a $Q$-learning-based algorithm called QAvg, and proves their convergence. Liu & Olshevsky (2023) investigated distributed TD-learning with linear function approximation and achieves linear convergence speedup under the assumption that the samples are generated in an identical and independently distributed (i.i.d.) fashion. Khodadadian et al. (2022) proposed federated versions of TD-learning and $Q$-learning and proved a linear convergence speedup with respect to the number of agents for both algorithms under Markovian sampling. Woo et al. (2023) studied infinite-horizon tabular Markov decision processes (MDP) and proposed both the synchronous and asynchronous variants of federated Q-learning. Both algorithms exhibit a linear speedup in the sample complexity. We note that under the aforementioned algorithms, *the clients do not adaptively update their exploration policy during the learning process.* As a result, they do not have any theoretical guarantees on the total regret among the agents[1].

In this work, we aim to answer the following question:

*Is it possible to design a federated model-free RL algorithm that enjoys both linear regret speedup and low communication cost?*

We give an affirmative answer to this question under the tabular episodic MDP setting. Specifically, we assume a central server and $M$ local agents exist in the system, where each agent interacts with an episodic MDP with $S$ states, $A$ actions, and $H$ steps in each episode independently. The server coordinates the behavior of the agents by designating their exploration policies, while the clients execute the policies, collect trajectories, and form "local updates". The local updates will be sent to the server periodically to form "global updates" and refine the exploration policy. Our contributions can be summarized as follows.

- **Algorithmic Design.** We propose two federated variants of the $Q$-learning algorithm (Jin et al., 2018), termed as FedQ-Hoeffding and FedQ-Bernstein, respectively. Those two algorithms feature the following elements in their design: 1) *Adaptive exploration policy selection.* In order to achieve linear regret speedup, it becomes necessary to adaptively select the exploration policy for all clients, which is in stark contrast to the static sampling policy adopted in Khodadadian et al. (2022); Woo et al. (2023). 2) *Event-triggered policy switching and communication.* On the other hand, to reduce the communication cost, it is desirable to keep the exploration policy switching to a minimum extent. This motivates us to adopt an event-triggered policy switching and communication mechanism, where communication and subsequent policy switching only happen when a certain condition is satisfied. This naturally partitions the learning process into rounds. 3) *Equal weight assignment for global aggregation.* When the central server updates the global estimates of the $Q$-value for a given state-action pair $(x, a)$ at step $h$, we assign equal weights for all new visits to the tuple $(x, a, h)$ within the current round. As a result, local agents do not need to send the collected trajectories to the server. Instead, it only needs to send the empirical average of the estimated values of the next states after visiting $(x, a, h)$ to the server.

- **Performance Guarantees.** Thanks to the careful design of the policy switching, communication, and global aggregation mechanisms, FedQ-Hoeffding and FedQ-Bernstein provably achieve linear regret speedup in the number of agents compared with their single-agent counterparts (Jin et al., 2018; Bai et al., 2019) when the total number of steps $T$ is sufficiently large, while the communication cost scales in $O(M^2 H^4 S^2 A \log(T/M))$. To the best of our knowledge, those are the first model-free federated RL algorithms that achieve linear regret speedup with logarithmic communication cost. We compare the regret and communication costs under multi-agent tabular episodic MDPs in Table 1.

---

[1]A comprehensive literature review is provided in Appendix A.

Table 1: Comparison of Related Algorithms

| Type | Algorithm (Reference) | Regret | Communication cost |
|------|----------------------|--------|--------------------|
| Model-based | Multi-batch RL (Zhang et al., 2022) | $\tilde{O}(\sqrt{H^2SAMT})$ | - |
| | APEVE (Qiao et al., 2022) | $\tilde{O}(\sqrt{H^4S^2AMT})$ | - |
| | Byzan-UCBVI (Chen et al., 2023) | $\tilde{O}(\sqrt{H^3S^2AMT})$ | $O(M^2H^2S^2A^2\log T)$ |
| Model-free | Concurrent Q-UCB2H (Bai et al., 2019) | $\tilde{O}(\sqrt{H^4SAMT})$ | $O(MT)$ |
| | Concurrent Q-UCB2B (Bai et al., 2019) | $\tilde{O}(\sqrt{H^3SAMT})$ | $O(MT)$ |
| | Concurrent UCB-Advantage (Zhang et al., 2020) | $\tilde{O}(\sqrt{H^2SAMT})$ | $O(MT)$ |
| | FedQ-Hoeffding (**this work**) | $\tilde{O}(\sqrt{H^4SAMT})$ | $O(M^2H^4S^2A\log(T/M))$ |
| | FedQ-Bernstein (**this work**) | $\tilde{O}(\sqrt{H^3SAMT})$ | $O(M^2H^4S^2A\log(T/M))$ |

$H$: number of steps per episode; $T$: total number of steps; $S$: number of states; $A$: number of actions; $M$: number of agents. -: not discussed.

- **Technical Novelty.** While the equal weight assignment during global aggregation is critical to reducing the communication cost in our design, it also leads to a non-trivial challenge for the corresponding theoretical analysis. This is because the specific weight assigned to each new visit depends on the total number of visits between two model aggregation points, which is not causally known when $(x,a,h)$ is visited. As a result, the weights assigned to all visits of $(x,a,h)$ do not form a martingale difference sequence. Such non-martingale property makes the cumulative estimation error in the global estimates of the value functions difficult to track. In order to characterize the **concentration of the sum of non-martingale differences**, we relate the non-martingale difference sequence with another martingale difference sequence within each round. Due to the common factor between those two sequences in each round, we are then able to bound their differences roundwisely. We believe that the techniques developed for proving concentration inequalities on the sum of non-martingale differences will be useful in future analysis of other model-free federated RL algorithms.

## 2 BACKGROUND AND PROBLEM FORMULATION

**Notations.** Throughout this paper, we assume that $0/0 = 0$. For any $C \in \mathbb{N}$, we use $[C]$ to denote the set $\{1, 2, \ldots C\}$. We use $\mathbb{I}[x]$ to denote the indicator function, which equals 1 when the event $x$ is true and equals 0 otherwise.

### 2.1 PRELIMINARIES

We first introduce the mathematical model and background on Markov decision processes.

**Tabular Episodic Markov Decision Process (MDP).** A tabular episodic MDP is denoted as $\mathcal{M} := (\mathcal{S}, \mathcal{A}, H, \mathbb{P}, r)$, where $\mathcal{S}$ is the set of states with $|\mathcal{S}| = S$, $\mathcal{A}$ is the set of actions with $|\mathcal{A}| = A$, $H$ is the number of steps in each episode, $\mathbb{P} := \{\mathbb{P}_h\}_{h=1}^{H}$ is the transition kernel so that $\mathbb{P}_h(\cdot \mid x, a)$ characterizes the distribution over the next state given the state action pair $(x, a)$ at step $h$, and $r := \{r_h\}_{h=1}^{H}$ is the collection of reward functions. In this work, we assume $r_h(x, a) \in [0, 1]$ is a deterministic function of $(x, a)$, while the results can be easily extended to the case when $r_h$ is random.

In each episode of $\mathcal{M}$, an initial state $x_1$ is selected arbitrarily by an adversary. Then, at each step $h \in [H]$, an agent observes state $x_h \in \mathcal{S}$, picks an action $a_h \in \mathcal{A}$, receives reward $r_h = r_h(x_h, a_h)$ and then transits to next state $x_{h+1}$. The episode ends when an absorbing state $x_{H+1}$ is reached.

**Policy, State Value Functions and Action Value Functions.** A policy $\pi$ is a collection of $H$ functions $\left\{\pi_h : \mathcal{S} \to \Delta^{\mathcal{A}}\right\}_{h \in [H]}$, where $\Delta^{\mathcal{A}}$ is the set of probability distributions over $\mathcal{A}$. A policy is deterministic if for any $x \in \mathcal{S}$, $\pi_h(x)$ concentrates all the probability mass on an action $a \in \mathcal{A}$. In this case, we simply denote $\pi_h(x) = a$.

We use $V_h^\pi : \mathcal{S} \to \mathbb{R}$ to denote the state value function at step $h$ under policy $\pi$ so that $V_h^\pi(x)$ equals the expected return under policy $\pi$ starting from $x_h = x$. Mathematically,

$$V_h^\pi(x) := \sum_{h'=h}^{H} \mathbb{E}_{(x_{h'}, a_{h'}) \sim (\mathbb{P}, \pi)} \left[ r_{h'}(x_{h'}, a_{h'}) \mid x_h = x \right].$$

Accordingly, we also use $Q_h^\pi : \mathcal{S} \times \mathcal{A} \to \mathbb{R}$ to denote the action value function at step $h$, i.e.,

$$Q_h^\pi(x,a) := r_h(s,a) + \sum_{h'=h+1}^{H} \mathbb{E}_{(x_{h'},a_{h'}) \sim (\mathbb{P},\pi)} \left[ r_{h'}(x_{h'},a_{h'}) \mid x_h = x, a_h = a \right].$$

Since the state and action spaces and the horizon are all finite, there always exists an optimal policy $\pi^\star$ that achieves the optimal value $V_h^\star(x) = \sup_\pi V_h^\pi(x) = V_h^{\pi^\star}(x)$ for all $x \in \mathcal{S}$ and $h \in [H]$ (Azar et al., 2017). For ease of exposition, we denote $[\mathbb{P}_h V_{h+1}](x,a) := \mathbb{E}_{x' \sim \mathbb{P}_h(\cdot|x,a)} V_{h+1}(x')$. Then, the Bellman equation and the Bellman optimality equation can be expressed as:

$$\begin{cases} V_h^\pi(x) = \mathbb{E}_{a \sim \pi_h(x)}[Q_h^\pi(x,a)] \\ Q_h^\pi(x,a) := (r_h + \mathbb{P}_h V_{h+1}^\pi)(x,a) \\ V_{H+1}^\pi(x) = 0, \quad \forall x \in \mathcal{S} \end{cases} \quad \text{and} \quad \begin{cases} V_h^\star(x) = \max_{a \in \mathcal{A}} Q_h^\star(x,a) \\ Q_h^\star(x,a) := (r_h + \mathbb{P}_h V_{h+1}^\star)(x,a) \\ V_{H+1}^\star(x) = 0, \quad \forall x \in \mathcal{S}. \end{cases} \quad (1)$$

## 2.2 THE FEDERATED RL FRAMEWORK

In this work, we consider a federated RL setting with a central server and $M$ agents, each interacting with an independent copy of the MDP $\mathcal{M}$ in parallel. The agents can communicate with the server periodically. Depending on the specific algorithm design, the agents may send different information (e.g., reward $r_h$, or estimated $V$-values $V_h$) to the central server. Upon receiving the local information, the central server then aggregates and broadcasts certain information to the clients to coordinate their exploration. Note that, just as in FL, communication is one of the major bottlenecks, and the algorithm has to be conscious of its usage. In this work, we define the communication cost of an algorithm as the number of scalars (integers or real numbers) communicated between the server and clients. We also make the assumption that there is no latency during the communications, and the agents and server are fully synchronized (McMahan et al., 2017).

Let $\pi_h^{m,s}$ be the policy adopted by agent $m$ at step $h$ in the $s$-th episode, and $x_1^{m,s}$ be the corresponding initial state. Then, the overall learning regret of the $M$ clients over $T = HJ$ steps can be expressed as

$$\text{Regret}(T) = \sum_{m \in [M]} \sum_{s=1}^{J} \left( V_1^\star(x_1^{m,s}) - V_1^{\pi_h^{m,s}}(x_1^{m,s}) \right).$$

Here, $J$ is the number of episodes and stays the same across different agents due to the synchronization assumption.

## 3 ALGORITHM DESIGN

In this section, we elaborate on our model-free federated RL algorithm termed as FedQ-Hoeffding. The Bernstein-type algorithm, termed as FedQ-Bernstein, will be introduced afterward.

### 3.1 THE FEDQ-HOEFFDING ALGORITHM

The algorithm proceeds in rounds, indexed by $k \in [K]$. Round $k$ consists of $n^k$ episodes for each agent, where the specific value of $n^k$ will be determined later. Before we proceed, we first introduce the following notations. For the $j$-th ($j \in [n^k]$) episode in the $k$-th round, we use $x_1^{m,k,j}$ to denote the initial state for the $m$-th agent, and use $\{(x_h^{m,k,j}, a_h^{m,k,j}, r_h^{m,k,j})_{h=1}^{H}\}$ to denote the corresponding trajectory. Denote $n_h^{m,k}(x,a)$ as the total number of times that the state-action pair $(x,a)$ has been visited at step $h$ during round $k$ by agent $m$, i.e., $n_h^{m,k}(x,a) = \sum_{j'=1}^{n^k} \mathbb{I}\{(x_h^{m,k,j'}, a_h^{m,k,j'}) = (x,a)\}$, and let $n_h^k(x,a) = \sum_{m=1}^{M} n_h^{m,k}(x,a)$, i.e., the total number of visits for $(x,a)$ at step $h$ during round $k$ among all agents. We also denote $N_h^k(x,a)$ as the total number of visits for $(x,a,h)$ among all agents before round $k$, i.e, $N_h^k(x,a) = \sum_{m=1}^{M} \sum_{k'=1}^{k-1} \sum_{j=1}^{n^{k'}} \mathbb{I}\{(x_h^{m,k',j}, a_h^{m,k',j}) = (x,a)\}$.

We also use $\{V_h^k : \mathcal{S} \to \mathbb{R}\}_{h=1}^{H}$ and $\{Q_h^k : \mathcal{S} \times \mathcal{A} \to \mathbb{R}\}_{h=1}^{H}$ to denote the "global" estimates of the state value function and action value function before the beginning of round $k$. Meanwhile, we use

$v_{h+1}^{m,k}(x,a)$ to denote the "local" estimate of the expected return starting at step $h+1$ at agent $m$ in round $k$ given $(x_h, a_h) = (x, a)$, and use $v_{h+1}^k(x, a)$ to denote the corresponding global estimate.

We then specify each individual component of the algorithm as follows.

**Coordinated Exploration for Agents.** At the beginning of round $k$, the server decides a deterministic policy $\pi^k = \{\pi_h^k\}_{h=1}^H$, and then broadcasts it along with $\{N_h^k(x, \pi_h^k(x))\}_{x,h}$ and $\{V_h^k(x)\}_{x,h}$ to all of the agents. When $k = 1$, $N_h^1(x, a) = 0, Q_h^1(x, a) = V_h^1(x) = H, \forall(x, a, h) \in \mathcal{S} \times \mathcal{A} \times [H]$ and $\pi^1$ is an arbitrary deterministic policy.

Once receiving such information, the agents will execute policy $\pi^k$ and start collecting trajectories.

**Event-Triggered Termination of Exploration.** During exploration, every agent $m$ will monitor $n_h^{m,k}(x, a)$, i.e., the total number of visits for each $(x, a, h)$ triple within the current round. For any agent $m$, at the end of each episode, if any $(x, a, h)$ has been visited by $\max\left\{1, \lfloor \frac{1}{MH(H+1)} N_h^k(x, a) \rfloor\right\}$ times by agent $m$, the agent will send a signal to the server, which will then request all agents to abort the exploration.

The termination condition guarantees that for any $(x, a, h, k) \in \mathcal{S} \times \mathcal{A} \times [H] \times [K]$,

$$n_h^{m,k}(x, a) \leq \max\left\{1, \left\lfloor \frac{N_h^k(x, a)}{MH(H+1)} \right\rfloor\right\}, \tag{2}$$

and for each $k \in [K]$, there exists at least one agent $m$ such that equality is met for a $(x, a, h, m)$-tuple. The inequality limits the number of visits in a round and is important for introducing our server-side information aggregation design shortly. Meanwhile, the existence of equality guarantees that a sufficient number of new samples will be generated in a round, which is the key to the proof of Theorem 4.2 about the low communication cost.

**Local Updating of the Estimated Expected Return.** Each agent updates the local estimate of the expected return $v_{h+1}^{m,k}(x, a)$ at the end of round $k$ as follows:

$$v_{h+1}^{m,k}(x, a) = \frac{1}{n_h^{m,k}(x, a)} \sum_{j=1}^{n^k} V_{h+1}^k\left(x_{h+1}^{m,k,j}\right) \mathbb{I}\{(x_h^{m,k,j}, a_h^{m,k,j}) = (x, a)\}, \forall h \in [H],$$

i.e., for each $(x, a)$ visited at step $h$ during round $k$, $v_{h+1}^{m,k}$ is obtained by taking the empirical average of the global estimates of the value of the next visited state in the current round $k$.

Next, each agent $m$ sends $\{r_h(x, \pi_h^k(x))\}_{x,h}, \{n_h^{m,k}(x, \pi_h^k(x))\}_{x,h}$ and $\{v_{h+1}^{m,k}(x, \pi_h^k(x))\}_{x,h}$ to the central server for aggregation.

**Server-side Information Aggregation.** Denote $\alpha_t = \frac{H+1}{H+t}$, $\theta_0^0 = 1$, $\theta_t^0 = 0$ for $t \geq 1$, and $\theta_t^i = \alpha_i \prod_{i'=i+1}^t (1 - \alpha_{i'}), \forall 1 \leq i \leq t$. We also denote $\alpha^c(t_1, t_2) = \prod_{t=t_1}^{t_2} (1 - \alpha_t)$ for any positive integers $t_1 < t_2$.

Then, after receiving the information sent by the agents, for each $(x, a, h)$ tuple visited by the agents, the server sets $t^{k-1} = N_h^k(x, a), t^k = N_h^{k+1}(x, a), \alpha_{agg} = 1 - \alpha^c(t^{k-1} + 1, t^k)$ and $\beta^k(x, a, h) = 2\sum_{t=t^{k-1}+1}^{t^k} \theta_{t^k}^t b_t$ for some confidence bound $b_t$ to be determined later. When there is no ambiguity, we will also use $\beta^k$ to represent $\beta^k(x, a, h)$. Then the server updates the global estimate of the value functions according to one of the following two cases.

- **Case 1:** $N_h^k(x, a) < 2MH(H+1) =: i_0$. Due to Equation (2), this case implies that each client can visit each $(x, a)$ pair at step $h$ at most once. Then, we denote $1 \leq m_1 < m_2 \ldots < m_{t^k - t^{k-1}} \leq M$ as the agent indices with $n_h^{m,k}(x, a) > 0$. The server then updates the global estimate of action values as follows:

$$Q_h^{k+1}(x, a) = (1 - \alpha_{agg})Q_h^k(x, a) + \alpha_{agg}r_h(x, a) + \sum_{t=1}^{t^k - t^{k-1}} \theta_{t^k}^{t^{k-1}+t} v_{h+1}^{m_t, k}(x, a) + \beta^k/2. \tag{3}$$

- **Case 2:** $N_h^k(x, a) \geq i_0$. In this case, the central server calculates $v_{h+1}^k(x, a)$ as

$$v_{h+1}^k(x, a) = \frac{1}{n_h^k(x, a)} \sum_{m=1}^M v_{h+1}^{m,k}(x, a) n_h^{m,k}(x, a)$$

and updates the $Q$-estimate as

$$Q_h^{k+1}(x,a) = (1 - \alpha_{agg})Q_h^k(x,a) + \alpha_{agg}\left(r_h(x,a) + v_{h+1}^k(x,a)\right) + \beta^k/2. \tag{4}$$

After finishing updating the estimated $Q$ function, the central server updates the estimated value function and the policy as follows:

$$V_h^{k+1}(x) = \min\left\{H, \max_{a' \in \mathcal{A}} Q_h^{k+1}(x,a')\right\}, \quad \forall (x,h) \in \mathcal{S} \times [H], \tag{5}$$

$$\pi_h^{k+1}(x) = \arg\max_{a' \in \mathcal{A}} Q_h^{k+1}(x,a'), \forall (x,h) \in \mathcal{S} \times [H]. \tag{6}$$

The algorithm then proceeds to round $k+1$.

Algorithms 1 and 2 formally present the Hoeffding-type design. Inputs $K_0, T_0$ in Algorithms 1 are termination conditions where $K_0$ limits the total number of rounds and $T_0$ limits the total number of samples generated by all the agents before the last round.

---

**Algorithm 1** FedQ-Hoeffding (Central Server)

---

1: **Input:** $T_0, K_0 \in \mathbb{N}_+$.
2: **Initialization:** $k = 1$, $N_h^1(x,a) = 0, Q_h^1(x,a) = V_h^1(x) = H, \forall (x,a,h) \in \mathcal{S} \times \mathcal{A} \times [H]$ and $\pi^1 = \left\{\pi_h^1 : \mathcal{S} \to \mathcal{A}\right\}_{h \in [H]}$ is an arbitrary deterministic policy.
3: **while** $H \sum_{k'=1}^{k-1} Mn^{k'} < T_0 \,\&\, k \leq K_0$ **do**
4:     Broadcast $\pi^k$, $\{N_h^k(x, \pi_h^k(x))\}_{x,h}$ and $\{V_h^k(x)\}_{x,h}$ to all clients.
5:     Wait until receiving an abortion signal and send the signal to all agents.
6:     Receive $\{r_h(x, \pi_h^k(x))\}_{x,h}, \{n_h^{m,k}(x, \pi_h^k(x))\}_{x,h,m}$ and $\{v_{h+1}^{m,k}(x, \pi_h^k(x))\}_{x,h,m}$ from clients.

7:     Calculate $N_h^{k+1}(x,a), n_h^k(x,a), v_{h+1}^k(x,a), \forall (x,h) \in \mathcal{S} \times [H]$ with $a = \pi_h^k(x)$.
8:     **for** $(x,a,h) \in \mathcal{S} \times \mathcal{A} \times [H]$ **do**
9:       **if** $a \neq \pi_h^k(x)$ **or** $n_h^k(x,a) = 0$ **then**
10:         $Q_h^{k+1}(x,a) \leftarrow Q_h^k(x,a)$.
11:       **else if** $N_h^k(x,a) < i_0$ **then**
12:         Update $Q_h^{k+1}(x,a)$ according to Equation (3).
13:       **else**
14:         Update $Q_h^{k+1}(x,a)$ according to Equation (4).
15:       **end if**
16:     **end for**
17:     Update $V_h^{k+1}$ and $\pi^{k+1}$ according to Equation (5) and Equation (6).
18:     $k \leftarrow k + 1$.
19: **end while**

---

**Algorithm 2** FedQ-Hoeffding (Agent $m$ in round $k$)

---

1: $n_h^m(x,a) = v_{h+1}^m(x,a) = r_h(x,a) = 0, \forall (x,a,h) \in \mathcal{S} \times \mathcal{A} \times [H]$.
2: Receive $\pi^k$, $\{N_h^k(x, \pi_h^k(x))\}_{x,h}$ and $\{V_h^k(x)\}_{x,h}$ from the central server.
3: **while** no abortion signal from the central server **do**
4:     **while** $n_h^m(x_h, a_h) < \max\left\{1, \lfloor \frac{1}{MH(H+1)} N_h^k(x_h, a_h)\rfloor\right\}, \forall (x,a,h) \in \mathcal{S} \times \mathcal{A} \times [H]$ **do**
5:       Collect a new trajectory $\{(x_h, a_h, r_h)\}_{h=1}^H$ with $a_h = \pi_h^k(x_h)$.
6:       $n_h^m(x_h, a_h) \leftarrow n_h^m(x_h, a_h) + 1$, $v_{h+1}^m(x_h, a_h) \leftarrow v_{h+1}^m(x_h, a_h) + V_{h+1}^k(x_{h+1})$, and $r_h(x_h, a_h) \leftarrow r_h, \forall h \in [H]$.
7:     **end while**
8:     Send an abortion signal to the central server.
9: **end while**
10: $n_h^{m,k}(x,a) \leftarrow n_h^m(x,a), v_{h+1}^{m,k}(x,a) \leftarrow \frac{v_{h+1}^m(x,a)}{n_h^m(x,a)}, \forall (x,h) \in \mathcal{S} \times [H]$ with $a = \pi_h^k(x)$.
11: Send $\{r_h(x, \pi_h^k(x))\}_{x,h}, \{n_h^{m,k}(x, \pi_h^k(x))\}_{x,h}$ and $\{v_{h+1}^{m,k}(x, \pi_h^k(x))\}_{x,h}$ to the central server.

---

## 3.2 Intuition behind the Algorithm Design

$Q$**-estimate Update in Single-agent Setting.** Before we elaborate the intuition behind our algorithm design, we first provide a brief review of the Q-value estimate updating step under the Q-UCB2H algorithm (Bai et al., 2019) in the single-agent setting. Similar to FedQ-Hoeffding, Q-UCB2H also has a round-based design, where the agent updates the value function estimates at the end of each round. With a slight abuse of the notation, we use the same symbols as in Section 3.1 to denote the quantities in the single-agent setting.

In round $k$, for a given triple $(x, a, h)$ such that $n^k(x, a) > 0$, we denote the next states for all of the visits within the round as $\{x_{h+1,t}\}_{t=t^{k-1}+1}^{t^k}$. Then, the $Q$-estimate is updated sequentially and recursively for each visit as

$$Q_h(x, a) \leftarrow (1 - \alpha_t) Q_h(x, a) + \alpha_t (r_h(x, a) + V_{h+1}^k(x_{h+1,t}) + b_t), t = t^{k-1} + 1, \ldots t^k. \quad (7)$$

As a result, at the end of round $k$, we have

$$Q_h^{k+1}(x, a) = \alpha^c(t^{k-1} + 1, t^k) Q_h^k(x, a) + \sum_{t=t^{k-1}+1}^{t^k} \theta_{t^k}^t \left( r_h(x, a) + V_{h+1}^k(x_{h+1,t}) \right) + \beta^k/2. \quad (8)$$

If we treat $r_h(x, a) + V_{h+1}^k(x_{h+1,t})$ as a new estimate of the $Q_h(x, a)$ induced by one visit within round $k$, then, *all new samples are assigned with different weights* $\theta_{t^k}^t$. Together with the weight assigned for the old estimate $Q_h^k(x, a)$, it satisfies that $\alpha^c(t^{k-1} + 1, t^k) + \sum_{t=t^{k-1}+1}^{t^k} \theta_{t^k}^t = 1$.

**Major Challenge in Federated Setting.** The sequential updating rule in Equation (7) relies on full accessibility of the trajectories to the agent, which is infeasible for the central server in the federated setting due to the high communication cost. Instead of sharing the raw data, in Algorithm 2, the local agents only send $\{v_{h+1}^{m,k}(x, a)\}_{m=1}^M$ to the server. Since this is the sample average of the estimated values over all states visited after $(x, a, h)$, it does not preserve the temporal structure of the trajectories. It thus becomes impossible for the server to infer the next state for each visit and sequentially update the global estimate as in Equation (7) in general.

**Equal Weight Assignment for Q-estimate Aggregation.** We overcome the aforementioned challenge through a two-case design and new weight assignment for each visit.

In the first case, we have $N_h^k(x, a) < i_0$. Equation (2) indicates that each client visits $(x, a, h)$ at most once, which implies that $v_{h+1}^{m_t,k}$ in Equation (3) is exactly $V_{h+1}^k(x_{h+1,t})$. Thus, Equation (3) is a sequential update and is the same as Equation (8). We also remark that the design of the first case aims at early-stage accuracy and shares similar technical motivations as Bai et al. (2019).

The second case shows the key difference between our algorithm and non-federated algorithms. Since the temporal structure is no longer preserved, we cannot track the next state for a given visit to $(x, a, h)$, and it thus becomes impossible to assign a different weight to each new visit as in Equation (8). To resolve this issue, we choose to assign all visits with the same weight, while ensuring the total weight assigned to all visits unchanged, i.e.,

$$Q_h^{k+1}(x, a) = (1 - \alpha_{agg}) Q_h^k(x, a) + \sum_{t=t^{k-1}+1}^{t^k} \frac{\alpha_{agg}}{n_h^k(x, a)} \left( r_h(x, a) + V_{h+1}^k(x_{h+1,t}) \right) + \beta^k/2,$$

which is equivalent to the updating rule in Equation (4).

## 4 Performance Guarantees

Next, we provide regret upper bound for FedQ-Hoeffding as follows.

**Theorem 4.1** (Regret Upper Bound for FedQ-Hoeffding)**.** *Let* $\tilde{C} = 1/(H(H+1))$, $\iota = \max\{\iota_0, \iota_1\}$ *where* $\iota_0 = \log(2SA(T_0 + HM)(1 + \tilde{C})/p)$, $\iota_1 = \log \frac{2K_0 SAH(T_0/H+M)(1+\tilde{C})}{p}$, *and* $p \in (0, 1)$. *Define* $b_t = c\sqrt{H^3 \iota/t}$. *Under Algorithms 1 and 2, there exists a positive constant* $c > 0$ *such that, for any* $K \in [K_0]$ *and* $p \in (0, 1)$, *with probability at least* $1 - p$,

$$Regret(T) \leq O\left( \sqrt{H^4 \iota M T S A} + HSA(M-1)\sqrt{H^3 \iota} + MH^2 SA + H^4 SA(M-1) \right), \quad (9)$$

*where $T = H \sum_{k=1}^{K} n^k$ is the total number of steps in the first $K$ rounds.*

Theorem 4.1 indicates that the total regret scales as $O(\sqrt{H^4 \iota M T S A}) + \tilde{O}(M\text{poly}(H, S, A))^2$. The overhead term $\tilde{O}(M\text{poly}(H, S, A))$ is contributed by the $O(M)$ samples collected in the first stage, i.e., the burn-in cost. Such a burn-in cost is arguably inevitable in federated RL (e.g., Woo et al. (2023)). When $M = 1$, our result recovers those in Jin et al. (2018) and Bai et al. (2019). In the general federated setting, our algorithm enjoys a linear speedup in terms of $M$ when $T \geq \Omega(M\text{poly}(H, S, A))$ and the first term dominates the burn-in cost.

*Proof Sketch of Theorem 4.1.* First, for any given $(x, a, h) \in \mathcal{S} \times \mathcal{A} \times [H]$, we assign a global visiting index $i$ to each local visit before starting the $(k+1)$-th round, and denote the weight assigned to the $i$-th visit as $\tilde{\theta}_{t_k}^i$ with $t^k = N_h^{k+1}(x, a)$. Specifically, for visits within the same round $k'$, we index them by $i \in [N_h^{k'}(x, a) + 1 : N_h^{k'+1}(x, a)] := \mathcal{I}^{k'}$ according to a pre-defined order. Then, for all visits within the first case, $\tilde{\theta}_{t^k}^i$ equals to $\theta_{t^k}^i$, and for all visits within round $k'$ in the second case, $\tilde{\theta}_{t^k}^i = \sum_{i \in \mathcal{I}^{k'}} \theta_{t^k}^i / n_h^{k'}(x, a)$.

The proof mainly consists of two major steps. **Step 1** is to bound the global estimation error $Q_h^{k+1} - Q_h^\star$ for each round $k$. Based on the recursive updating rule, we can show that, with high probability,

$$0 \leq Q_h^{k+1}(x, a) - Q_h^\star(x, a) \leq \theta_{t^k}^0 H + \sum_{k'=1}^{k} \sum_{i \in \mathcal{I}^{k'}} \tilde{\theta}_{t^k}^i (V_{h+1}^{k'} - V_{h+1}^\star)(x_{h+1,i}) + \beta_{t^k},$$

with $\beta_{t^k} = \sum_{i=1}^{t^k} \theta_{t^k}^i b_i$. As shown in Lemma C.2, it suffices to bound $\left| \sum_{i=1}^{t^k} \tilde{\theta}_{t^k}^i X_i \right|$ where $X_i = V_{h+1}^\star(x_{h+1,i}) - \mathbb{E}\left[ V_{h+1}^\star(x_{h+1}) | (x_h, a_h) = (x, a) \right]$. Similar to the single-agent setting (Jin et al., 2018), $\{X_i\}_{i=1}^\infty$ is a martingale difference sequence. However, since our weight assignment for the $i$-th visit depends on the total number of visits in the same round, which is determined after that round completes, $\{\tilde{\theta}_{t^k}^i\}_i$ does not preserve the original martingale structure in $\{\theta_{t^k}^i\}_i$. Therefore, it necessitates novel approaches to bound the sum of **non-martingale differences**. We would like to emphasize that the techniques required to bound those non-martingale differences are fundamentally different from the commonly used techniques in federated learning (FL), which usually construct an "averaged parameter update path" and then bound each local term's "drift" from it. This is because such bounding techniques in FL rely on certain assumptions that do not exist in federated RL. Due to the inherent randomness in the environment, even if the same policy is taken at all local agents, it may result in very different trajectories. Thus, it is hard to obtain an easy-to-track "averaged parameter update path" in federated RL, or a tight bound on the local terms' drifts from such averaged parameter update path. We overcome this challenge by relating $\{\tilde{\theta}_{t^k}^i\}_i$ with $\{\theta_{t^k}^i\}_i$. Instead of bounding the local drift $\tilde{\theta}_{t^k}^i - \theta_{t^k}^i$ in each time step, we choose to group the "drift" terms based on the corresponding rounds and then leverage the round-wise equal weight assignment adopted in our algorithm to obtain a tight bound. The detailed analysis is elaborated in Lemma C.3.

Built upon the estimation error bound obtained in Step 1, **Step 2** then utilizes the recursive Bellman equation to relate the total learning error among all agents in round $k$ at step $h$ with that at step $h + 1$ (see Appendix C.3), which directly translates into a regret upper bound. The detailed proof is deferred to Appendix C. □

Next, we discuss the communication cost under Algorithms 1 and 2 as follows.

**Theorem 4.2** (Communication Cost). *Under Algorithms 1 and 2, for a given number of steps $T$, the total number of rounds must satisfy*

$$K \leq \max \left\{ \frac{HSA}{\log\left(1 + \frac{1}{2MH(H+1)}\right)} \log \frac{T}{H^2(H+1)M} + H^2(H+1)MSA, H^2(H+1)SAM \right\}.$$

Theorem 4.2 indicates that, when $T$ is sufficiently large, $K = O\left(MH^3SA\log(T/M)\right)$. Since the total number of communicated scalars is $O(MHS)$ in each round, the total communication cost scales in $O(M^2H^4S^2A\log(T/M))$.

---

²$\tilde{O}$ ignores logarithmic factors.

*Proof Sketch of Theorem 4.2.* Due to the fact that the equality in Equation (2) is met for at least one agent, by the Pigeonhole principle, during the first $K$ rounds, there exists one tuple $(x, a, h, m)$ such that the equality in Equation (2) holds for at least $\Omega(K/(HSAM))$ rounds. In these rounds, as $(x, a, h)$ are visited at least once in each round, at most $O(i_0)$ rounds belong to the first case. So, when $K$ is large, there are at least $\Omega(K/(HSAM))$ rounds in the second case and $n_h^k(x, a) = \hat{O}(N_h^k(x, a))$ in these rounds. Thus we have $HN_h^{K+1}(x, a)/M$, which is smaller than or equal to $T$, and roughly exponential in $K$ when $K$ is large. A detailed proof can be found in Appendix D. □

## 5 EXTENSION TO BERNSTEIN-TYPE ALGORITHM

The Bernstein-type algorithm differs from FedQ-Hoeffding on the construction of the upper confidence bound. Similar to the design in Jin et al. (2018), we define

$$\beta_t(x, a, h) = c' \min\left\{\sqrt{\frac{H\iota}{t}(W^t(x, a, h) + H)} + \iota\frac{\sqrt{H^7SA} + \sqrt{MSAH^6}}{t}, \sqrt{\frac{H^3\iota}{t}}\right\}, \quad (10)$$

in which $c' > 0$ is a positive constant and $W^t(x, a, h)$ is a variance estimator of $X_i$s whose specific form is introduced in Appendix E. FedQ-Bernstein then replaces $\beta^k$ in Equation (3) and Equation (4) by $\tilde{\beta} = \beta_{t^k}(x, a, h) - \alpha^c(t^{k-1} + 1, t^k)\beta_{t^{k-1}}(x, a, h)$. In terms of communication, during round $k$, in addition to all the quantities sent in Algorithm 2, each agent $m$ sends $\{\mu_h^{m,k}(x, \pi_h^k(x))\}_{x,h}$ to the central server where $\mu_h^{m,k}(x, a) = \frac{1}{n_h^{m,k}(x,a)}\sum_{j=1}^{n^k}\left[V_{h+1}^k\left(x_{h+1}^{m,k,j}\right)\right]^2 \mathbb{I}[(x_h^{m,k,j}, a_h^{m,k,j}) = (x, a)]$. The complete algorithm description can be found in Appendix E.

As FedQ-Bernstein uses tighter upper confidence bounds compared with FedQ-Hoeffding, it enjoys a reduced regret upper bound, as stated in Theorem 5.1 below.

**Theorem 5.1** (Regret Upper Bound for FedQ-Bernstein). *Let $\tilde{C} = 1/(H(H+1))$, $\iota = \max\{\iota_0, \iota_1\}$ with $\iota_0 = \log(2SA(T_0 + HM)(1 + \tilde{C})/p)$, $\iota_1 = \log\frac{2K_0SAH(T_0/H+M)(1+\tilde{C})}{p}$, and $p \in (0, 1)$. For Algorithms 3 and 4 in Appendix E with the upper confidence bound defined in Equation (10), there exists a constant $c' > 0$ such that, for any $K \in [K_0], p \in (0, 1)$, with probability at least $1 - p$,*

$$Regret(T) \leq O\left(MH^2SA + H^4SA(M-1) + HSA(M-1)\sqrt{H^3\iota}\right.$$
$$\left. + \iota^2\sqrt{H^9S^3A^3} + \iota^2\sqrt{MS^3A^3H^8} + \sqrt{H^3SAMT\iota^2}\right).$$

*Here, $T = HJ$ and $J$ is the total number of of episodes generated by an agent in the first $K$ rounds.*

Theorem 5.1 improves the regret upper bound in Theorem 4.1 by a factor of $\sqrt{H}$, and also enjoys a linear speedup in $M$ compared with its single-agent counterparts (Jin et al., 2018; Bai et al., 2019) when $T \geq \tilde{\Omega}(M\text{poly}(H, S, A))$ and the first term becomes the dominating term. Here $\tilde{\Omega}$ hides a log factor that takes the form $\log^2(MT\text{poly}(H, S, A)))$.

We also remark that the upper bound in Theorem 4.2 applies to FedQ-Bernstein as well. Since the amount of shared data is $O(MHS)$ in each round for both algorithms, FedQ-Bernstein has the same order of communication cost upper bound as FedQ-Hoeffding.

## 6 CONCLUSION

In this paper, we have developed model-free algorithms in federated reinforcement learning with provably linear regret speedup and logarithmic communication cost. More specifically, two federated $Q$-learning algorithms - FedQ-Hoeffding and FedQ-Bernstein - have been proposed, and we proved that they achieve regret of $\tilde{O}(\sqrt{H^4SAMT})$ and $\tilde{O}(\sqrt{H^3SAMT})$ respectively with communication cost $O(M^2H^4S^2A\log(T/M))$. Technically, our algorithm design features a novel equal weight assignment during global information aggregation, and we developed new approaches to characterizing the concentration properties for non-martingale differences, which could be of broader applications for other RL problems.

## ACKNOWLEDGMENTS

The work of Z. Zheng and L. Xue was supported in part by the U.S. National Science Foundation under the grants DMS-1811552, DMS-1953189, and CCF-2007823. The work of F. Gao and J. Yang was supported in part by the U.S. National Science Foundation under the grants CNS-1956276, CNS-2003131, and CNS-2114542.

## IMPACT STATEMENT

This paper presents work whose goal is to advance the field of reinforcement learning. There are potential societal consequences of our work but none of them must be specifically highlighted in our opinion.

## REPRODUCIBILITY

Numerical experiments in this paper can be fully reproduced via the publicly available code[3].

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

# A   RELATED WORKS

**Single-agent episodic MDPs.** Significant contributions have been made in both model-based and model-free frameworks. In the model-based category, a series of algorithms have been proposed by Auer et al. (2008), Agrawal & Jia (2017), Azar et al. (2017), Kakade et al. (2018), Agarwal et al. (2020), Dann et al. (2019), Zanette & Brunskill (2019), and Zhang et al. (2021), with more recent contributions from Zhou et al. (2023) and Zhang et al. (2023). Notably, Zhang et al. (2023) proved that a modified version of MVP (proposed by Zhang et al. (2021)) achieves a regret of $\tilde{O}\left(\min\{\sqrt{SAH^2T}, T\}\right)$ which matches the minimax lower bound. Within the model-free framework, Jin et al. (2018) proposed a Q-learning with UCB exploration algorithm, achieving regret of $\tilde{O}\left(\sqrt{SAH^3T}\right)$, which has been advanced further by Yang et al. (2021), Zhang et al. (2020), Li et al. (2021) and Ménard et al. (2021). The latter three have introduced algorithms that achieve minimax regret of $\tilde{O}\left(\sqrt{SAH^2T}\right)$.

**Federated and distributed RL**. Existing literature on federated and distributed RL algorithms sheds light on different aspects. Guo & Brunskill (2015) showed that applying concurrent RL to identical MDPs can linearly speed up sample complexity. Agarwal et al. (2021) proposed a parallel RL algorithm with low communication cost. Jin et al. (2022), Khodadadian et al. (2022), Fan et al. (2023) and Woo et al. (2023) investigated federated Q-learning algorithms in different settings. Fan et al. (2021), Wu et al. (2021) and Chen et al. (2023) focused on robustness. Particularly, Chen et al. (2023) proposed algorithms in both offline and online settings, obtaining near-optimal sample complexities and achieving a superior robustness guarantee. Doan et al. (2019), Doan et al. (2021), Chen et al. (2021b), Sun et al. (2020), Wai (2020), Wang et al. (2020a), Zeng et al. (2021) and Liu & Olshevsky (2023) analyzed the convergence of decentralized temporal difference algorithms. Fan et al. (2021) and Chen et al. (2021a) studied communication-efficient policy gradient algorithms. Shen et al. (2023b), Shen et al. (2023a) and Chen et al. (2022) have analyzed the convergence of distributed actor-critic algorithms. Assran et al. (2019), Espeholt et al. (2018) and Mnih et al. (2016) explored federated actor-learner architectures. Liu & Zhu (2022) and Liu & Zhu (2024) explored distributed inverse reinforcement learning.

**RL with low switching cost and batched RL**. Research in RL with low-switching cost aims to minimize the number of policy switching while maintaining comparable regret bounds to its fully adaptive counterparts and can be applied to federated RL. In batched RL (e.g., Perchet et al. (2016), Gao et al. (2019)), the agent sets the number of batches and length of each batch upfront, aiming for fewer batches and lower regret. Bai et al. (2019) first introduced the problem of RL with low-switching cost and proposed a $Q$-learning algorithm with lazy update, achieving $\tilde{O}(SAH^3 \log T)$ switching costs. This work was advanced by Zhang et al. (2020), which improved the regret upper bound. Besides, Wang et al. (2021) studied the problem of RL under the adaptivity constraint. Recently, Qiao et al. (2022) proposed a model-based algorithm with $\tilde{O}(\log \log T)$ switching costs. Zhang et al. (2022) proposed a batched RL algorithm that is well-suited for the federated setting.

**Federated/distributed bandits.** Federated bandits with low communication costs have been studied extensively recently in the literature Wang et al. (2020b); Li & Wang (2022); Shi & Shen (2021); Shi et al. (2021); Huang et al. (2021); Wang et al. (2022); He et al. (2022); Li et al. (2022b). Shi & Shen (2021) and Shi et al. (2021) investigated efficient client-server communication and coordination protocols for federated MAB without and with personalization, respectively. Wang et al. (2020b) investigated communication-efficient distributed linear bandits, while Huang et al. (2021) studied federated linear contextual bandits. Li & Wang (2022) focused on the asynchronous communication protocol.

When data privacy is explicitly considered, Li et al. (2020); Zhu et al. (2021) studied federated bandits with item-level differential privacy (DP) guarantee. Dubey & Pentland (2022) considered private and byzantine-proof cooperative decision making in multi-armed bandits. Dubey & Pentland (2020); Zhou & Chowdhury (2023) considered the linear contextual bandit model with joint DP guarantee. Huang et al. (2023) recently investigated linear contextual bandits under user-level DP constraints. Private distributed bandits with partial feedback was also studied in Li et al. (2022a).

# B AUXILIARY LEMMAS

In this section, we introduce some useful lemmas which will be used in the proofs. Before starting, we describe the global indexing mechanism mentioned in Section 4. Global visiting indices $i = 1, 2 \ldots$ are assigned, based on the chronological order, to the visits of any given $(x, a, h) \in \mathcal{S} \times \mathcal{A} \times [H]$. With this, we can establish a map between the global visiting index $i$, and $k, m, j$, where $k$ is the round index, $m$ is the agent index and $j$ is the episode index for a given round and a given agent. For $(x, a, h)$, we define functions that recover $k, m, j$ from $i$ as $k_h(i; x, a), m_h(i; x, a), j_h(i; x, a)$. When there is no ambiguity, we will use the simplified notations $k^i, m^i, j^i$. The visiting indices are utilized to construct a sequence, ensuring that quantities with smaller indices are observed prior to those with larger indices. Under the synchronization and zero-latency assumption, we have the following formulas for $m^i, k^i, j^i$.

$$k_h(i; x, a) = \sup \left\{ k \in \mathbb{N}_+ : N_h^k(x, a) < i \right\},$$

$$j_h(i; x, a) = \sup \left\{ j \in \mathbb{N}_+ : \sum_{j'=1}^{j-1} \sum_{m=1}^{M} \mathbb{I}\left[ (x, a) = (x_h^{m,k^i,j'}, a_h^{m,k^i,j'}) \right] < i - N_h^{k^i}(x, a) \right\},$$

$$m_h(i; x, a) = \sup \left\{ m \in \mathbb{N}_+ : \sum_{m'=1}^{m-1} \mathbb{I}\left[ (x, a) = (x_h^{m',k^i,j^i}, a_h^{m',k^i,j^i}) \right] \right.$$
$$\left. < i - N_h^{k^i}(x, a) - \sum_{j'=1}^{j^i-1} \sum_{m=1}^{M} \mathbb{I}\left[ (x, a) = (x_h^{m,k^i,j'}, a_h^{m,k^i,j'}) \right] \right\}.$$

We also introduce a new notation $\hat{T} = MT$ that represents the total number of samples generated by all the agents.

Next, we begin to introduce the lemmas. First, Lemma B.1 establishes some relationships between some quantities used in Algorithms 1 and 2.

**Lemma B.1.** *Denote $\tilde{C} = 1/(H(H+1))$. The following relationships hold for both algorithms.*

(a) $K \leq K_0$.

(b) $N_h^K(x, a) \leq T_0/H$.

(c) *For any $(x, a, h, k) \in \mathcal{S} \times \mathcal{A} \times [H] \times [K]$, we have*

$$n_h^{m,k}(x, a) \leq \max \left\{ 1, \frac{\tilde{C} N_h^k(x, a)}{M} \right\}, \forall m \in [M]. \tag{11}$$

*and*

$$n_h^k(x, a) \leq \max\{M, \tilde{C} N_h^k(x, a)\}. \tag{12}$$

*If $N_h^k(x, a) \geq i_0$,*

$$n_h^k(x, a) \leq \tilde{C} N_h^k(x, a).$$

(d) *For any $(x, a, h) \in \mathcal{S} \times \mathcal{A} \times [H]$, $N_h^{K+1}(x, a) \leq (1 + \tilde{C})T_0/H + M$.*

(e) $\hat{T} \leq (1 + \tilde{C})T_0 + HM$.

*Proof of Lemma B.1.* (a)-(c) are obvious given Algorithms 1 and 4. (d) and (e) can be directly obtained from (b) and (c). □

Next, Lemma B.2 provides some properties about $\theta_t^i$'s.

**Lemma B.2.** *(Lemma 4.1 in Jin et al. (2018) and beyond) The following properties hold for all $t \in \mathbb{N}_+$ for both algorithms.*

(a) $1/\sqrt{t} \le \sum_{i=1}^{t} \theta_t^i/\sqrt{i} \le 2/\sqrt{t}$, which implies that $\beta_t \in [2c\sqrt{H^3\iota/t}, 4c\sqrt{H^3\iota/t}]$, $\forall t \in \mathbb{N}_+$.

(b) $\max_{i \in [t]} \theta_t^i \le 2H/t$.

(c) $\sum_{i=1}^{t} \left(\theta_t^i\right)^2 \le 2H/t$.

(d) $\sum_{t=i}^{\infty} \theta_t^i = 1 + 1/H$.

(e) For any $t \in \mathbb{N}_+$ and $i \in [t] - \{t\}$, $\theta_t^{i+1}/\theta_t^i = 1 + H/i > 1$.

(f) For both algorithms, for any $t \in \mathbb{N}_+$ and $(x, a, h) \in \mathcal{S} \times \mathcal{A} \times [H]$, if $i_1, i_2 \in [t]$, $k_h(i_1, x, a) = k_h(i_2, x, a)$ and $N_h^{k_h(i_1, x, a)}(x, a) \ge i_0$ we have that $\theta_t^{i_1}/\theta_t^{i_2} \le \exp(1/H)$.

*Proof of Lemma B.2.* (a)-(e) are obvious based on $\theta_t^i$'s definition and Lemma 4.1 in Jin et al. (2018). For (f), denoting $t_0 = N_h^{k_h(i_1, x, a)}(x, a) + 1$ and $t_1 = N_h^{k_h(i_1, x, a)+1}(x, a)$, based on (e), we have

$$\theta_t^{i_1}/\theta_t^{i_2} \le \theta_t^{t_1}/\theta_t^{t_0} = \prod_{t'=t_0}^{t_1-1} (1 + H/t').$$

Based on (c) in Lemma B.1, we further have that

$$\prod_{t'=t_0}^{t_1-1} (1 + H/t') \le (1 + H/t_0)^{t_1-t_0} \le \exp(H(t_1 - t_0)/t_0) \le \exp(1/H).$$

$\square$

Next, we rigorously define the weights $\tilde{\theta}_t^i$ mentioned in Section 4. For any $(x, a, h, K') \in \mathcal{S} \times \mathcal{A} \times [H] \times [K]$, we let $t = N_h^{K'}(x, a)$ and $i \in [t] \bigcup \{0\}$. Letting $t' = N_h^{k^i}(x, a)$ and $t'' = N_h^{k^i+1}(x, a)$, we denote

$$\tilde{\theta}_t^i(x, a, h) = \theta_t^i \mathbb{I}[t' < i_0] + \frac{1 - \alpha^c(t'+1, t'')}{t'' - t'} \alpha^c(t''+1, t) \mathbb{I}[t' \ge i_0],$$

and we will use the simplified notation $\tilde{\theta}_t^i$ when there is no ambiguity. Lemma B.3 provides properties of $\tilde{\theta}_t^i$ and its relationship with $\theta_t^i$.

**Lemma B.3.** *The following relationships hold for any $(x, a, h, K') \in \mathcal{S} \times \mathcal{A} \times [H] \times [K]$ with $t = N_h^{K'}(x, a)$ for both algorithms.*

(a) $\tilde{\theta}_t^i(x, a, h) = \tilde{\theta}_{t'}^i(x, a, h) \alpha^c(t'+1, t)$ with $t' = N_h^{k_h(i;x,a)+1}(x, a)$.

(b) For any $i_1, i_2 \in [t]$, if $k_h(i_1, x, a) = k_h(i_2, x, a)$ and $N_h^{k_h(i_1, x, a)}(x, a) \ge i_0$, we have that $\tilde{\theta}_t^{i_1}(x, a, h) = \tilde{\theta}_t^{i_2}(x, a, h)$.

(c) For any $k' \le K'$, we have that

$$\sum_{i'=N_h^{k'}(x,a)+1}^{N_h^{k'+1}(x,a)} \tilde{\theta}_t^{i'}(x, a, h) = \sum_{i'=N_h^{k'}(x,a)+1}^{N_h^{k'+1}(x,a)} \theta_t^{i'},$$

which indicates that

$$\sum_{i=1}^{t} \tilde{\theta}_t^i = \mathbb{I}[t > 0].$$

(d) For any $i \in [t]$, when $N_h^{k_i(x,a,h)}(x, a) \ge i_0$, we have that

$$(1 + H/(1 + N_i))^{1-n_i} \le \tilde{\theta}_t^i/\theta_t^i \le (1 + H/(1 + N_i))^{n_i-1},$$

in which $N_i = N_h^{k_h(i,x,a)}(x, a)$ and $n_i = n_h^{k_h(i,x,a)}(x, a)$.

*(e) For any $i \in [t]$, when $N_h^{k_i(x,a,h)}(x,a) \geq i_0$, we have that*

$$\tilde{\theta}_t^i / \theta_t^i \leq \exp(1/H).$$

*Proof of Lemma B.3.* (a)-(c) can be obtained directly through the definition of $\tilde{\theta}_t^i$. Next, we prove (d) and (e). Denote $t_0 = N_h^{k_h(i,x,a)}(x,a) + 1$ and $t_1 = N_h^{k_h(i,x,a)+1}(x,a)$. By (c) and (e) in Lemma B.2, we have that $\theta_{t_1}^{t_0} / \theta_{t_1}^{t_1} \leq \tilde{\theta}_t^i / \theta_t^i \leq \theta_{t_1}^{t_1} / \theta_{t_1}^{t_0}$. Then, (d) can be proved by noticing that $\theta_{t_1}^{t_1} / \theta_{t_1}^{t_0} \leq (1 + H/(1 + N_i))^{n_i - 1}$. This implies that (e) holds because of (c) in Lemma B.1. $\quad\square$

## C PROOF OF THEOREM 4.1

### C.1 ROBUSTNESS AGAINST ASYNCHRONIZATION

In this subsection, we discuss a more general situation for Algorithms 1 and 2, where agent $m$ generates $n^{m,k}$ episodes during round $k$. We no longer assume that $n^{m,k}$ has the same value $n^k$ for different clients. The difference can be caused by latency (the time gap between an agent sending an abortion signal and other agents receiving the signal) and asynchronization (the heterogeneity among clients on the computation speed and process of collecting trajectories). In this case, for $K$ rounds, the total number of samples generated by all the clients is

$$\hat{T} = H \sum_{k=1}^{K} \sum_{m=1}^{M} n^{m,k}.$$

Thus, we generalize the notation $T = \hat{T}/M$, which characterizes the mean number of samples generated by an agent. Accordingly, the definition of $\text{Regret}(T)$ can be generalized as

$$\text{Regret}(T) = \sum_{k=1}^{K} \sum_{m=1}^{M} \sum_{j=1}^{n^{m,k}} V_1^\star(x_1^{m,k,j}) - V_1^{\pi^k}(x_1^{m,k,j}),$$

Similarly, the definitions of $n_h^{m,k}(x,a), N_h^{m,k}(x,a), v_h^{m,k}(x,a)$ are also generalized by replacing $\sum_{j=1}^{n^k}$ with $\sum_{j=1}^{n^{m,k}}$.

We note that Algorithms 1 and 2 naturally accommodate such asynchronicity. Therefore, in the following analysis of the regret, we adopt the general notation $n^{m,k}$. However, for the proof of Theorem 4.2 pertaining to communication, we will maintain the synchronization assumption that $n^{m,k} = n^k, \forall m \in [M]$.

### C.2 BOUNDS ON $Q_h^k - Q_h^\star$

**Lemma C.1.** *For Algorithms 1 and 2, there exists a positive constant $c > 0$ such that, for any $p \in (0,1)$, the following relationship holds for all $(x,a,h,K') \in \mathcal{S} \times \mathcal{A} \times [H] \times [K]$ with probability at least $1 - p$:*

$$0 \leq Q_h^{K'}(x,a) - Q_h^\star(x,a) \leq \theta_t^0 H + \sum_{i=1}^{t} \tilde{\theta}_t^i (V_{h+1}^{k^i} - V_{h+1}^\star)(x_{h+1}^{m^i,k^i,j^i}) + \beta_t, \quad (13)$$

*in which $t = N_h^{K'}(x,a)$.*

We first provide Lemma C.2 to formally state Equation (14) and Equation (15), which establish the relationship between $Q_h^k$ and $Q_h^\star$. The proof is the same as the proof of Equation (4.3) in Jin et al. (2018).

**Lemma C.2.** *For the Hoeffding-type Algorithms 1 and 2 , for all $(x, a, h, K') \in \mathcal{S} \times \mathcal{A} \times [H] \times [K]$, denoting $t = N_h^{K'}(x, a)$, we have*

$$
Q_h^{K'}(x, a) = \theta_t^0 H + \sum_{i=1}^t \tilde{\theta}_t^i \left( r_h(x, a) + V_{h+1}^{k^i}(x_{h+1}^{m^i, k^i, j^i}) \right) + \sum_{i=1}^t \theta_t^i b_i, \tag{14}
$$

$$
Q_h^{\star}(x, a) = \tilde{\theta}_t^0 Q_h^{\star} + \sum_{i=1}^t \tilde{\theta}_t^i \left( r_h(x, a) + \left( [\mathbb{P}_h V_{h+1}^{\star}](x, a) - \tilde{\mathbb{E}}_{x,a,h,i} V_{h+1}^{\star} \right) + \tilde{\mathbb{E}}_{x,a,h,i} V_{h+1}^{\star} \right).
$$

*Furthermore, we have*

$$
(Q_h^{K'} - Q_h^{\star})(x, a) = \tilde{\theta}_t^0 \left( H - Q_h^{\star}(x, a) \right) + \sum_{i=1}^t \tilde{\theta}_t^i (\tilde{\mathbb{E}}_{x,a,h,i} - \mathbb{E}_{x,a,h}) V_{h+1}^{\star}
$$

$$
+ \sum_{i=1}^t \tilde{\theta}_t^i (V_{h+1}^{k^i} - V_{h+1}^{\star})(x_{h+1}^{m^i, k^i, j^i}) + \sum_{i=1}^t \theta_t^i b_i, \tag{15}
$$

$$
(Q_h^{K'} - Q_h^{\star})(x, a) = \tilde{\theta}_t^0 \left( H - Q_h^{\star}(x, a) \right) + \sum_{i=1}^t \tilde{\theta}_t^i (\tilde{\mathbb{E}}_{x,a,h,i} - \mathbb{E}_{x,a,h}) V_{h+1}^{\star}
$$

*in which*

$$
\mathbb{E}_{x,a,h} V_{h+1}^{\star} = \mathbb{E}_{x,a,h} V_{h+1}^{\star}(x_{h+1}) = \mathbb{E}\left[ V_{h+1}^{\star}(x_{h+1}) | (x_h, a_h) = (x, a) \right],
$$

$$
\tilde{\mathbb{E}}_{x,a,h,i} V_{h+1}^{\star} = \tilde{\mathbb{E}}_{x,a,h,i} V_{h+1}^{\star}(x_{h+1}) = V_{h+1}^{\star}(x_{h+1}^{m^i, k^i, j^i}).
$$

With this lemma, we derive a probabilistic upper bound for $|\sum_{i=1}^t \tilde{\theta}_t^i X_i|$ with $X_i = (\tilde{\mathbb{E}}_{x,a,h,i} - \mathbb{E}_{x,a,h}) V_{h+1}^{\star}$ in Lemma C.3.

**Lemma C.3.** *There exists $c_0 > 0$ such that, for any $p \in (0, 1)$, with probability at least $1 - p$, the following relationship holds for all $(x, a, h, K') \in \mathcal{S} \times \mathcal{A} \times [H] \times [K]$ with $t = N_h^{K'}(x, a)$:*

$$
\left| \sum_{i=1}^t \tilde{\theta}_t^i (\tilde{\mathbb{E}}_{x,a,h,i} - \mathbb{E}_{x,a,h}) V_{h+1}^{\star}(x_{h+1}) \right| \leq c_0 \sqrt{H^3 \iota / t}. \tag{16}
$$

*Proof.* For a given $(x, a, h) \in \mathcal{S} \times \mathcal{A} \times [H]$, denote $X_i(x, a, h) = (\tilde{\mathbb{E}}_{x,a,h,i} - \mathbb{E}_{x,a,h}) V_{h+1}^{\star}(x_{h+1})$. When there is no ambiguity, we use the simplified notation $X_i = X_i(x, a, h)$. We know that $\{X_i\}_{i=1}^\infty$ is a sequence of martingale differences with $|X_i| \leq H$. We decompose the summation as follows:

$$
\sum_{i=1}^t \tilde{\theta}_t^i X_i = \sum_{i=1}^t \theta_t^i X_i + \sum_{i=1}^t (\tilde{\theta}_t^i - \theta_t^i) X_i.
$$

Note that $t \leq T_0/H$.

First, we focus on the first term. By Azuma-Hoeffding Inequality, for any given $(x, a, h) \in \mathcal{S} \times \mathcal{A} \times [H]$ and a given $t' \in \mathbb{N}_+$, for any $p \in (0, 1)$, with probability $1 - p$, there exists a numerical constant $c_1 > 0$ such that

$$
\left| \sum_{i=1}^{t'} \theta_{t'}^i X_i \right| \leq c_1 H \sqrt{ \left( \sum_{i=1}^{t'} (\theta_{t'}^i)^2 \right) \log \frac{2}{p} },
$$

which indicates that $\left| \sum_{i=1}^{t'} \theta_{t'}^i X_i \right| \leq \frac{c_1}{\sqrt{2}} \sqrt{(H^3/t') \log \frac{2}{p}}$ based on (c) in Lemma B.2.

By considering all the possible combinations $(x, a, h, t') \in \mathcal{S} \times \mathcal{A} \times [H] \times [T_0/H]$, with a union bound and the realization of $t = t'$, we have, for any $p \in (0, 1)$, with at probability at least $1 - p$, the following relationship holds simultaneously for all $(x, a, h, K') \in \mathcal{S} \times \mathcal{A} \times [H] \times [K]$:

$$
\left| \sum_{i=1}^t \theta_t^i X_i \right| \leq \frac{c_1}{\sqrt{2}} \sqrt{(H^3/t) \log \frac{2SAT_0}{p}} \leq \frac{c_1}{\sqrt{2}} \sqrt{\iota_0 H^3 / t}.
$$

We then focus on the second term $\sum_{i=1}^{t}(\tilde{\theta}_t^i - \theta_t^i)X_i$. For any given $(x, a, h, k_s) \in \mathcal{S} \times \mathcal{A} \times [H] \times [K]$, we consider the part with samples generated by the $k_s$-th round, which is

$$\sum_{i=t_2}^{t_3}(\tilde{\theta}_{t_3}^i - \theta_{t_3}^i)X_i,$$

in which $t_2 = N_h^{k_s}(x, a) + 1$, $t_3 = N_h^{k_s+1}(x, a)$. We can control the second term by controlling $|\sum_{i=t_2}^{t_3}(\tilde{\theta}_{t_3}^i - \theta_{t_3}^i)X_i|$ for all $k_s \in [K]$.

We have

$$\sum_{i=1}^{t}(\tilde{\theta}_t^i - \theta_t^i)X_i = \sum_{k_s=1}^{K'-1}\left[\prod_{t'=t_3+1}^{t}(1 - \alpha_{t'})\right]\sum_{i=t_2}^{t_3}(\tilde{\theta}_{t_3}^i - \theta_{t_3}^i)X_i. \tag{17}$$

To begin with, we prove that there exists a numerical constant $c_3 > 0$ such that

$$\sqrt{\sum_{i=t_2}^{t_3}(\tilde{\theta}_{t_3}^i - \theta_{t_3}^i)^2} \leq c_3 \sum_{t'=t_2}^{t_3} \theta_{t_3}^{t'}/\sqrt{t'}. \tag{18}$$

This relationship obviously holds when $N_h^{k_s}(x, a) < i_0$ as $LHS = 0$. When $N_h^{k_s}(x, a) \geq i_0$, we have

$$\sqrt{\sum_{i=t_2}^{t_3}(\tilde{\theta}_{t_3}^i - \theta_{t_3}^i)^2} \leq O\left(\sqrt{\sum_{i=t_2}^{t_3}\frac{H^2(t_3 - t_2)^2}{t_2^2}(\theta_{t_3}^i)^2}\right) \leq O\left(\frac{H(t_3 - t_2)^{3/2}}{t_2}\theta_{t_3}^{t_2}\right),$$

where the first inequality comes from (d) in Lemma B.3 and the second one comes from (f) in Lemma B.2.

We also have that

$$O\left(\frac{H(t_3 - t_2)^{3/2}}{t_2}\theta_{t_3}^{t_2}\right) = O\left(\frac{H(t_3 - t_2)^{1/2}}{\sqrt{t_2}}(t_3 - t_2)\theta_{t_3}^{t_2}/\sqrt{t_2}\right) = O\left(\sum_{t'=t_2}^{t_3}\theta_{t_3}^{t'}/\sqrt{t'}\right),$$

where the second relationship comes from (f) in Lemma B.2. This completes the proof of Equation (18).

Next, we proceed with discussions conditioning on all the information before starting the $k_s$-th round, which means that $N_h^{k_s}(x, a)$ and $t_2$ can be treated as constants. If $N_h^{k_s}(x, a) < i_0$, this quantity is equal to 0. Otherwise, given any $t_3' \geq t_2$ and $i \in [t_2, t_3']$, we denote

$$\hat{\theta}_{t_3'} = \left[1 - \prod_{t'=t_2}^{t_3'}(1 - \alpha_{t'})\right]/(t_3' - t_2 + 1),$$

Therefore, in the expression $\sum_{i=t_2}^{t_3'}(\hat{\theta}_{t_3'} - \theta_{t_3'}^i)X_i$, we can treat $\{X_{t_2}, X_{t_2+1} \ldots X_{t_3'}\}$ as martingale differences and $\theta_{t_3'}^i$s and $\hat{\theta}_{t_3'}$ as constants. Hence, by Azuma-Hoeffding Inequality, there exists a positive numerical constant $c_2$ such that, for any $p \in (0, 1)$, with probability at least $1 - p$,

$$\left|\sum_{i=t_2}^{t_3'}(\hat{\theta}_{t_3'} - \theta_{t_3'}^i)X_i\right| \leq c_2 H\sqrt{\log\frac{2}{p}\sum_{i=t_2}^{t_3'}(\hat{\theta}_{t_3'} - \theta_{t_3'}^i)^2}.$$

By considering all possible values of $t_3'$ with a union bound, we have that with probability at least $1 - p$, the following relationship holds simultaneously for any $t_2 \leq t_3 \leq t_2 + (1 + \tilde{C})T_0/H + M - 1$.

$$\left|\sum_{i=t_2}^{t_3'}(\hat{\theta}_{t_3'} - \theta_{t_3'}^i)X_i\right| \leq c_2 H\sqrt{\log\frac{2(T_0/H + M)(1 + \tilde{C})}{p}\sum_{i=t_2}^{t_3'}(\hat{\theta}_{t_3'} - \theta_{t_3'}^i)^2}.$$

So, noticing that $\hat{\theta}_{t'_3} = \tilde{\theta}^i_{t_3}$ when $t'_3 = t_3$ and $i \in [t_2, t_3]$ and applying Equation (18), we have that, for any $k_s \in \mathbb{N}_+$ and any $p \in (0, 1)$, with probability at least $1 - p$,

$$\left| \sum_{i=t_2}^{t_3} (\tilde{\theta}^i_{t_3} - \theta^i_{t_3}) X_i \right| \leq c_2 c_3 H \sqrt{\log \frac{2(T_0/H + M)(1 + \tilde{C})}{p}} \sum_{i=t_2}^{t_3} \theta^i_{t_3}/\sqrt{i}.$$

We apply the union bound and claim that for any $p \in (0, 1)$, the following relationship holds with probability at least $1 - p$ for all $(x, a, h, k_s) \in \mathcal{S} \times \mathcal{A} \times [H] \times [K_0]$.

$$\left| \sum_{i=t_2}^{t_3} (\tilde{\theta}^i_{t_3} - \theta^i_{t_3}) X_i \right| \leq c_2 c_3 H \sqrt{\log \frac{2SAHK_0(T_0/H + M)(1 + \tilde{C})}{p}} \sum_{i=t_2}^{t_3} \theta^i_{t_3}/\sqrt{i}$$

$$= c_2 c_3 H \sqrt{\iota_1} \sum_{i=t_2}^{t_3} \theta^i_{t_3}/\sqrt{i}.$$

Under this event, with Equation (17), we have that

$$\left| \sum_{i=1}^{t} (\tilde{\theta}^i_t - \theta^i_t) X_i \right| \leq c_2 c_3 H \sum_{i=1}^{t} \sqrt{\iota_1} \theta^i_t/\sqrt{i}. \tag{19}$$

By (a) in Lemma B.2, we have $\sum_{i=1}^{t} \theta^i_t/\sqrt{i} \leq \sqrt{4/t}$. Combining the results for the two terms completes the proof. $\qquad \square$

Finally, we provide the proof for Lemma C.1.

*Proof of Lemma C.1.* We pick $c = c_0$ such that the event in Lemma C.3 holds. Under the event given in Lemma C.3 and noting Equation (15), we claim the conclusion by using the same proof as that for Lemma 4.3 in Jin et al. (2018). $\qquad \square$

## C.3 PROOF OF THEOREM 4.1

Having proved Lemma C.1, we turn our attention to demonstrating the remaining parts of the proof. We use $n^{m,k}$ to denote the number of episodes by agent $m$ in round $k$.

We first provide some additional notations. Define

$$\delta^k_h = \sum_{m=1}^{M} \sum_{j=1}^{n^{m,k}} \left( V^k_h - V^{\pi^k}_h \right) (x^{m,k,j}_h),$$

$$\phi^k_h = \sum_{m=1}^{M} \sum_{j=1}^{n^{m,k}} \left( V^k_h - V^{\star}_h \right) (x^{m,k,j}_h), \forall h \in [H+1],$$

in which $\delta^k_{H+1} = \phi^k_{H+1} = 0$. We also define

$$\xi^k_{h+1} = \sum_{m=1}^{M} \sum_{j=1}^{n^{m,k}} (\mathbb{P} - \hat{\mathbb{P}}) \left( V^{\star}_{h+1} - V^{\pi^k}_{h+1} \right) (x^{m,k,j}_h, a^{m,k,j}_h), h \in [H]$$

with $\xi^k_{H+1} = 0$. Here,

$$(\mathbb{P}) \left( V^{\star}_{h+1} - V^{\pi^k}_{h+1} \right) (x^{m,k,j}_h, a^{m,k,j}_h) = \mathbb{E} \left[ \left( V^{\star}_{h+1} - V^{\pi^k}_{h+1} \right) (x^{m,k,j}_{h+1}) | (\pi^k, x^{m,k,j}_h, a^{m,k,j}_h) \right],$$

and

$$(\hat{\mathbb{P}}) \left( V^{\star}_{h+1} - V^{\pi^k}_{h+1} \right) (x^{m,k,j}_h, a^{m,k,j}_h) = \left( V^{\star}_{h+1} - V^{\pi^k}_{h+1} \right) (x^{m,k,j}_{h+1}).$$

We first provide a Lemma related to $\xi^k_{h+1}$.

**Lemma C.4.** *There exists a numerical constant $c_5 > 0$ such that, for any $p \in (0, 1)$, with probability at least $1 - p$,*

$$\left| \sum_{k=1}^{K} C_h \sum_{h=1}^{H} \xi_{h+1}^{k} \right| \le c_5 H \sqrt{\hat{T} \iota}, \tag{20}$$

*where $C_h = \exp(3(h-1)/H)$.*

*Proof.* Denote $V(m, k, j, h) = C_h (\mathbb{P} - \hat{\mathbb{P}}) \left( V_{h+1}^{\star} - V_{h+1}^{\pi^k} \right) (x_h^{m,k,j}, a_h^{m,k,j})$ and use $\sum_{m,k,j,h}$ as a simplified notation for $\sum_{k=1}^{K} \sum_{m=1}^{M} \sum_{j=1}^{n^{m,k}} \sum_{h=1}^{H-1}$. The quantity of interest can be rewritten as $\sum_{m,k,j,h} V(m, k, j, h)$, with $|V(m, k, j, h)| \le O(H)$ as $C_h \le \exp(3)$.

Let $\tilde{V}(\tilde{i})$ be the $\tilde{i}$-th term in the summation that contains $\hat{T}(H-1)/H$ terms, in which the order follows a "round first, episode second, step third, agent fourth" rule. Then the sequence $\{\tilde{V}(\tilde{i})\}$ is a martingale difference. By Azuma-Hoeffding Inequality, for any $p \in (0, 1)$ and $t \in N_{+}$, with probability at least $1 - p$,

$$\left| \sum_{\tilde{i}=1}^{t} \tilde{V}(\tilde{i}) \right| \le O \left( H \sqrt{t \log \frac{2}{p}} \right).$$

Then by applying a union bound over $t \in [(1 + \tilde{C})T_0 + HM]$ and knowing that $\hat{T}(H-1)/H \le T_0(1 + \tilde{C}) + HM$ due to (e) in Lemma B.1, we have that, for any $p \in (0, 1)$, with probability at least $1 - p$,

$$\left| \sum_{k=1}^{K} C_h \sum_{h=1}^{H} \xi_{h+1}^{k} \right| = \left| \sum_{\tilde{i}=1}^{\hat{T}(H-1)/H} \tilde{V}(\tilde{i}) \right| \le O(H \sqrt{\hat{T} \iota}).$$

This completes the proof. $\qquad \square$

Noticing that $\text{Regret}(T) \le \sum_{k=1}^{K} \delta_1^k$ due to $\text{Regret}(T) = \sum_{k=1}^{K} \delta_1^k - \sum_{k=1}^{K} \phi_1^k$ and $\phi_1^k \ge 0$ shown in Equation (13), we attempt to establish a probability upper bound for $\sum_{k=1}^{K} \delta_1^k$. First, we have

$$\delta_h^k \le \sum_{m=1}^{M} \sum_{j=1}^{n^{m,k}} (Q_h^k - Q_h^{\star})(x_h^{m,k,j}, a_h^{m,k,j}) + \sum_{m=1}^{M} \sum_{j=1}^{n^{m,k}} (Q_h^{\star} - Q_h^{\pi^k})(x_h^{m,k,j}, a_h^{m,k,j}), \tag{21}$$

which holds because $V_h^{\pi^k}(x_h^{m,k,j}) = Q_h^{\pi^k}(x_h^{m,k,j}, a_h^{m,k,j})$ and

$$V_h^k \left( x_h^{m,k,j} \right) \le \max_{a' \in \mathcal{A}} Q_h^k \left( x_h^{m,k,j}, a' \right) = Q_h^k \left( x_h^{m,k,j}, a_h^{m,k,j} \right).$$

Next, we attempt to bound the terms in RHS of Equation (21) separately. Our discussions are based on the events outlined in Lemma C.1. For any given $h$, denote $t_h^{m,k,j} = N_h^k(x_h^{m,k,j}, a_h^{m,k,j})$ and the corresponding $k, m, j$ (round index, agent index, and episode index) for the $i$-th global visiting for $(x_h^{m,k,j}, a_h^{m,k,j}, h)$ are $k_{i,h}^{m,k,j}, m_{i,h}^{m,k,j}, j_{i,h}^{m,k,j}, i = 1, 2 \ldots t_h^{m,k,j}$. For the first term, due to Equation (13), we have

$$\sum_{m=1}^{M} \sum_{j=1}^{n^{m,k}} (Q_h^k - Q_h^{\star})(x_h^{m,k,j}, a_h^{m,k,j})$$

$$\le \sum_{m=1}^{M} \sum_{j=1}^{n^{m,k}} \tilde{\theta}_{t_h^{m,k,j}}^{0} H + \sum_{m=1}^{M} \sum_{j=1}^{n^{m,k}} \sum_{i=1}^{t_h^{m,k,j}} \tilde{\theta}_{t_h^{m,k,j}}^{i} (V_{h+1}^{k_{i,h}^{m,k,j}} - V_{h+1}^{\star})(x_{h+1}^{(m,k,j)_{i,h}^{m,k,j}})$$

$$+ \sum_{m=1}^{M} \sum_{j=1}^{n^{m,k}} \beta_{t_h^{m,k,j}}. \tag{22}$$

For the second term, due to Equation (1),

$$\sum_{m=1}^{M}\sum_{j=1}^{n^{m,k}}(Q_h^\star - Q_h^{\pi^k})(x_h^{m,k,j}, a_h^{m,k,j}) = \delta_{h+1}^k - \phi_{h+1}^k + \xi_{h+1}^k. \tag{23}$$

Next, we try to find some bounds related to $\sum_{k=1}^{K}\delta_h^k$. For notation simplicity, we use $\sum_{m,k,j}$ to represent $\sum_{k=1}^{K}\sum_{m=1}^{M}\sum_{j=1}^{n^{m,k}}$. We can prove the following relationships with details referred to Appendix C.4:

$$\sum_{m,k,j}\tilde{\theta}_{t_h^{m,k,j}}^0 H \le MHSA. \tag{24}$$

$$\sum_{m,k,j}\sum_{i=1}^{t_h^{m,k,j}}\theta_{t_h^{m,k,j}}^i(V_{h+1}^{k_i^{m,k,j}} - V_{h+1}^\star)(x_{h+1}^{(m,k,j)_i^{m,k,j}}) \le e^{3/H}\sum_{k=1}^{K}\phi_{h+1}^k + O(H^3 SA(M-1)). \tag{25}$$

$$\sum_{k=1}^{K}\sum_{m=1}^{M}\sum_{j=1}^{n_h^{m,k}}\beta_{t_h^{m,k,j}} \le O(\sqrt{H^2\iota\hat{T}SA} + SA(M-1)\sqrt{H^3\iota}). \tag{26}$$

Combining Equations (22) to (26), we have that for any $h \in [H]$,

$$\sum_{k=1}^{K}\delta_h^k \le \exp(3/H)\sum_{k=1}^{K}\phi_{h+1}^k + \sum_{k=1}^{K}\delta_{h+1}^k - \sum_{k=1}^{K}\phi_{h+1}^k + \sum_{k=1}^{K}\xi_{h+1}^k$$
$$+ O\left(\sqrt{H^2\iota\hat{T}SA} + SA(M-1)\sqrt{H^3\iota} + MHSA + H^3 SA(M-1)\right).$$

Noticing that $\delta_h^k \ge \phi_h^k, \forall (h,k) \in [H] \times [K]$ due to the optimality of $\pi^\star$ and $\exp(3/H)^H = O(1)$, by recursions on $1, 2 \dots H$, we have

$$\sum_{k=1}^{K}\delta_1^k \le \sum_{h=1}^{H-1}C_h\sum_{k=1}^{K}\xi_{h+1}^k$$
$$+ O\left(\sqrt{H^4\iota\hat{T}SA} + HSA(M-1)\sqrt{H^3\iota} + MH^2 SA + H^4 SA(M-1)\right),$$

in which $C_h = \exp(3(h-1)/H)$. With Lemma C.4, we can also show that, with high probability,

$$\left|\sum_{k=1}^{K}C_h\sum_{h=1}^{H}\xi_{h+1}^k\right| \le O(H\sqrt{\hat{T}\iota}). \tag{27}$$

This indicates that $\sum_{k=1}^{K}\delta_1^k = O(\sqrt{H^4\iota\hat{T}SA} + HSA(M-1)\sqrt{H^3\iota} + MH^2 SA + H^4 SA(M-1))$. With these discussions, we have already shown that, under the intersection of events given in Lemma C.1 and Lemma C.4,

$$\text{Regret}(T) \le \sum_{k=1}^{K}\delta_1^k$$
$$\le O\left(\sqrt{H^4\iota\hat{T}SA} + HSA(M-1)\sqrt{H^3\iota} + MH^2 SA + H^4 SA(M-1)\right).$$

By replacing $p$ for the events in Lemma C.1 and Lemma C.4 with $p/2$, we finish the proof.

### C.4 PROOFS OF EQUATIONS (24) TO (26)

In this subsection, we try to give bounds the terms in RHS of Equation (21) separately. We make discussions based on the intersection of events given in Lemma C.1 and Lemma C.4. Under these events, we have already shown that Equation (22), Equation (23) and Equation (27) hold. So, we will provide the proof by establishing Equations (24) to (26).

**Proof of Equation (24).** First, we note that

$$\sum_{m,k,j} \tilde{\theta}^0_{t_h^{m,k,j}} H \le \sum_{m,k,j} H \cdot \mathbb{I}[t_h^{m,k,j} = 0].$$

For each $(x, a, h) \in \mathcal{S} \times \mathcal{A} \times [H]$, we consider all the rounds indexed as $0 < k_1 < k_2 < ...$ satisfying the condition $n_h^k(x, a) > 0$. Here, $k_s$s are simplified notations for functions of $(x, a, h)$, and we use the simplified notations when there is no ambiguity and the stated meaning of these notations is applicable only to the proof of Equation (24). So,

$$\sum_{m,k,j} \mathbb{I}[t_h^{m,k,j} = 0]\mathbb{I}[(x_h^{m,k,j}, a_h^{m,k,j}) = (x, a)] \le n_h^{k_1}(x, a).$$

As $N_h^{k_1}(x, a) = 0$, due to Equation (12), we have $n_h^{k_1}(x, a) \le M$. Therefore,

$$\sum_{m,k,j} \tilde{\theta}^0_{t_h^{m,k,j}} H = \sum_{(x,a) \in \mathcal{S} \times \mathcal{A}} \sum_{m,k,j} H\mathbb{I}[t_h^{m,k,j} = 0]\mathbb{I}[(x_h^{m,k,j}, a_h^{m,k,j}) = (x, a)] \le MHSA.$$

This completes the proof for Equation (24). $\qquad\square$

**Proof of Equation (25).** We denote $i_1 = (M-1)H(H+1)$ and split the summation into two parts:

$$\sum_{k,m,j} \sum_{i=1}^{t_h^{m,k,j}} \tilde{\theta}^i_{t_h^{m,k,j}} (V_{h+1}^{k_{i,h}^{m,k,j}} - V_{h+1}^\star)(x_{h+1}^{(m,k,j)_{i,h}^{m,k,j}})$$

$$= \sum_{m,k,j} \mathbb{I}[t_h^{m,k,j} \le i_1] \sum_{i=1}^{t_h^{m,k,j}} \tilde{\theta}^i_{t_h^{m,k,j}} (V_{h+1}^{k_{i,h}^{m,k,j}} - V_{h+1}^\star)(x_{h+1}^{(m,k,j)_{i,h}^{m,k,j}})$$

$$+ \sum_{m,k,j} \mathbb{I}[t_h^{m,k,j} > i_1] \sum_{i=1}^{t_h^{m,k,j}} \tilde{\theta}^i_{t_h^{m,k,j}} (V_{h+1}^{k_{i,h}^{m,k,j}} - V_{h+1}^\star)(x_{h+1}^{(m,k,j)_{i,h}^{m,k,j}}).$$

To bound the first term, we first notice that

$$\sum_{m,k,j} \mathbb{I}[t_h^{m,k,j} \le i_1] \sum_{i=1}^{t_h^{m,k,j}} \tilde{\theta}^i_{t_h^{m,k,j}} (V_{h+1}^{k_{i,h}^{m,k,j}} - V_{h+1}^\star)(x_{h+1}^{(m,k,j)_{i,h}^{m,k,j}}) \le H \cdot \sum_{m,k,j} \mathbb{I}[0 < t_h^{m,k,j} \le i_1]$$

due to the fact that $(V_{h+1}^{k_{i,h}^{m,k,j}} - V_{h+1}^\star)(x_{h+1}^{(m,k,j)_{i,h}^{m,k,j}}) \le H$ and $\sum_{i=1}^{t_h^{m,k,j}} \tilde{\theta}^i_{t_h^{m,k,j}} = \mathbb{I}[t_h^{m,k,j} > 0]$ given in (c) in Lemma B.3. For every $(x, a, h) \in \mathcal{S} \times \mathcal{A} \times [H]$, suppose that $k'$ is the round index such that $N_h^{k'}(x, a) \le i_1$ and $N_h^{k'+1}(x, a) > i_1$, and $k''$ is the round index such that $N_h^{k''}(x, a) = 0$ and $N_h^{k''+1}(x, a) > 0$. Here, $k'$ and $k''$ are simplified notations for functions of $(x, a, h)$, and we use the simplified notations when there is no ambiguity and the stated meaning is only valid in the proof of Equation (25).

We have

$$\sum_{m,k,j} \mathbb{I}[0 < t_h^{m,k,j} \le i_1]\mathbb{I}[(x_h^{m,k,j}, a_h^{m,k,j}) = (x, a)] \le i_1 + n_h^{k'}(x, a) - n_h^{k''}(x, a).$$

As $n_h^{k''}(x, a) \ge 1$ and $n_h^{k'}(x, a) \le M$ due to $N_h^{k''}(x, a) < i_0$ and Equation (12),

$$\sum_{m,k,j} \mathbb{I}[0 < t_h^{m,k,j} \le i_1]\mathbb{I}[(x_h^{m,k,j}, a_h^{m,k,j}) = (x, a)] \le i_1 + (M - 1) = O\left(H^2(M - 1)\right).$$

So,

$$\sum_{m,k,j} \mathbb{I}[t_h^{m,k,j} \le i_1] \sum_{i=1}^{t_h^{m,k,j}} \tilde{\theta}^i_{t_h^{m,k,j}} (V_{h+1}^{k_{i,h}^{m,k,j}} - V_{h+1}^\star)(x_{h+1}^{(m,k,j)_{i,h}^{m,k,j}})$$

$$\le H \sum_{(x,a) \in \mathcal{S} \times \mathcal{A}} \sum_{m,k,j} \mathbb{I}[0 < t_h^{m,k,j} \le i_1]\mathbb{I}[(x_h^{m,k,j}, a_h^{m,k,j}) = (x, a)]$$

$$= O\left(H^3 SA(M - 1)\right). \tag{28}$$

To bound the second term, we first notice that $V_{h+1}^{k_{i,h}^{m,k,j}} - V_{h+1}^\star \geq 0$ due to Equation (13). Then we regroup the summations in a different way. For every $(m', k', j')$, the term $(V_{h+1}^{k'} - V_{h+1}^\star)(x_{h+1}^{m',k',j'})$ appears in the term $\mathbb{I}[t_h^{m,k,j} > i_1] \sum_{i=1}^{t_h^{m,k,j}} \theta_{t_h^{m,k,j}}^i (V_{h+1}^{k_{i,h}^{m,k,j}} - V_{h+1}^\star)(x_{h+1}^{(m,k,j)_{i,h}^{m,k,j}})$ for $(k, m, j)$ if and only if $k > k'$ and $(x_h^{m,k,j}, a_h^{m,k,j}) = (x_h^{m',k',j'}, a_h^{m',k',j'})$. Thus, for each $(m', k', j')$, we denote $(x, a) = (x_h^{m',k',j'}, a_h^{m',k',j'})$. We consider all the later round indices $k' = k_0 < k_1 < k_2 < ...$ that satisfy $n_s = n_h^{k_s}(x, a) > 0, s \in \mathbb{N}$. Here, $k_s$'s are simplified notations for functions of $(m', k', j', h)$, and we use the simplified notations when there is no ambiguity and the stated meaning is only valid in proof of Equation (25). Then, the coefficient of summation related to $(m', k', j')$ can be upper bounded by

$$\sum_{s=1}^\infty \mathbb{I}[N_h^{k_s}(x, a) > i_1] n_s \tilde{\theta}_{N_h^{k_s}(x,a)}^{i'},$$

in which $i'$ is the global visiting number for $(x, a, h)$ at $(m', k', j')$, which means that $(m', k', j') = m_h(i'; x, a), k_h(i'; x, a), j_h(i'; x, a)$.

First, based on (e) in Lemma B.3, we have that

$$\sum_{s=1}^\infty \mathbb{I}[N_h^{k_s}(x, a) > i_1] n_s \tilde{\theta}_{N_h^{k_s}(x,a)}^{i'} \leq \exp(1/H) \sum_{s=1}^\infty \mathbb{I}[N_h^{k_s}(x, a) > i_1] n_s \theta_{N_h^{k_s}(x,a)}^{i'}.$$

We know that, $N_h^{k_s}(x, a) + n_s = N_h^{k_{s+1}}(x, a)$, $i' \leq N_h^{k_1}(x, a)$, and by (d) in Lemma B.2, $\sum_{i=i'}^\infty \theta_i^{i'} = (1 + 1/H)$. Therefore, if we can find $C' > 1$ such that

$$C' \geq \max_{(i', i'') \in \tilde{A}} \frac{\theta_{N_h^{k_s}(x,a)}^{i'}}{\theta_{N_h^{k_s}(x,a)+i''}^{i'}}, \forall (x, a, h, s) \in \mathcal{S} \times \mathcal{A} \times [H] \times \mathbb{N},$$

where $\tilde{A} = \{(i', i'') \in \mathbb{N}^2 : N_h^{k_s}(x, a) > i_1, 0 < i'' < n_s\}$, we can have

$$\sum_{s=1}^\infty \mathbb{I}[N_h^{k_s}(x, a) > i_1] n_s \tilde{\theta}_{N_h^{k_s}(x,a)}^{i'} \leq C' \exp(2/H).$$

Next, we prove that, for any $(i', i'') \in \tilde{A}$, if $N_h^{k_s}(x, a) > i_1$,

$$\frac{\theta_{N_h^{k_s}(x,a)}^{i'}}{\theta_{N_h^{k_s}(x,a)+i''}^{i'}} \leq \frac{\theta_{N_h^{k_s}(x,a)}^{i'}}{\theta_{N_h^{k_s}(x,a)+n_s-1}^{i'}}$$

$$= \prod_{d=N_h^{k_s}(x,a)+1}^{N_h^{k_s}(x,a)+n_s-1} (1 - \alpha_d)^{-1}$$

$$\leq (1 - \alpha_{d_0})^{1-n_s}$$

$$\leq \exp(1/H),$$

in which $d_0 = N_h^{k_s}(x, a) + 1$ so that we can let $C' = \exp(1/H)$. The first inequality holds because $\theta_t^i = \alpha_i \prod_{i'=i+1}^t (1 - \alpha_{i'})$ is a decreasing function with respect to $t$. The equality follows from the definition of $\theta_t^i$. The second inequality holds because $\alpha_t = \frac{H+1}{H+t}$ is a decreasing function with respect to $t$. Then we focus on the last inequality. According to the definition of $\alpha_t$, we have

$$(1 - \alpha_{d_0})^{1-n_s} = \left(1 - \frac{H+1}{H + N_h^{k_s}(x,a) + 1}\right)^{1-n_s} = \left(1 + \frac{H+1}{N_h^{k_s}(x,a)}\right)^{n_s-1}.$$

If $N_h^{k_s}(x, a) > MH(H + 1)$, according to Equation (12), we have that $n_s \leq \frac{N_h^{k_s}(x,a)}{H(H+1)}$. Then we have

$$\left(1 + \frac{H+1}{N_h^{k_s}(x,a)}\right)^{n_s-1} \leq \left(1 + \frac{H+1}{N_h^{k_s}(x,a)}\right)^{\frac{N_h^{k_s}(x,a)}{H(H+1)}} \leq \exp(1/H).$$

If $i_1 < N_h^{k_s}(x,a) \le MH(H+1)$, we can prove that

$$
\left(1 + \frac{H+1}{N_h^{k_s}(x,a)}\right)^{n_s - 1} \overset{(a)}{\le} \left(1 + \frac{H+1}{N_h^{k_s}(x,a)}\right)^{M-1}
$$

$$
\overset{(b)}{<} \left(1 + \frac{1}{H(M-1)}\right)^{M-1}
$$

$$
\le \exp(1/H)
$$

where $(a)$ holds because according to Equation (12) we have $n_s \le M$ and $(b)$ holds because $N_h^{k_i}(x,a) > (M-1)H(H+1)$.

Putting the two cases together, we have

$$
\sum_{s=1}^{\infty} \mathbb{I}[N_h^{k_s}(x,a) > i_1] n_s \tilde{\theta}_{N_h^{k_s}(x,a)}^{i'} \le \exp(1/H)\exp(2/H) \le \exp(3/H).
$$

Then we conclude that

$$
\sum_{m,k,j} \mathbb{I}[t_h^{m,k,j} > i_1] \sum_{i=1}^{t_h^{m,k,j}} \tilde{\theta}_{t_h^{m,k,j}}^{i}(V_{h+1}^{k_{i,h}^{m,k,j}} - V_{h+1}^{\star})(x_{h+1}^{(m,k,j)_{i,h}^{m,k,j}})
$$

$$
\le \exp\left(\frac{3}{H}\right) \sum_{m',k',j'} \left(V_{h+1}^{k'} - V_{h+1}^{\star}\right)(x_{h+1}^{m',k',j'})
$$

$$
= \exp\left(\frac{3}{H}\right) \sum_{k=1}^{K} \phi_{h+1}^{k}.
$$

Combining with Equation (28), we complete the proof for Equation (25). □

**Proof of Equation (26).** We split the summation into two parts:

$$
\sum_{m,k,j} \beta_{t_h^{m,k,j}} = \sum_{m,k,j} \beta_{t_h^{m,k,j}} \mathbb{I}[0 < t_h^{m,k,j} \le M-1] + \sum_{m,k,j} \beta_{t_h^{m,k,j}} \mathbb{I}[t_h^{m,k,j} \ge M].
$$

For every pair $(x,a,h)$, we consider all the rounds indexed as $0 < k_1 < k_2 < ...$ satisfying the condition $n_s = n_h^{k_s}(x,a) > 0$. Suppose that $k_p$ is the round index such that $N_h^{k_p}(x,a) \le M-1$ and $N_h^{k_{p+1}(x,a,h)}(x,a) > M-1$. Here, $k_s$s and $p$ are simplified notations for functions of $(x,a,h)$, and we use the simplified notations when there is no ambiguity and the stated meaning is only valid in the proof of Equation (26).

To bound the first term, we use the fact that $\beta_{t_h^{m,k,j}} \le O(1)\sqrt{H^3\iota}$. Then we have

$$
\sum_{m,k,j} \beta_{t_h^{m,k,j}} \mathbb{I}[0 < t_h^{m,k,j} \le M-1] \le O(1)\sqrt{H^3\iota} \sum_{m,k,j} \mathbb{I}[0 < t_h^{m,k,j} \le M-1]
$$

and

$$
\sum_{m,k,j} \mathbb{I}[0 < t_h^{m,k,j} \le M-1] \le \sum_{(x,a)\in\mathcal{S}\times\mathcal{A}} \left(M - 1 + n_h^{k_p}(x,a) - n_h^{k_1}(x,a)\right).
$$

It is straightforward that $n_h^{k_1}(x,a) \ge 1$. Additionally, due to Equation (12), we can establish that $n_h^{k_p}(x,a) \le M$. Therefore

$$
\sum_{m,k,j} \mathbb{I}[0 < t_h^{m,k,j} \le M-1] \le 2SA(M-1)
$$

and

$$
\sum_{m,k,j} \beta_{t_h^{m,k,j}} \mathbb{I}[0 < t_h^{m,k,j} \le M-1] = O(SA(M-1)\sqrt{H^3\iota}). \tag{29}
$$

To establish bounds for the second term, we define some notions first. For every pair $(x, a, h) \in \mathcal{S} \times \mathcal{A} \times [H]$, we consider all the rounds indexed as $0 < \tilde{k}_1 < \tilde{k}_2 < ... < \tilde{k}_g \leq K$ satisfying $n_h^{\tilde{k}_s}(x, a) > 0$, $N_h^{\tilde{k}_1}(x, a) \geq M$ and $N_h^{\tilde{k}_1 - 1}(x, a) < M$. Here, $\tilde{k}_s$s and $g$ are simplified notations for functions of $(x, a, h)$, and we use the simplified notations when there is no ambiguity and the stated meaning is only valid in proof of Equation (26). Then we have

$$\sum_{m,k,j} \beta_{t_h^{m,k,j}} \mathbb{I}[t_h^{m,k,j} \geq M] = O(1) \sum_{(x,a) \in \mathcal{S} \times \mathcal{A}} \sum_{s=1}^{g} n_h^{\tilde{k}_s}(x, a) \sqrt{\frac{H^3 \iota}{N_h^{\tilde{k}_s}(x, a)}}.$$

Firstly, we prove that

$$\sum_{(x,a) \in \mathcal{S} \times \mathcal{A}} \sum_{s=1}^{g} \sum_{j''=1}^{n_h^{\tilde{k}_s}(x,a)} \sqrt{\frac{H^3 \iota}{N_h^{\tilde{k}_s}(x, a) + j'' - 1}} = O(1) \sum_{(x,a) \in \mathcal{S} \times \mathcal{A}} \sqrt{H^3 \iota (N_h^{\tilde{k}_g}(x, a) + n_h^{\tilde{k}_g}(x, a) - 1)}$$

$$= O(1) \sum_{(x,a) \in \mathcal{S} \times \mathcal{A}} \sqrt{H^3 \iota N_h^{K+1}(x, a)}$$

$$\overset{(a)}{\leq} O(\sqrt{H^2 \iota \hat{T} S A})$$

where $(a)$ holds because of the concavity of $f(x) = \sqrt{H^3 \iota x}$ and the fact that $\sum_{(x,a) \in \mathcal{S} \times \mathcal{A}} N_h^{K+1}(x, a) \leq \hat{T}/H$.

Then we bound $\dfrac{1 \big/ \sqrt{N_h^{\tilde{k}_s}(x,a)}}{1 \big/ \sqrt{N_h^{\tilde{k}_s}(x,a)+d}}$. If we can find some numerical constant $C'' > 1$ such that

$$C'' \geq \max_{(j,d) \in \tilde{B}} \frac{1 \big/ \sqrt{N_h^{\tilde{k}_s}(x, a)}}{1 \big/ \sqrt{N_h^{\tilde{k}_s}(x, a) + d}}, \forall (x, a, h) \in \mathcal{S} \times \mathcal{A} \times [H],$$

in which $\tilde{B} = \{(s, d) \in \mathbb{N}^2 : 1 \leq s \leq g, 1 \leq d \leq n_h^{\tilde{k}_s}(x, a) - 1\}$, then we can have

$$\sum_{m,k,j} \beta_{t_h^{m,k,j}} \mathbb{I}[t_h^{m,k,j} \geq M] = O(1) \sum_{(x,a) \in \mathcal{S} \times \mathcal{A}} \sum_{s=1}^{g} n_h^{\tilde{k}_s}(x, a) \sqrt{\frac{H^3 \iota}{N_h^{\tilde{k}_s}(x, a)}}$$

$$\leq O(C'') \sum_{(x,a) \in \mathcal{S} \times \mathcal{A}} \sum_{s=1}^{g} \sum_{j''=1}^{n_h^{\tilde{k}_s}(x,a)} \sqrt{\frac{H^3 \iota}{N_h^{\tilde{k}_s}(x, a) + j'' - 1}}$$

$$= O(\sqrt{H^2 \iota \hat{T} S A}). \tag{30}$$

Next, we will prove that we can choose $C'' = \sqrt{2}$. We notice that

$$\max_{d \in [n_h^{\tilde{k}_s}(x,a)-1]} \frac{1 \big/ \sqrt{N_h^{\tilde{k}_s}(x, a)}}{1 \big/ \sqrt{N_h^{\tilde{k}_s}(x, a) + d}} = \sqrt{\frac{N_h^{\tilde{k}_s}(x, a) + n_h^{\tilde{k}_s}(x, a) - 1}{N_h^{\tilde{k}_s}(x, a)}}.$$

If $M \leq N_h^{\tilde{k}_s}(x, a) < i_0$, according to Equation (12), $\tilde{n}_j(x, a) \leq M$, which indicates that

$$\sqrt{\frac{N_h^{\tilde{k}_s}(x, a) + n_h^{\tilde{k}_s}(x, a) - 1}{N_h^{\tilde{k}_s}(x, a)}} \leq \sqrt{2}.$$

If $N_h^{\tilde{k}_j(x,a,h)}(x, a) \geq i_0$, according to Equation (12), $n_h^{\tilde{k}_s}(x, a) \leq \tilde{C} N_h^{\tilde{k}_s}(x, a)$

$$\sqrt{\frac{N_h^{\tilde{k}_s}(x, a) + n_h^{\tilde{k}_s}(x, a) - 1}{N_h^{\tilde{k}_s}(x, a)}} \leq \sqrt{1 + \tilde{C}} \leq \sqrt{2}.$$

So, we can choose $C'' = \sqrt{2}$.

Combining Equation (29) and Equation (30), we obtain Equation (26). $\qquad\square$

# D  PROOF OF THEOREM 4.2

*Proof of Theorem 4.2.* This theorem is proved under the synchronization assumption, i.e., $n^{m,k} = n^k, \forall m \in [M]$. We only need to prove that when $k \geq H^2(H+1)SAM$,

$$\left[\left(1 + \frac{1}{2H(H+1)M}\right)^{\lceil K/(HSA)\rceil - H(H+1)M}\right] H^2(H+1)M^2 \leq \hat{T}.$$

For each $k \in [K]$, there exists at least one $(x, m, h) \in \mathcal{S} \times [M] \times [H]$ with $a = \pi_h^k(x)$ such that equality in Equation (11) holds. Thus, there exist at least $K$ different tuples of $(x, a, h, m, k) \in \mathcal{S} \times \mathcal{A} \times [H] \times [M] \times [K]$ such that equality in Equation (11) holds. Define set $\mathcal{K}$ to have all the different $k$'s satisfying that there exists $m \in [M]$ such that the equality in Equation (11) holds. Then, by Pigeonhole principle, there must exist a triple $(x, a, h) \in \mathcal{S} \times \mathcal{A} \times [H]$ such that $|\mathcal{K}| \geq \lceil K/(HSA)\rceil$.

We order the elements of $\mathcal{K}$ as $0 < k_1 < k_2 \ldots < k_g \leq K$, where $g \geq \lceil K/(HSA)\rceil$. We also denote $N_s = N_h^{k_s+1}(x, a)$, $m_s$ as the first agent index such that equality in Equation (11) holds, and $n_s = n_h^{m_s,k_s}(x, a)$. Due to the synchronization assumption, we have

$$\hat{T} \geq HM \sum_{s=1}^{g} n_s. \tag{31}$$

When $N_s \geq H(H+1)M$, we have that

$$N_s \geq \sum_{s'=1}^{s} n_{s'}, n_{s+1} \geq \tilde{C}N_s/(2M)$$

due to Equation (11) and $\left\lfloor \frac{s'}{H(H+1)M}\right\rfloor \geq \frac{s'}{2H(H+1)M}, \forall s' \geq M(H+1)H$.

Thus, we have that $\sum_{s=1}^{H(H+1)M} n_s \geq H(H+1)M$ and

$$\sum_{s'=1}^{s+1} n_{s'} \geq \sum_{s'=1}^{s} n_{s'} + \tilde{C}N_s/(2M) \geq (1 + \tilde{C}/(2M)) \sum_{s'=1}^{s} n_{s'}, s \geq H(H+1)M.$$

Therefore,

$$\sum_{s'=1}^{g} n_{s'} \geq \left[\left(1 + \tilde{C}/(2M)\right)^{g-H(H+1)M}\right] H(H+1)M$$

$$\geq \left[\left(1 + \tilde{C}/(2M)\right)^{\lceil K/(HSA)\rceil - H(H+1)M}\right] H(H+1)M.$$

Combining with Equation (31), we have

$$\left[\left(1 + \frac{1}{2H(H+1)M}\right)^{\lceil K/(HSA)\rceil - H(H+1)M}\right] H^2(H+1)M^2 \leq \hat{T},$$

which directly leads to the conclusion. $\qquad\square$

# E  THE BERNSTEIN-TYPE ALGORITHM

## E.1  ALGORITHM DESIGN

The Bernstein-type algorithm differs from the Hoeffding-type algorithm Algorithms 1 and 2, in that it selects the upper confidence bound based on a variance estimator of $X_i$, akin to the approach used in the Bernstein-type algorithm in Jin et al. (2018). This is done to determine a probability upper bound of $|\sum_{i=1}^{t} \theta_t^i X_i|$. In this subsection, we first introduce the algorithm design.

To facilitate understanding, we introduce additional notations exclusive to Bernstein-type algorithms, supplementing the already provided notations for the Hoeffding-type algorithm.

$$\mu_h^{m,k}(x,a) = \frac{1}{n_h^{m,k}(x,a)} \sum_{j=1}^{n^{m,k}} \left[ V_{h+1}^k \left( x_{h+1}^{m,k,j} \right) \right]^2 \mathbb{I}[(x_h^{m,k,j}, a_h^{m,k,j}) = (x,a)].$$

$$\mu_h^k(x,a) = \frac{1}{N_h^{k+1}(x,a) - N_h^k(x,a)} \sum_{m=1}^M \mu_h^{m,k}(x,a) n_h^{m,k}(x,a).$$

Here, $\mu_h^{m,k}(x,a)$ is the sample mean of $\left[ V_{h+1}^k (x_{h+1}^{m,k,j}) \right]^2$ for all the visits of $(x,a,h)$ for the $m-$th agent during the $k-$th round and $\mu_h^k(x,a)$ corresponds to the mean for all the visits during the $k-$th round. We emphasize here that we adopt the general notation $n^{m,k}$ in the definition of $\mu_h^{m,k}$. We define $W_k(x,a,h)$ to denote the sample variance of all the visits before the $k-$th round, calculated using $V_{h+1}^{k^i}(x_{h+1}^{m^i,k^i,j^i})$, i.e.

$$W_k(x,a,h) = \frac{1}{N_h^k(x,a)} \sum_{i=1}^{N_h^k(x,a)} \left( V_{h+1}^{k^i}(x_{h+1}^{m^i,k^i,j^i}) - \frac{1}{N_h^k(x,a)} \sum_{i'=1}^{N_h^k(x,a)} V_{h+1}^{k^i}(x_{h+1}^{m^i,k^i,j^i}) \right)^2.$$

We can find that

$$W_k(x,a,h) = \frac{1}{N_h^k(x,a)} \sum_{k'=1}^{k-1} \mu_h^{k'}(x,a) n_h^{k'}(x,a) - \left[ \frac{1}{N_h^k(x,a)} \sum_{k'=1}^{k-1} v_{h+1}^{k'}(x,a) n_h^{k'}(x,a) \right]^2,$$

which means that this quantity can be calculated efficiently in practice in the following way. Define

$$W_{1,k}(x,a,h) = \sum_{k'=1}^{k-1} \mu_h^{k'}(x,a) n_h^{k'}(x,a), \quad W_{2,k}(x,a,h) = \sum_{k'=1}^{k-1} v_{h+1}^{k'}(x,a) n_h^{k'}(x,a), \tag{32}$$

we have that

$$W_{1,k+1}(x,a,h) = W_{1,k}(x,a,h) + \mu_h^k(x,a) n_h^k(x,a), \tag{33}$$

$$W_{2,k+1}(x,a,h) = W_{2,k}(x,a,h) + v_{h+1}^k(x,a) n_h^k(x,a) \tag{34}$$

and

$$W_{k+1}(x,a,h) = \frac{W_{1,k+1}(x,a,h)}{N_h^{k+1}(x,a)} - \left[ \frac{W_{2,k+1}(x,a,h)}{N_h^{k+1}(x,a)} \right]^2. \tag{35}$$

This indicates that the central server, by actively maintaining and updating the quantities $W_{1,k}$ and $W_{2,k}$ and systematically collecting $n_h^{m,k}$s, $\mu_h^{m,k}$s and $v_{h+1}^{m,k}$s, is able to compute $W_{k+1}$.

Next, we define

$$\beta_t(x,a,h) = c' \left( \min \left\{ \sqrt{\frac{H\iota}{t} (W_{k^t+1}(x,a,h) + H)} + \iota \frac{\sqrt{H^7 SA} + \sqrt{MSAH^6}}{t}, \sqrt{\frac{H^3 \iota}{t}} \right\} \right),$$

in which $c' > 0$ is a positive constant. Here, $W_{k^t+1}(x,a,h) = W^t(x,a,h)$ which is mentioned in Section 5. With this, the upper confidence bound $b_t(x,a,h)$ for a single visit is determined by

$$\beta_t(x,a,h) = 2 \sum_{i=1}^t \theta_t^i b_t(x,a,h),$$

which can be calculated as follows:

$$b_1(x,a,h) := \frac{\beta_1(x,a,h)}{2},$$

$$b_t(x,a,h) := \frac{\beta_t(x,a,h) - (1 - \alpha_t) \beta_{t-1}(x,a,h)}{2\alpha_t}.$$

When there is no ambiguity, we adopt the simplified notation $\tilde{b}_t = b_t(x, a, h)$ and $\tilde{\beta}_t = \beta_t(x, a, h)$. In the Bernstein-type algorithm, we let $\tilde{\beta} = \beta_{t^k}(x, a, h) - \alpha^c(t^{k-1} + 1, t^k)\beta_{t^{k-1}}(x, a, h)$ in replace of $\beta^k$ in Equation (3) and Equation (4). We know that $\tilde{\beta}_t \leq \beta_t$ when $c = c'$, indicating that the Bernstein-type algorithm operates with a smaller upper confidence bound.

Next, we will delve into certain components of the algorithm in round $k$. We remark that we discuss our algorithm based on the general situation where there is no necessity for zero latency and synchronization assumptions. In this general scenario, agent $m$ generates $n^{m,k}$ episodes in round $k$.

**Coordinated Exploration for Agents.** At the beginning of round $k$, the server decides a deterministic policy $\pi^k = \{\pi_h^k\}_{h=1}^H$, and then broadcasts it along with $\{N_h^k(x, \pi_h^k(x))\}_{x,h}$ and $\{V_h^k(x)\}_{x,h}$ to all of the agents. When $k = 1$, $N_h^1(x, a) = 0$, $Q_h^1(x, a) = V_h^1(x) = H, \forall(x, a, h) \in \mathcal{S} \times \mathcal{A} \times [H]$ and $\pi^1$ is an arbitrary deterministic policy.

Once receiving such information, the agents will execute policy $\pi^k$ and start collecting trajectories.

**Event-Triggered Termination of Exploration.** During exploration, every agent $m$ will monitor $n_h^{m,k}(x, a)$, i.e., the total number of visits for each $(x, a, h)$ triple within the current round. For any agent $m$, at the end of each episode, it sequentially conducts two procedures. First, if any $(x, a, h)$ has been visited by $\max\left\{1, \lfloor \frac{\tilde{C}}{M} N_h^k(x, a) \rfloor \right\}$ times by agent $m$, it will abort its own exploration and send an abortion signal to the server and other clients. Second, it checks whether it has received an abortion signal. If so, it will abort its exploration. We remark that Equation (2) still holds, and for any $k \in [K]$, there exists a tuple $(x, a, h)$ such that the equality is met.

**Local Updating of the Estimated Expected Return.** Each agent updates the local estimate of the expected return $v_{h+1}^{m,k}(x, a)$ at the end of round $k$. Next, each agent $m$ sends $\{r_h(x, \pi_h^k(x))\}_{x,h}, \{n_h^{m,k}(x, \pi_h^k(x))\}_{x,h}, \{v_{h+1}^{m,k}(x, \pi_h^k(x))\}_{x,h}$ and $\{\mu_h^{m,k}(x, \pi_h^k(x))\}_{x,h}$ to the central server for aggregation.

**Server-side Information Aggregation.** After receiving the information sent by the agents, for each $(x, a, h)$ tuple visited by the agents, the server first calculates $W_{1,k+1}(x, a, h)$, $W_{2,k+1}(x, a, h)$ and $W_{k+1}(x, a, h)$ based on Equation (32), Equation (33), Equation (34) and Equation (35) for each pair $(x, h)$ with $a = \pi_h^k(x)$. Then it sets $t^{k-1} = N_h^k(x, a)$, $t^k = N_h^{k+1}(x, a)$, $\alpha_{agg} = 1 - \alpha^c(t^{k-1} + 1, t^k)$ and $\tilde{\beta} = \beta_{t^k}(x, a, h) - \alpha^c(t^{k-1} + 1, t^k)\beta_{t^{k-1}}(x, a, h)$, and updates the global estimate of the value functions according to one of the following two cases.

- **Case 1:** $N_h^k(x, a) < i_0$. Due to Equation (2), this case implies that each client can visit each $(x, a)$ pair at step $h$ at most once. Then, we denote $1 \leq m_1 < m_2 \ldots < m_{t^k - t^{k-1}} \leq M$ as the agent indices with $n_h^{m,k}(x, a) > 0$. The server then updates the global estimate of action values as follows:

$$Q_h^{k+1}(x, a) = (1 - \alpha_{agg})Q_h^k(x, a) + \alpha_{agg}r_h(x, a) + \sum_{t=1}^{t^k - t^{k-1}} \theta_{t^k}^{t^{k-1}+t} v_{h+1}^{m_t,k}(x, a) + \tilde{\beta}/2. \quad (36)$$

- **Case 2:** $N_h^k(x, a) \geq i_0$. In this case, the central server calculates $v_{h+1}^k(x, a)$ as and updates the $Q$-estimate as

$$Q_h^{k+1}(x, a) = (1 - \alpha_{agg})Q_h^k(x, a) + \alpha_{agg}\left(r_h(x, a) + v_{h+1}^k(x, a)\right) + \tilde{\beta}/2. \quad (37)$$

After finishing updating the estimated $Q$ function, the central server updates the estimated value function and the policy based on Equations (5) and (6). The algorithm then proceeds to round $k + 1$.

Algorithms 3 and 4 formally present the Bernstein-type design. Inputs $K_0$ and $T_0$ in Algorithms 3 are termination conditions, where $K_0$ limits the total number of rounds and $T_0$ limits the total number of samples generated by all the agents before the last round.

We provide some remarks. First, the coordinated exploration for agents is designed based on the general situation where $n^{m,k}$ might be different across different agents, and clients share $\mu_h^{m,k}$s in addition to the information shared during the coordinated exploration in the Hoeffding-type Algorithm 2. Second, the information aggregation at the central server differs from that in the Hoeffding-type Algorithm 1, in terms of specifying $\tilde{\beta}$ to set the upper confidence bound and in maintaining $W_{1,k}, W_{2,k}$ and $W_k$.

---

**Algorithm 3** FedQ-Bernstein (Central Server)

1: **Input:** $T_0, K_0 \in \mathbb{N}_+$.
2: **Initialization:** $k = 1, N_h^1(x, a) = W_{1,k}(x, a, h) = W_{2,k}(x, a, h) = 0, Q_h^1(x, a) = V_h^1(x) = H, \forall (x, a, h) \in \mathcal{S} \times \mathcal{A} \times [H]$ and $\pi^1 = \left\{\pi_h^1 : \mathcal{S} \to \mathcal{A}\right\}_{h \in [H]}$ is an arbitrary deterministic policy.

3: **while** $H \sum_{k'=1}^{k-1} M n^{k'} < T_0 \ \& \ k \leq K_0$ **do**
4:     Broadcast $\pi^k$, $\{N_h^k(x, \pi_h^k(x))\}_{x,h}$ and $\{V_h^k(x)\}_{x,h}$ to all clients.
5:     Wait until receiving an abortion signal and send the signal to all agents.
6:     Receive     $\{r_h(x, \pi_h^k(x))\}_{x,h}, \{n_h^{m,k}(x, \pi_h^k(x))\}_{x,h,m}$,      $\{v_{h+1}^{m,k}(x, \pi_h^k(x))\}_{x,h,m}$     and $\{\mu_h^{m,k}(x, \pi_h^k(x))\}_{x,h,m}$ from clients.
7:     Calculate $N_h^{k+1}(x, a), n_h^k(x, a), v_{h+1}^k(x, a), \forall (x, h) \in \mathcal{S} \times [H]$ with $a = \pi_h^k(x)$.
8:     Calculate $W_k(x, a, h), W_{k+1}(x, a, h), W_{1,k+1}(x, a, h), W_{2,k+1}(x, a, h), \forall (x, h) \in \mathcal{S} \times [H]$ with $a = \pi_h^k(x)$ based on Equation (32), Equation (33), Equation (34) and Equation (35).
9:     **for** $(x, a, h) \in \mathcal{S} \times \mathcal{A} \times [H]$ **do**
10:         **if** $a \neq \pi_h^k(x)$ or $n_h^k(x, a) = 0$ **then**
11:             $Q_h^{k+1}(x, a) \leftarrow Q_h^k(x, a)$.
12:         **else if** $N_h^k(x, a) < i_0$ **then**
13:             Update $Q_h^{k+1}(x, a)$ according to Equation (36).
14:         **else**
15:             Update $Q_h^{k+1}(x, a)$ according to Equation (37).
16:         **end if**
17:     **end for**
18:     Update $V_h^{k+1}$ and $\pi^{k+1}$ according to Equation (5) and Equation (6).
19:     $k \leftarrow k + 1$.
20: **end while**

---

**Algorithm 4** FedQ-Bernstein (Agent $m$ in round $k$)

1: $n_h^m(x, a) = v_{h+1}^m(x, a) = r_h(x, a) = \mu_h^m(x, a) = 0, \forall (x, a, h) \in \mathcal{S} \times \mathcal{A} \times [H]$.
2: Receive $\pi^k$, $\{N_h^k(x, \pi_h^k(x))\}_{x,h}$ and $\{V_h^k(x)\}_{x,h}$ from the central server.
3: **while** no abortion signal from the central server **do**
4:     **while** $n_h^m(x_h, a_h) < \max\left\{1, \lfloor \frac{\tilde{C}}{M} N_h^k(x_h, a_h)\rfloor\right\}, \forall (x, a, h) \in \mathcal{S} \times \mathcal{A} \times [H]$ **do**
5:         Collect a new trajectory $\{(x_h, a_h, r_h)\}_{h=1}^H$ with $a_h = \pi_h^k(x_h)$.
6:         $n_h^m(x_h, a_h) \leftarrow n_h^m(x_h, a_h) + 1, \ v_{h+1}^m(x_h, a_h) \leftarrow v_{h+1}^m(x_h, a_h) + V_{h+1}^k(x_{h+1})$,
        $\mu_h^m(x_h, a_h) \leftarrow \mu_h^m(x_h, a_h) + \left[V_{h+1}^k(x_{h+1})\right]^2$, and $r_h(x_h, a_h) \leftarrow r_h, \forall h \in [H]$.
7:     **end while**
8:     Send an abortion signal to the central server.
9: **end while**
10: $n_h^{m,k}(x, a) \leftarrow n_h^m(x, a), v_{h+1}^{m,k}(x, a) \leftarrow v_{h+1}^m(x, a)/n_h^m(x, a)$ and $\mu_h^{m,k}(x, a) \leftarrow \mu_h^m(x, a)/n_h^m(x, a), \forall (x, h) \in \mathcal{S} \times [H]$ with $a = \pi_h^k(x)$.
11: Send $\{r_h(x, \pi_h^k(x))\}_{x,h}, \{n_h^{m,k}(x, \pi_h^k(x))\}_{x,h}, \{\mu_h^{m,k}(x, \pi_h^k(x))\}_{x,h}$ and $\{v_{h+1}^{m,k}(x, \pi_h^k(x))\}_{x,h}$ to the central server.

---

### E.2 PROOF OF THEOREM 5.1

In this subsection, we provide proof for Theorem 5.1 which provides the regret of Algorithms 3 and 4.

### E.2.1 BOUNDS ON $Q_h^k - Q_h^\star$

We first try to provide a Lemma that has stronger results than Lemma C.1.

**Lemma E.1.** *Using Algorithms 3 and 4, there exists a positive constant $c' > 0$ such that, for any $p \in (0, 1)$, the following relationship holds simultaneously for all $(x, a, h, K') \in \mathcal{S} \times \mathcal{A} \times [H] \times [K]$*

with probability at least $1 - p$.

$$0 \le Q_h^{K'}(x, a) - Q_h^\star(x, a) \le \theta_t^0 H + \sum_{i=1}^t \tilde{\theta}_t^i (V_{h+1}^{k^i} - V_{h+1}^\star)(x_{h+1}^{m^i, k^i, j^i}) + \beta_t(x, a, h), \qquad (38)$$

in which $t = N_h^{K'}(x, a)$.

The remaining content of Appendix E.2.1 is dedicated to proving this Lemma. First, we can easily find that Lemma C.2 still holds with $b_t, \beta_t$ replaced by $\tilde{b}_t, \tilde{\beta}_t$, and Lemma C.3 still holds. Next, due to Equation (16) and Equation (15) with $b_t, \beta_t$ replaced, we can easily obtain a similar one-sided result summarized in the following Lemma.

**Lemma E.2.** *Using the Bernstein-type algorithm, there exists a positive constant $c_0' > 0$ such that, for any $p \in (0, 1)$, the following relationship holds simultaneously for all $(x, a, h, K') \in \mathcal{S} \times \mathcal{A} \times [H] \times [K]$ with probability at least $1 - p$.*

$$Q_h^{K'}(x, a) - Q_h^\star(x, a) \le \theta_t^0 H + \sum_{i=1}^t \tilde{\theta}_t^i (V_{h+1}^{k^i} - V_{h+1}^\star)(x_{h+1}^{m^i, k^i, j^i}) + c_0' \sqrt{H^3 \iota / t}, \qquad (39)$$

in which $t = N_h^{K'}(x, a)$.

*Proof.* This relationship can be directly obtained from Equation (16) and Equation (15) with $b_t, \beta_t$ replaced. $\qquad \square$

With this, we can introduce the following technical Lemma.

**Lemma E.3.** *Suppose that Equation (39) holds. For any given $K' \in \mathbb{N}$, denote $\sum_{m,k,j}^{K'} = \sum_{k=1}^{K'} \sum_{m=1}^M \sum_{j=1}^{n_h^{m,k}}$ and $w = vec(\{w_{mkj}\})$ with $m \in [M], k \in [K'], j \in [n^{m,k}]$ be a non-negative vector. Then there exists a numerical constant $c_1' > 0$ such that, for all $(h, K') \in [H] \times [K]$,*

$$\sum_{m,k,j}^{K'} w_{mkj} \left( V_h^k(x_h^{m,k,j}) - V_h^\star(x_h^{m,k,j}) \right)$$

$$\le c_1' \left( \|w\|_\infty MSA \sqrt{H^5 \iota} + \sqrt{SA \|w\|_\infty \|w\|_1 H^5 \iota} + H^4 SA(M-1) \|w\|_\infty \right). \qquad (40)$$

*Proof.* We denote
$$\tilde{V}_h^{m,k,j} = V_h^k(x_h^{m,k,j}) - V_h^\star(x_h^{m,k,j}).$$

Noticing that $Q_h^k \left( x_h^{m,k,j}, a_h^{m,k,j} \right) \ge V_h^k \left( x_h^{m,k,j} \right)$ and $Q_h^\star \left( x_h^{m,k,j}, a_h^{m,k,j} \right) = \max_{a \in \mathcal{A}} Q_h^\star \left( x_h^{m,k,j} \right) \le V_h^\star \left( x_h^{m,k,j} \right)$, letting $k = K'$ and $(x, a) = (x_h^{m,k,j}, a_h^{m,k,j})$, we have that

$$\tilde{V}_h^{m,k,j} \le \theta_{t_h^{m,k,j}}^0 H + \sum_{i=1}^{t_h^{m,k,j}} \tilde{\theta}_{t_h^{m,k,j}}^i \tilde{V}_{h+1}^{(m,k,j)_{i,h}^{m,k,j}} + c_0' \sqrt{H^3 \iota / t_h^{m,k,j}}.$$

Taking the summation with regard to $k$ from 1 to $K'$ and noticing that $\theta_{t_h^{m,k,j}}^0 = \mathbb{I}[t_h^{m,k,j} > 0]$, we have

$$\sum_{m,k,j}^{K'} w_{mkj} \tilde{V}_h^{m,k,j} \le H \sum_{m,k,j}^{K'} w_{mkj} \mathbb{I}[t_h^{m,k,j} = 0] + \sum_{m,k,j} w_{mkj} \sum_{i=1}^{t_h^{m,k,j}} \theta_{t_h^{m,k,j}}^i \tilde{V}_{h+1}^{(m,k,j)_{i,h}^{m,k,j}}$$

$$+ \sum_{m,k,j}^{K'} w_{mkj} \Omega \left( \sqrt{\frac{H^3 \iota}{t_h^{m,k,j}}} \right).$$

Next, we find upper bounds with regard to the three terms.

**Step 1:** finding an upper bound for $H \sum_{m,k,j}^{K'} w_{mkj} \mathbb{I}[t_h^{m,k,j} = 0]$. Noticing that $w_{mkj} \le \|w\|_\infty$, using the same way as Proof of Equation (24) in Appendix C.4, we can find that

$$H \sum_{m,k,j} w_{mkj} \mathbb{I}[t_h^{m,k,j} = 0] \le MHSA \|w\|_\infty.$$

**Step 2:** finding an upper bound for $\sum_{m,k,j}^{K'} w_{mkj} \sum_{i=1}^{t_h^{m,k,j}} \tilde{\theta}_{t_h^{m,k,j}}^i \tilde{V}_{h+1}^{(m,k,j)_{i,h}^{m,k,j}}$. Similar to Proof of Equation (25) in Appendix C.4, with $i_1 = (M-1)H(H+1)$, we still split it into two parts as follows:

$$\sum_{m,k,j}^{K'} w_{mkj} \sum_{i=1}^{t_h^{m,k,j}} \tilde{\theta}_{t_h^{m,k,j}}^i \tilde{V}_{h+1}^{(m,k,j)_{i,h}^{m,k,j}} = \sum_{m,k,j}^{K'} w_{mkj} \mathbb{I}[t_h^{m,k,j} \le i_1] \sum_{i=1}^{t_h^{m,k,j}} \tilde{\theta}_{t_h^{m,k,j}}^i \tilde{V}_{h+1}^{(m,k,j)_{i,h}^{m,k,j}}$$
$$+ \sum_{m,k,j}^{K'} w_{mkj} \mathbb{I}[t_h^{m,k,j} > i_1] \sum_{i=1}^{t_h^{m,k,j}} \tilde{\theta}_{t_h^{m,k,j}}^i \tilde{V}_{h+1}^{(m,k,j)_{i,h}^{m,k,j}}.$$

For the first part, applying $w_{mkj} \le \|w\|_\infty$, using the same way as Proof of Equation (25) in Appendix C.4, we can find that

$$\sum_{m,k,j}^{K'} w_{mkj} \mathbb{I}[t_h^{m,k,j} \le i_1] \sum_{i=1}^{t_h^{m,k,j}} \tilde{\theta}_{t_h^{m,k,j}}^i \tilde{V}_{h+1}^{(m,k,j)_{i,h}^{m,k,j}} \le O(H^3 SA(M-1)\|w\|_\infty).$$

For the second part, we regroup the summations in a different way. For every $(m',k',j')$, the term $\tilde{V}_{h+1}^{m',k',j'}$ appears in the term $w_{mkj} \mathbb{I}[t_h^{m,k,j} > i_1] \sum_{i=1}^{t_h^{m,k,j}} \theta_{t_h^{m,k,j}}^i \tilde{V}_{h+1}^{(m,k,j)_{i,h}^{m,k,j}}$ for $(k,m,j)$ if and only if $K' \ge k > k'$ and $(x_h^{m,k,j}, a_h^{m,k,j}) = (x_h^{m',k',j'}, a_h^{m',k',j'})$. So, for each $(m',k',j')$, we denote $(x,a) = (x_h^{m',k',j'}, a_h^{m',k',j'})$. We consider all the later round indices $k' = k_0 < k_1 < k_2 < ... < k_g \le K'$ that satisfy $n_s = n_h^{k_s}(x,a) > 0, s \in \mathbb{N}$. Here, $k_s$s and $g$ are simplified notations for functions of $(m',k',j',h)$, and we use the simplified notations when there is no ambiguity and the stated meaning is only valid in proof of step 2. So, the summation of coefficients related to $(m',k',j')$ equals to

$$\tilde{w}_{m'k'j'} = \sum_{s=1}^{g} \left( \sum_{i=N_h^{k_s}(x,a)+1}^{N_h^{k_s}(x,a)+n_s} w_{(mkj)_{i,h}^{m,k,j}} \right) \mathbb{I}[N_h^{k_s}(x,a) > i_1] \tilde{\theta}_{N_h^{k_s}(x,a)}^{i'},$$

in which $i'$ is the global visiting number for $(x,a,h)$ at $(m',k',j')$, which means that $(m',k',j') = m_h(i';x,a), k_h(i';x,a), j_h(i';x,a)$. This means that

$$\sum_{m,k,j}^{K'} w_{mkj} \mathbb{I}[t_h^{m,k,j} > i_1] \sum_{i=1}^{t_h^{m,k,j}} \tilde{\theta}_{t_h^{m,k,j}}^i \tilde{V}_{h+1}^{(m,k,j)_{i,h}^{m,k,j}} = \sum_{m',k',j'}^{K'} \tilde{w}_{m'k'j'} \tilde{V}_{h+1}^{m',k',j'}.$$

Denote $\tilde{w} = \mathrm{vec}(\{\tilde{w}_{m'k'j'}\})$, we have that

$$\|\tilde{w}\|_1 = \sum_{m,k,j}^{K'} w_{mkj} \mathbb{I}[t_h^{m,k,j} > i_1] \sum_{i=1}^{t_h^{m,k,j}} \tilde{\theta}_{t_h^{m,k,j}}^i \le \sum_{m,k,j}^{K'} w_{mkj} \sum_{i=1}^{t_h^{m,k,j}} \tilde{\theta}_{t_h^{m,k,j}}^i \le \|w\|_1$$

due to (c) in Lemma B.3. We can also find that

$$\tilde{w}_{m'k'j'} \le \sum_{s=1}^{g} n_s \|w\|_\infty \mathbb{I}[N_h^{k_s}(x,a) > i_1] \tilde{\theta}_{N_h^{k_s}(x,a)}^{i'} \le \exp(3/H)\|w\|_\infty,$$

where the proof of the last inequality is the same as the Proof of Equation (25). Combining the two parts, we have that

$$\sum_{m,k,j}^{K'} w_{mkj} \sum_{i=1}^{t_h^{m,k,j}} \tilde{\theta}_{t_h^{m,k,j}}^i \tilde{V}_{h+1}^{(m,k,j)_{i,h}^{m,k,j}} \le \sum_{m',k',j'}^{K'} \tilde{w}_{m'k'j'} \tilde{V}_{h+1}^{m',k',j'} + O\left(H^3 SA(M-1)\|w\|_\infty\right).$$

**Step 3:** finding an upper bound for $\sum_{m,k,j}^{K'} w_{mkj}\Omega\left(\sqrt{\frac{H^3\iota}{t_h^{m,k,j}}}\right)$. We split it into two parts as follows.

$$\sum_{m,k,j}^{K'} w_{mkj}\Omega\left(\sqrt{\frac{H^3\iota}{t_h^{m,k,j}}}\right) = \sum_{m,k,j}^{K'} w_{mkj}\mathbb{I}[0 < t_h^{m,k,j} \leq M - 1]\Omega\left(\sqrt{\frac{H^3\iota}{t_h^{m,k,j}}}\right)$$
$$+ \sum_{m,k,j}^{K'} w_{mkj}\mathbb{I}[t_h^{m,k,j} \geq M]\Omega\left(\sqrt{\frac{H^3\iota}{t_h^{m,k,j}}}\right).$$

For the first part, applying that $w_{mkj} \leq \|w\|_\infty$, similar to Proof of Equation (26) in Appendix C.4, we have that

$$\sum_{m,k,j}^{K'} w_{mkj}\mathbb{I}[0 < t_h^{m,k,j} \leq M - 1]\Omega\left(\sqrt{\frac{H^3\iota}{t_h^{m,k,j}}}\right) = \Omega\left(\|w\|_\infty SA(M - 1)\sqrt{H^3\iota}\right).$$

For the second part, we denote $w_k'(x, a, h) = \sum_{m=1}^M \sum_{j=1}^{n^{m,k}} w_{mkj}\mathbb{I}[(x_h^{m,k,j}, a_h^{m,k,j}) = (x, a)]$. We also introduce the following notation. For every pair $(x, a, h) \in \mathcal{S} \times \mathcal{A} \times [H]$, we consider all the rounds indexed as $0 < \tilde{k}_1 < \tilde{k}_2 < ... < \tilde{k}_g \leq K$ satisfying $n_h^{\tilde{k}_s}(x, a) > 0$, $N_h^{\tilde{k}_1}(x, a) \geq M$ and $N_h^{\tilde{k}_1-1}(x, a) < M$. Here, $\tilde{k}_s$s and $g$ are simplified notations for functions of $(x, a, h)$, and we use the simplified notations when there is no ambiguity and the stated meaning is only valid in proof of step 3. Then we have

$$\sum_{m,k,j}^{K'} w_{mkj}\mathbb{I}[t_h^{m,k,j} \geq M]\Omega\left(\sqrt{\frac{H^3\iota}{t_h^{m,k,j}}}\right) = \Omega(1) \sum_{(x,a)\in\mathcal{S}\times\mathcal{A}} \sum_{s=1}^g w_{\tilde{k}_s}'(x, a, h)\sqrt{\frac{H^3\iota}{N_h^{\tilde{k}_s}(x, a)}}.$$

We also define that

$$w'(i, x, a, h) = w_{\tilde{k}_s}'(x, a, h)/n_h^{\tilde{k}_s}(x, a), \forall i \in \mathbb{N}_+, j \in [N_h^{\tilde{k}_s}(x, a), N_h^{\tilde{k}_s}(x, a) + n_h^{\tilde{k}_s}(x, a) - 1],$$

which indicates that

$$w'(j, x, a, h) \leq \|w\|_\infty.$$

Similar to Proof of Equation (26), we have that

$$\sqrt{2} \geq \max_{(j,d)\in\tilde{B}} \frac{1/\sqrt{N_h^{\tilde{k}_s}(x, a)}}{1/\sqrt{N_h^{\tilde{k}_s}(x, a) + d}}, \forall (x, a, h) \in \mathcal{S} \times \mathcal{A} \times [H],$$

in which $\tilde{B} = \{(s, d) \in \mathbb{N}^2 : 1 \leq s \leq g, 1 \leq d \leq n_h^{\tilde{k}_s}(x, a) - 1\}$. So we have

$$\sum_{(x,a)\in\mathcal{S}\times\mathcal{A}} \sum_{s=1}^g w_{\tilde{k}_s}'(x, a, h)\sqrt{\frac{H^3\iota}{N_h^{\tilde{k}_s}(x, a)}} = O(1) \sum_{(x,a)\in\mathcal{S}\times\mathcal{A}} \sum_{i=N_h^{\tilde{k}_1}(x,a)}^{N_h^{\tilde{k}_g+1}(x,a)-1} w'(i, x, a, h)\sqrt{H^3\iota/i}$$

with

$$\sum_{(x,a)\in\mathcal{S}\times\mathcal{A}} \sum_{i=N_h^{\tilde{k}_1}(x,a)}^{N_h^{\tilde{k}_g+1}(x,a)-1} w'(i, x, a, h) \leq \|w\|_1.$$

Denote $w''(x, a, h) = \|w\|_\infty\left[\sum_{i=N_h^{\tilde{k}_1}(x,a)}^{N_h^{\tilde{k}_g+1}(x,a)-1} w'(i, x, a, h)/\|w\|_\infty\right]$, which indicates that

$$w''(x, a, h) \leq \sum_{i=N_h^{\tilde{k}_1}(x,a)}^{N_h^{\tilde{k}_g+1}(x,a)-1} w'(i, x, a, h) + \|w\|_\infty$$

so that

$$\sum_{(x,a)\in\mathcal{S}\times\mathcal{A}} w''(x,a,h) \le \|w\|_1 + SA\|w\|_\infty.$$

Then by letting the mass related to $\{w'(i,x,a,h)\}_i$ concentrate at large values for $\{\sqrt{H^3\iota/i}\}_i$ as much as possible, we have

$$\sum_{i=N_h^{\tilde{k}_1}(x,a)}^{N_h^{\tilde{k}_g+1}(x,a)-1} w'(i,x,a,h)\sqrt{H^3\iota/i} \le \|w\|_\infty \sum_{i=N_h^{k_1}(x,a)}^{N_h^{k_1}(x,a)+w''(x,a,h)/\|w\|_\infty-1} \sqrt{H^3\iota/i}$$

$$= O\left(\sqrt{H^3\iota w''(x,a,h)\|w\|_\infty}\right).$$

By the concavity of $f(x) = \sqrt{H^3\iota x}$, we have

$$\sum_{m,k,j}^{K'} w_{mkj}\mathbb{I}[t_h^{m,k,j} \ge M]\Omega\left(\sqrt{\frac{H^3\iota}{t_h^{m,k,j}}}\right) \le \sum_{(x,a)\in\mathcal{S}\times\mathcal{A}} O\left(\sqrt{H^3\iota w''(x,a,h)\|w\|_\infty}\right)$$

$$\le \left(\sqrt{H^3SA\iota(\|w\|_1 + SA\|w\|_\infty)\|w\|_\infty}\right)$$

$$= \left(\sqrt{H^3SA\iota\|w\|_1\|w\|_\infty} + SA\|w\|_\infty\sqrt{H^3\iota}\right).$$

To conclude, for step 3, we have that

$$\sum_{m,k,j}^{K'} w_{mkj}\Omega\left(\sqrt{\frac{H^3\iota}{t_h^{m,k,j}}}\right) \le O\left(\sqrt{H^3SA\iota\|w\|_1\|w\|_\infty} + MSA\|w\|_\infty\sqrt{H^3\iota}\right).$$

Combining the results for the three different steps, we have

$$\sum_{m,k,j} w_{mkj}\tilde{V}_h^{m,k,j} \le \sum_{m,k,j} \tilde{w}_{mkj}\tilde{V}_{h+1}^{m,k,j} + O\left(\sqrt{H^3SA\iota\|w\|_1\|w\|_\infty}\right.$$

$$\left. + MSA\|w\|_\infty\sqrt{H^3\iota} + H^3SA(M-1)\|w\|_\infty\right),$$

with $\|w\|_1 \le \exp(3/H)\|\tilde{w}\|_1$ and $\|w\|_\infty \le \|\tilde{w}\|_\infty$. So, by recursions with regard to $h, h+1\ldots H$, we can get the result. $\qquad\square$

Next, we will establish relationships between $W_k(x,a,h)$ and $\left[\mathbb{V}_h V_{h+1}^\star\right](x,a)$, in which $\left[\mathbb{V}_h V_{h+1}^\star\right](x,a)$ is a variance operator define below. We also need these definitions for any $(x,a,h,K') \in \mathcal{S}\times\mathcal{A}\times[H]\times[K+1]$ with $t = N_h^{K'}(x,a)$.

$$\left[\mathbb{P}_h V_{h+1}^\star\right](x,a) = \mathbb{E}[V_{h+1}^\star(x_{h+1})|(x_h,a_h)=(x,a)].$$

$$\left[\mathbb{V}_h V_{h+1}^\star\right](x,a) = \mathbb{E}_{x'\sim\mathbb{P}_h(\cdot|x,a)}\left[V_{h+1}^\star(x') - \left[\mathbb{P}_h V_{h+1}^\star\right](x,a)\right]^2 =: P_1$$

Here, $P_1$ depends on $(x,a,h)$ and we will use the simplified notation when there is no ambiguity.

$$\frac{1}{t}\sum_{\tilde{i}=1}^{t}\left[V_{h+1}^\star\left(x_{h+1}^{(m,k,j)_h(i;x,a)}\right) - \left[\mathbb{P}_h V_{h+1}^\star\right](x,a)\right]^2 =: P_2.$$

$$\frac{1}{t}\sum_{\tilde{i}=1}^{t}\left[V_{h+1}^\star\left(x_{h+1}^{(m,k,j)_h(i;x,a)}\right) - \frac{1}{t}\sum_{i'=1}^{t}V_{h+1}^\star\left(x_{h+1}^{(m,k,j)_h(i';x,a)}\right)\right]^2 =: P_3$$

$$W_{K'}(x,a,h) = \frac{1}{t}\sum_{i=1}^{t}\left[V_{h+1}^{k_h(i;x,a)}\left(x_{h+1}^{(m,k,j)_h(i;x,a)}\right) - \frac{1}{t}\sum_{i'=1}^{t}V_{h+1}^{k_h(i';x,a)}\left(x_{h+1}^{(m,k,j)_h(i';x,a)}\right)\right]^2 =: P_4.$$

Here, $P_2, P_3, P_4$ depend on $(x,a,h,k)$ and we use the simplified notations when there is no ambiguity. The following Lemmas establish the closeness of these quantities to illustrate the closeness between $W_{K'}(x,a,h)$ and $\left[\mathbb{V}_h V_{h+1}^\star\right](x,a)$.

**Lemma E.4.** *For any $p \in (0,1)$ with probability at least $1 - p$, the following holds simultaneously for all $(x, a, h, K') \in \mathcal{S} \times \mathcal{A} \times [H] \times [K+1]$ with $t = N_h^{K'}(x, a)$.*

$$|P_1 - P_2| \leq O\left(H^2 \sqrt{\iota/t}\right).$$

*Proof.* We have that $\left\{\left[V_{h+1}^\star\left(x_{h+1}^{(m,k,j)_h(i;x,a)}\right) - [\mathbb{P}_h V_{h+1}^\star](x,a)\right]^2 - P_1\right\}_{i=1}^\infty$ is a martingale difference bounded by $O(H^2)$, and the random variable $t \leq T_0/H(1 + \tilde{C})$. By Azuma-Hoeffding Inequality, for any given $(x, a, h) \in \mathcal{S} \times \mathcal{A} \times [H]$ and a given $t' \in \mathbb{N}_+$, for any $p \in (0,1)$, with probability $1 - p$,

$$\frac{1}{t'}\left|\sum_{i=1}^{t'}\left(\left[V_{h+1}^\star\left(x_{h+1}^{(m,k,j)_h(i;x,a)}\right) - [\mathbb{P}_h V_{h+1}^\star](x,a)\right]^2 - P_1\right)\right| \leq O\left(H^2 \sqrt{\frac{1}{t'}\log\frac{2}{p}}\right).$$

By considering all the possible combinations $(x, a, h, t') \in \mathcal{S} \times \mathcal{A} \times [H] \times \left[[T_0(1 + \tilde{C})/H + M]\right]$, with a union bound and the realization of $t = t'$, we can claim the conclusion. $\square$

**Lemma E.5.** *For any $p \in (0,1)$ with least $1 - p$ probability, the following holds simultaneously for all $(x, a, h, K') \in \mathcal{S} \times \mathcal{A} \times [H] \times [K+1]$ with $t = N_h^{K'}(x, a)$:*

$$|P_2 - P_3| \leq O\left(H^2 \sqrt{\iota/t}\right).$$

*Proof.* We can find that

$$|P_2 - P_3| \leq O\left(H\left|\frac{1}{t}\sum_{i'=1}^{t} V_{h+1}^\star\left(x_{h+1}^{(m,k,j)_h(i';x,a)}\right) - [\mathbb{P}_h V_{h+1}^\star](x,a)\right|\right).$$

Knowing that $\left\{V_{h+1}^\star\left(x_{h+1}^{(m,k,j)_h(i';x,a)}\right) - [\mathbb{P}_h V_{h+1}^\star](x,a)\right\}_{i'=1}^\infty$ is a martingale difference bounded by $O(H)$, using the same procedure as proof for Lemma E.4, we can claim the result. $\square$

For $|P_3 - P_4|$, similar to the proof of Lemma C.3 in Jin et al. (2018), we have

$$|P_3 - P_4| \leq O\left(\frac{H}{t}\sum_{i=1}^{t}\left|V_{h+1}^{k_h(i;x,a)}\left(x_{h+1}^{(m,k,j)_h(i;x,a)}\right) - V_{h+1}^\star\left(x_{h+1}^{(m,k,j)_h(i;x,a)}\right)\right|\right).$$

We mark an event Equation (41) here, which means that the difference is always non-negative.

$$\text{Event}(K') = \left\{\sum_{i=1}^{t}\left|V_{h+1}^{k^i}\left(x_{h+1}^{m^i,k^i,j^i}\right) - V_{h+1}^\star\left(x_{h+1}^{m^i,k^i,j^i}\right)\right|\right.$$

$$= \left.\sum_{i=1}^{t}\left(V_{h+1}^{k^i}\left(x_{h+1}^{m^i k^i,j^i}\right) - V_{h+1}^\star\left(x_{h+1}^{m^i,k^i,j^i}\right)\right), \forall (x, a, h) \in \mathcal{S} \times \mathcal{A} \times [H]\right\}.$$
$$\tag{41}$$

We do not need a new statistical lemma to prove that it holds with high probability. It will be shown to hold automatically based on some other statistical events that hold with high probability later. Under this event, we need to find an upper bound for

$$\frac{1}{t}\sum_{i=1}^{t}\left(V_{h+1}^{k_h(i;x,a)}\left(x_{h+1}^{(m,k,j)_h(i;x,a)}\right) - V_{h+1}^\star\left(x_{h+1}^{(m,k,j)_h(i;x,a)}\right)\right).$$

Under the event of Equation (39), based on Lemma E.3, letting $w_{mkj} = \frac{1}{t}\mathbb{I}[(x_h^{m,k,j}, a_h^{m,k,j}) = (x, a)]$, we have that

$$\frac{1}{t}\sum_{i=1}^{t}\left(V_{h+1}^{k_h(i;x,a)}\left(x_{h+1}^{(m,k,j)_h(i;x,a)}\right) - V_{h+1}^\star\left(x_{h+1}^{(m,k,j)_h(i;x,a)}\right)\right)$$

$$\leq O\left(\frac{MSA}{t}\sqrt{H^5\iota} + \sqrt{\frac{SA}{t}H^5\iota} + H^4 SA(M-1)\frac{1}{t}\right),$$

which indicates that, under the intersections of the events of Equation (39), and $\bigcap_{k=1}^{K'}$ Event($k$), for any $(x, a, h) \in \mathcal{S} \times \mathcal{A} \times [H]$, we have

$$|P_3 - P_4| \leq O\left(\frac{MSA}{t}\sqrt{H^7\iota} + \frac{\sqrt{SAH^7\iota}}{\sqrt{t}} + \frac{(M-1)SAH^5}{t}\right).$$

To conclude about the relationship between $W_k(x, a, h)$ and $\left[\mathbb{V}_h V_{h+1}^\star\right](x, a)$, we have that, under the interaction of the events of Equation (39), $\bigcap_{k=1}^{K'}$ Event($k$), Lemma E.4 and Lemma E.5, we have $\forall(x, a, h, k) \in \mathcal{S} \times \mathcal{A} \times [H] \times [K']$,

$$\left|W_k(x, a, h) - \left[\mathbb{V}_h V_{h+1}^\star\right](x, a)\right|$$
$$\leq O\left(\frac{MSA}{t}\sqrt{H^7\iota} + \frac{\sqrt{SAH^7\iota}}{\sqrt{t}} + \frac{(M-1)SAH^5}{t}\right). \tag{42}$$

With this relationship, we can provide the new concentration results. Similar to the proof of Lemma C.3, for a given $(x, a, h) \in \mathcal{S} \times \mathcal{A} \times [H]$, we decompose the summation $\sum_{i=1}^t \tilde{\theta}_t^i X_i$ as follows.

$$\sum_{i=1}^t \tilde{\theta}_t^i X_i = \sum_{i=1}^t \theta_t^i X_i + \sum_{i=1}^t (\tilde{\theta}_t^i - \theta_t^i) X_i.$$

Equation (19) has already provided an upper bound for all $(x, a, h, K') \in \mathcal{S} \times \mathcal{A} \times [H] \times [K]$ for the second summation. Next, we focus on $\left|\sum_{i=1}^t \theta_t^i X_i\right|$. By Azuma-Bernstein Inequality, for any fixed $t' \in \mathbb{N}_+$ and fixed $(x, a, h) \in \mathcal{S} \times \mathcal{A} \times [H]$, for any $p \in (0, 1)$, with probability at least $1 - p$, we have that

$$\left|\sum_{i=1}^{t'} \theta_{t'}^i X_i\right| \leq O\left(\sqrt{\frac{1}{t'}H\left[\mathbb{V}_h V_{h+1}^\star\right](x, a)\log\frac{2}{p}} + \frac{1}{t'}H^2\log\frac{2}{p}\right).$$

After considering the union bound with regard to $(x, a, h) \in \mathcal{S} \times \mathcal{A} \times [H]$ and $t' \leq T_0/H$, we can claim the following conclusion: for any $p \in (0, 1)$, with probability at least $1 - p$, the following relationship holds simultaneously for all $(x, a, h, K') \in \mathcal{S} \times \mathcal{A} \times [H] \times [K]$,

$$\left|\sum_{i=1}^t \theta_t^i X_i\right| \leq O\left(\sqrt{\frac{\iota}{t'}H\left[\mathbb{V}_h V_{h+1}^\star\right](x, a)} + \frac{\iota}{t}H^2\right), t = N_h^{K'}(x, a). \tag{43}$$

The intersection of events of Equation (43) and Equation (19) indicates that the following relationship holds simultaneously for all $(x, a, h, K') \in \mathcal{S} \times \mathcal{A} \times [H] \times [K]$ with $t = N_h^{K'}(x, a)$:

$$\left|\sum_{i=1}^t \tilde{\theta}_t^i(\tilde{\mathbb{E}}_{x,a,h,i} - \mathbb{E}_{x,a,h})V_{h+1}^\star(x_{h+1})\right| \leq O\left(\sqrt{\frac{\iota}{t}H\left[\mathbb{V}_h V_{h+1}^\star\right](x, a)} + \frac{\iota}{t}H^2 + \sqrt{H\iota/t}\right). \tag{44}$$

Combining with the event of Equation (42) for $K'$ replaced by $K' + 1$, we have that

$$\left|\sum_{i=1}^t \tilde{\theta}_t^i\left((\tilde{\mathbb{E}}_{x,a,h,i} - \mathbb{E}_{x,a,h})V_{h+1}^\star(x_{h+1})\right)\right|$$
$$\leq O\left(\sqrt{\frac{\iota}{t}H\left(W_{K'+1}(x, a, h) + \frac{MSA}{t}\sqrt{H^7\iota} + \frac{\sqrt{SAH^7\iota}}{\sqrt{t}} + \frac{(M-1)SAH^5}{t}\right)} + \frac{\iota}{t}H^2 + \sqrt{H\iota/t}\right).$$

Due to $2\sqrt{\frac{H^7 SA\iota}{t}} \le H + \frac{H^6 SA\iota}{t}$, we have that

$$O\left(\sqrt{\frac{\iota}{t}H\left(W_{K'+1}(x,a,h) + \frac{MSA}{t}\sqrt{H^7\iota} + \frac{\sqrt{SAH^7\iota}}{\sqrt{t}} + \frac{(M-1)SAH^5}{t}\right)}\right)$$

$$\le O\left(\sqrt{\frac{\iota}{t}H\left(W_{K'+1}(x,a,h) + \frac{MSA}{t}\sqrt{H^7\iota} + H + \frac{SAH^6\iota}{t} + \frac{(M-1)SAH^5}{t}\right)}\right)$$

$$\le O\left(\sqrt{\frac{\iota}{t}H\left(W_{K'+1}(x,a,h) + \frac{MSA}{t}\sqrt{H^7\iota} + H + \frac{SAH^6\iota}{t} + \frac{(M-1)SAH^5}{t}\right)}\right).$$

Noticing that

$$\frac{MSA}{t}\sqrt{H^7\iota} + \frac{(M-1)SAH^5}{t} = O\left(\frac{MSA}{t}H^5\iota\right),$$

we have

$$\left|\sum_{i=1}^{t} \tilde{\theta}_t^i \left((\tilde{\mathbb{E}}_{x,a,h,i} - \mathbb{E}_{x,a,h})V_{h+1}^\star(x_{h+1}) + r_h^{(m,k,j)_h(i;x,a)} - r_h(x,a)\right)\right|$$

$$\le O\left(\sqrt{\frac{H\iota}{t}(W_{K'+1}(x,a,h) + H)} + \iota\frac{\sqrt{H^7 SA} + \sqrt{MSAH^6}}{t}\right), \tag{45}$$

which indicates that

$$\left|\sum_{i=1}^{t} \tilde{\theta}_t^i \left((\tilde{\mathbb{E}}_{x,a,h,i} - \mathbb{E}_{x,a,h})V_{h+1}^\star(x_{h+1})\right)\right| \le \beta_t(x,a,h)/2$$

when combining with Equation (39) and $c'$ is large enough.

Finally, we are ready to provide proof for Lemma E.1. We let $c'$ to be large enough and Will provide discussion under the intersections of events for Equation (19), Equation (39), Lemma E.4, Lemma E.5 and Equation (43). We know that these events hold simultaneously with probability $1 - c_p p$ for some $c_p > 0$. Next, we will prove Equation (38) by induction. It obviously holds that for all $(x,a,h) \in \mathcal{S} \times \mathcal{A} \times [H]$ when $K' = 1$. We suppose that it holds for every $K' \le K_0'$. When $K' = K_0' + 1$, LHS of Equation (38) indicates that $\bigcap_{k=1}^{K_0'+1}$ Event$(k)$ holds. By the discussion above, this indicates that Equation (45) holds for $K_0' + 1$, by recursions on $H, H-1, \ldots, 1$ (similar to the proof of Lemma 4.3 in Jin et al. (2018)), we can prove that Equation (38) holds for all $(x,a,h) \in \mathcal{S} \times \mathcal{A} \times [H]$ for $K_0' + 1$. This finishes the induction. After we replace $p$ with $p/c_p$, we finish the proof.

### E.2.2 REMAINING PARTS FOR PROVING THEOREM 5.1

Next, we begin to discuss the overall complexity. Similar to Lemma C.5 in Jin et al. (2018), we will provide the following Lemma.

**Lemma E.6.** *For any $p \in (0,1)$, with probability at least $1 - p$,*

$$\sum_{m,k,j,h} \left[\mathbb{V}_h V_{h+1}^{\pi^k}\right](x_h^{m,k,j}, a_h^{m,k,j}) \le O(H\hat{T} + H^3\iota).$$

*Proof.* We assign an order for all the episodes based on the "round first, episode second, agent third" rule and suppose $m(i), k(i), j(i)$ recovers the agent index, round index and within round episode index for the $i-$th episode. Denote $R_i = \sum_{h=1}^{H} \mathbb{V}_h V_{h+1}^{\pi^k}(x_h^{(m,k,j)(i)}, a_h^{(m,k,j)(i)})$ and $\mathcal{F}_{i-1}$ be the $\sigma-$field generated by the information before the $i-$th episode. Similar to the proof of Lemma C.5 in Jin et al. (2018), we have

$$\mathbb{E}[R_i | F_{i-1}] \le H^2,$$
$$0 \le R_i \le H^3,$$

$$\mathrm{Var}[R_i | F_{i-1}] \leq H^5.$$

So, by Azuma-Hoeffding Inequality based on $\sum_{i=1}^{t} R_i$ with regard to the filtration $\{\mathcal{F}_i\}_{i=1}^{\infty}$ and a union bound for $t \leq T_0(1 + \tilde{C})/H + M$, we conclude that

$$\sum_{m,k,j,h} \left[ \mathbb{V}_h V_{h+1}^{\pi^k} \right] (x_h^{m,k,j}, a_h^{m,k,j}) = \sum_{i=1}^{\hat{T}/H} R_i \leq O(H\hat{T} + H^3 \iota).$$

$\square$

We also provide a Lemma that focuses on the concentration of $\xi_h^k$.

**Lemma E.7.** *For any $p \in (0, 1)$, with probability at least $1 - p$, the following relationships holds simultaneously:*

$$\left| \sum_{k=1}^{K} C_h \sum_{h=h'}^{H} \xi_{h+1}^k \right| \leq O(H\sqrt{\hat{T}\iota}), \forall h' \in [H], \tag{46}$$

$$\left| \sum_{k=1}^{K} \sum_{h=1}^{H} \xi_{h+1}^k \right| \leq O(H\sqrt{\hat{T}\iota}), \tag{47}$$

*in which $C_h = \exp(3(h-1)/H)$.*

*Proof.* We first focus on the first event. Denote $V(m,k,j,h) = C_h(\mathbb{P} - \hat{\mathbb{P}})\left( V_{h+1}^{\star} - V_{h+1}^{\pi^k} \right)(x_h^{m,k,j}, a_h^{m,k,j})$ and a simplified notation $\sum_{m,k,j,h:h'} = \sum_{k=1}^{K} \sum_{m=1}^{M} \sum_{j=1}^{n^{m,k}} \sum_{h=h'}^{H-1}$. The quantity we focus on can be rewritten as

$$\sum_{m,k,j,h:h'} V(m,k,j,h),$$

with $|V(m,k,j,h)| \leq O(H)$ as $C_h \leq \exp(3)$. Let $\tilde{V}(\tilde{i})$ be the $\tilde{i}$−th term in the summation that contains $\hat{T}(H - h')/H$ terms, in which the order follows a "round first, episode second, step third, agent fourth" rule. Then the sequence $\{\tilde{V}(\tilde{i})\}$ is a martingale difference. By Azuma-Hoeffding Inequality, for any $p \in (0, 1)$ and $t \in N_+$, with probability at least $1 - p$,

$$\left| \sum_{\tilde{i}=1}^{t} \tilde{V}(\tilde{i}) \right| \leq O\left( H\sqrt{t} \right).$$

Then by applying a union bound with regard to $h' \in [H - 1]$ and all possible $t$ which is divisible by $H - h'$ and knowing that $\hat{T}(H - h')/H \leq T_0(1 + \tilde{C}) + HM$ due to (e) in Lemma B.1, we can claim that, for any $p \in (0, 1)$, with probability at least $1 - p$, the following relationship holds simultaneously:

$$\left| \sum_{k=1}^{K} C_h \sum_{h=h'}^{H} \xi_{h+1}^k \right| = \left| \sum_{\tilde{i}=1}^{\hat{T}(H-h')/H} \tilde{V}(\tilde{i}) \right| \leq O(H\sqrt{\hat{T}\iota}), \forall h' \in [H].$$

The second event can be analyzed similarly for the same conclusion. By combining these two events and re-scaling $p$, we can claim the result. $\square$

Next, we try to find the upper bound for the regret. We pick $c'$ to be large enough and discuss based on the intersection of events of Equation (19), Equation (39), Lemma E.4, Lemma E.5, Equation (43), Lemma E.7, Lemma E.1 and Lemma E.6. They hold simultaneously with probability at least $1 - c'_p p$ where $c'_p > 0$ is a numerical constant and indicates Equation (42). Similar to the discussions in Proof of Theorem 4.1, we can claim that for $\forall h \in [H]$,

$$\sum_{k=1}^{K} \delta_h^k \leq O\left( \sqrt{H^4 \iota \hat{T} SA} + HSA(M-1)\sqrt{H^3 \iota} + MH^2 SA + H^4 SA(M-1) \right), \tag{48}$$

due to $\beta_t(x,a,h) = O(\sqrt{H^3\iota/t})$. In addition, due to the relationship

$$\sum_{k=1}^K \delta_h^k \le \exp(3/H)\sum_{k=1}^K \delta_{h+1}^k + \sum_{k=1}^K \xi_{h+1}^k + O(1)\sum_{k,m,j}\beta_{t_h^{m,k,j}}(x_h^{m,k,j}, a_h^{m,k,j}, h)$$
$$+ O\left(MHSA + H^3SA(M-1)\right),$$

which can be obtained similar to the situation in Appendix C.3 for Proof of Theorem 4.1, denoting $\sum_{k,m,j} = \sum_{k,m,j}\sum_{h=1}^H$, we have

$$\sum_{k=1}^K \delta_1^k \le O(MH^2SA + H^4SA(M-1) + \sqrt{H^2\hat{T}\iota}) + O(1)\sum_{k,m,j,h}\beta_{t_h^{m,k,j}}(x_h^{m,k,j}, a_h^{m,k,j}, h) \quad (49)$$

We will bound the last term by splitting it into two parts.

$$\sum_{k,m,j,h}\beta_{t_h^{m,k,j}}(x_h^{m,k,j}, a_h^{m,k,j}, h) = \sum_{k,m,j,h}\beta_{t_h^{m,k,j}}(x_h^{m,k,j}, a_h^{m,k,j}, h)\mathbb{I}[t_h^{m,k,j} \le M-1]$$
$$+ \sum_{k,m,j,h}\beta_{t_h^{m,k,j}}(x_h^{m,k,j}, a_h^{m,k,j}, h)\mathbb{I}[t_h^{m,k,j} \ge M].$$

For the first part, knowing that $\beta_{t_h^{m,k,j}}(x_h^{m,k,j}, a_h^{m,k,j}, h) \le O(\sqrt{H^3\iota})$, using the similar technique as Proof of Equation (26), we have that

$$\sum_{k,m,j,h}\beta_{t_h^{m,k,j}}(x_h^{m,k,j}, a_h^{m,k,j}, h)\mathbb{I}[t_h^{m,k,j} \le M-1] \le O\left(HSA(M-1)\sqrt{H^3\iota}\right).$$

For the second part, we have that

$$\sum_{k,m,j,h}\beta_{t_h^{m,k,j}}(x_h^{m,k,j}, a_h^{m,k,j}, h)\mathbb{I}[t_h^{m,k,j} \ge M]$$
$$\le \sum_{k,m,j,h}O\left(\sqrt{\frac{H\iota}{t_h^{m,k,j}}(W_{k+1}(x_h^{m,k,j}, a_h^{m,k,j}, h) + H)} + \iota\frac{\sqrt{H^7SA} + \sqrt{MSAH^6}}{t_h^{m,k,j}}\right)$$
$$\cdot \mathbb{I}[t_h^{m,k,j} \ge M].$$

Later on, we use another simplified notation $\sum_{k,m,j,h:M} = \sum_{k,m,j,h}\mathbb{I}[t_h^{m,k,j} \ge M]$. Using the same technique of finding $C''$ in Equation (26), we can find that

$$\sum_{k,m,j,h:M}1/t_h^{m,k,j} \le O(1)\sum_{(x,a,h)\in\mathcal{S}\times\mathcal{A}\times[H]}\sum_{i=M}^{N_h^{K+1}(x,a)-1}1/i \le HSA\iota \quad (50)$$

and

$$\sum_{k,m,j,h:M}1/\sqrt{t_h^{m,k,j}} \le O(1)\sum_{(x,a,h)\in\mathcal{S}\times\mathcal{A}\times[H]}\sum_{i=M}^{N_h^{K+1}(x,a)-1}1/i \le \sqrt{HSA\hat{T}}. \quad (51)$$

So, we have

$$\sum_{k,m,j,h:M}\iota\frac{\sqrt{H^7SA} + \sqrt{MSAH^6}}{t_h^{m,k,j}} \le \iota^2 HSA\left(\sqrt{H^7SA} + \sqrt{MSAH^6}\right).$$

We also have

$$\sum_{k,m,j,h:M}\left(\sqrt{\frac{H}{t_h^{m,k,j}}(W_k(x_h^{m,k,j}, a_h^{m,k,j}, h) + H)}\right)$$
$$\le O(1)\sqrt{\left(\sum_{m,k,j,h:M}(W_{k+1}(x_h^{m,k,j}, a_h^{m,k,j}, h) + H)\right)\left(\sum_{m,k,j,h:M}\frac{H}{t_h^{m,k,j}}\right)}$$
$$\le O(1)\sqrt{H^3SA\hat{T}\iota} + O(1)\sqrt{H^2SA\iota}\sqrt{\sum_{m,k,j,h:M}W_{k+1}(x_h^{m,k,j}, a_h^{m,k,j}, h)},$$

where the first inequality follows from Cauchy's inequality and the second inequality is due to Equation (50).

To conclude, we have

$$\sum_{k,m,j,h} \beta_{t_h^{m,k,j}}(x_h^{m,k,j}, a_h^{m,k,j}, h)$$

$$= O\left(HSA(M-1)\sqrt{H^3\iota} + \iota^2\sqrt{H^9 S^3 A^3} + \iota^2\sqrt{MS^3 A^3 H^8} + \right.$$

$$\left. + \sqrt{H^3 SA\hat{T}\iota^2} + \sqrt{H^2 SA\iota^2} \sqrt{\sum_{m,k,j,h:M} W_{k+1}(x_h^{m,k,j}, a_h^{m,k,j}, h)}\right). \qquad (52)$$

Next, we try to find an upper bound for

$$\sqrt{\sum_{m,k,j,h:M} W_{k+1}(x_h^{m,k,j}, a_h^{m,k,j}, h)}.$$

We know that

$$W_k(x, a, h) \le \mathbb{V}_h\left[V_{h+1}^{\pi^k}\right](x, a) + \left|\left[\mathbb{V}_h V_{h+1}^{\star}\right](x, a) - W_k(x, a, h)\right|$$

$$+ \left|\left[\mathbb{V}_h V_{h+1}^{\star}\right](x, a) - \left[\mathbb{V}_h V_{h+1}^{\pi^k}\right](x, a)\right|.$$

By Lemma E.6,

$$\sqrt{\sum_{m,k,j,h:M} \mathbb{V}_h\left[V_{h+1}^{\pi^k}\right](x_h^{m,k,j}, a_h^{m,k,j})} \le O\left(\sqrt{H\hat{T} + H^3\iota}\right).$$

By Equation (42), denoting $\tilde{t}_h^{m,k,j} = N_h^{k+1}(x_h^{m,k,j}, a_h^{m,k,j})$,

$$\sqrt{\sum_{m,k,j,h:M} \left|\left[\mathbb{V}_h V_{h+1}^{\star}\right](x_h^{m,k,j}, a_h^{m,k,j}) - W_{k+1}(x_h^{m,k,j}, a_h^{m,k,j}, h)\right|}$$

$$\le O\left(\sqrt{\sum_{m,k,j,h:M}\left(\frac{MSA}{\tilde{t}_h^{m,k,j}}\sqrt{H^7\iota} + \frac{\sqrt{SAH^7\iota}}{\sqrt{\tilde{t}_h^{m,k,j}}} + \frac{(M-1)SAH^5}{\tilde{t}_h^{m,k,j}}\right)}\right).$$

As $\tilde{t}_h^{m,j,k} \ge t_h^{m,j,k}$, we have that

$$\sqrt{\sum_{m,k,j,h:M} \left|\left[\mathbb{V}_h V_{h+1}^{\star}\right](x_h^{m,k,j}, a_h^{m,k,j}) - W_{k+1}(x_h^{m,k,j}, a_h^{m,k,j}, h)\right|}$$

$$\le O\left(\sqrt{\sum_{m,k,j,h:M}\left(\frac{MSA}{t_h^{m,k,j}}\sqrt{H^7\iota} + \frac{\sqrt{SAH^7\iota}}{\sqrt{t_h^{m,k,j}}} + \frac{(M-1)SAH^5}{t_h^{m,k,j}}\right)}\right)$$

$$= O\left(\sqrt{MH^{4.5}S^2 A^2\iota^{1.5} + H^4 SA\sqrt{\hat{T}\iota} + (M-1)H^6 S^2 A^2}\right),$$

where the last inequality is due to Equation (50) and Equation (51). We also have

$$\sqrt{\sum_{m,k,j,h:M} \left|\left[\mathbb{V}_h V_{h+1}^{\star}\right](x_h^{m,k,j}, a_h^{m,k,j}) - \left[\mathbb{V}_h V_{h+1}^{\pi^k}\right](x_h^{m,k,j}, a_h^{m,k,j})\right|}$$

$$\le \sqrt{\sum_{m,k,j,h} \left|\left[\mathbb{V}_h V_{h+1}^{\star}\right](x_h^{m,k,j}, a_h^{m,k,j}) - \left[\mathbb{V}_h V_{h+1}^{\pi^k}\right](x_h^{m,k,j}, a_h^{m,k,j})\right|}.$$

Next, we will show that, for any $(x, a, h) \in \mathcal{S} \times \mathcal{A} \times [H]$,

$$\left| \left[ \mathbb{V}_h V_{h+1}^{\star} \right] (x, a) - \left[ \mathbb{V}_h V_{h+1}^{\pi^k} \right] (x, a) \right| \leq O(H) \left( \left[ \mathbb{P}_h V_{h+1}^{\star} \right] (x, a) - \left[ \mathbb{P}_h V_{h+1}^{\pi^k} \right] (x, a) \right).$$

Suppose that $u, v$ are random variables such that $u$ follows the distribution of $V_{h+1}^{\star}(x_{h+1})$ under $\pi_{\star}$ when $(x_h, a_h) = (x, a)$, and $v$ follows the distribution of $V_{h+1}^{\pi^k}(x_{h+1})$ under $\pi^k$ when $(x_h, a_h) = (x, a)$ and $u \geq v$. The third requirement is reasonable because the distribution of $x_{h+1}$ only depends on $(x, a)$ and $V_{h+1}^{\star}(x_{h+1}) \geq V_{h+1}^{\pi^k}(x_{h+1})$. We have that $u, v \leq H$. So,

$$\begin{aligned}
\left| \left[ \mathbb{V}_h V_{h+1}^{\star} \right] (x, a) - \left[ \mathbb{V}_h V_{h+1}^{\pi^k} \right] (x, a) \right| &= |\mathrm{Var}(u) - \mathrm{Var}(v)| \\
&\leq |\mathbb{E}(u^2) - \mathbb{E}(v^2) + (\mathbb{E}v)^2 - (\mathbb{E}u)^2| \\
&\leq |\mathbb{E}(u - v)(u + v) + (\mathbb{E}v - \mathbb{E}u)(\mathbb{E}v + \mathbb{E}u)| \\
&\leq O(H)(|\mathbb{E}(u - v)| + \mathbb{E}|u - v|) \\
&= O(H)\mathbb{E}(u - v).
\end{aligned}$$

This proves the conclusion. Using the conclusion, we can find that

$$\sum_{m,k,j,h} \left| \left[ \mathbb{V}_h V_{h+1}^{\star} \right] (x_h^{m,k,j}, a_h^{m,k,j}) - \left[ \mathbb{V}_h V_{h+1}^{\pi^k} \right] (x_h^{m,k,j}, a_h^{m,k,j}) \right|$$

$$\leq O(H) \sum_{m,k,j,h} \left( \left[ \mathbb{P}_h V_{h+1}^{\star} \right] (x_h^{m,k,j}, a_h^{m,k,j}) - \left[ \mathbb{P}_h V_{h+1}^{\pi^k} \right] (x_h^{m,k,j}, a_h^{m,k,j}) \right)$$

$$= O(H) \sum_{h=1}^{H} \sum_{k=1}^{K} (\delta_{h+1}^k - \phi_{h+1}^k + \xi_{h+1}^k)$$

$$\leq O(H) \sum_{h=1}^{H} \sum_{k=1}^{K} (\delta_{h+1}^k + \xi_{h+1}^k),$$

in which the last inequality is due to $\phi_h^k \geq 0$ based on Equation (38). By Equation (48) and Lemma E.7, we have

$$O(H) \sum_{h=1}^{H} \sum_{k=1}^{K} (\delta_{h+1}^k + \xi_{h+1}^k) = O\left( \sqrt{H^8 \iota \hat{T} S A} + H^3 SA(M - 1)\sqrt{H^3 \iota} + MH^4 SA + H^6 SA(M - 1) \right).$$

So, we have

$$\sum_{m,k,j,h:M} W_{k+1}(x_h^{m,k,j}, a_h^{m,k,j}, h)$$

$$\leq O\left( H\hat{T} + H^3 \iota + MH^4 SA + \sqrt{H^8 \hat{T} SA \iota} + H^6 SA(M - 1) + H^2 SA(M - 1)\sqrt{H^5 \iota} \right)$$

$$+ O\left( MS^2 A^2 \sqrt{H^9 \iota^3} + SA\sqrt{H^8 \hat{T} \iota} + S^2 A^2 H^6 (M - 1) \right)$$

$$= O\left( H\hat{T} + MH^{4.5} S^2 A^2 \iota^{1.5} + H^4 SA\sqrt{\hat{T} \iota} + H^6 S^2 A^2 (M - 1) \right),$$

where the last relationship is due to

$$(M - 1)H^6 S^2 A^2 \geq (M - 1)H^6 SA, \quad H^4 SA\sqrt{\hat{T} \iota} \geq \sqrt{H^8 \hat{T} SA \iota}$$

and

$$MS^2 A^2 \sqrt{H^9 \iota^3} \geq MH^4 SA + H^3 \iota + H^2 SA(M - 1)\sqrt{H^5 \iota}.$$

Inserting it into Equation (52), we have

$$
\sum_{k,m,j,h} \beta_{t_h^{m,k,j}}(x_h^{m,k,j}, a_h^{m,k,j}, h)
$$

$$
= O\left( HSA(M-1)\sqrt{H^3\iota} + \iota^2\sqrt{H^9 S^3 A^3} + \iota^2\sqrt{MS^3 A^3 H^8} + \right.
$$

$$
\left. + \sqrt{H^3 SA\hat{T}\iota^2} + \sqrt{H^2 SA\iota^2}\sqrt{\sum_{m,k,j,h:M} W_{k+1}(x_h^{m,k,j}, a_h^{m,k,j}, h)} \right)
$$

$$
= O\left( HSA(M-1)\sqrt{H^3\iota} + \iota^2\sqrt{H^9 S^3 A^3} + \iota^2\sqrt{MS^3 A^3 H^8} + \sqrt{H^3 SA\hat{T}\iota^2} \right.
$$

$$
\left. + \sqrt{MH^{6.5} S^3 A^3 \iota^{3.5}} + \sqrt{H^6 S^2 A^2 \hat{T}^{0.5}\iota^{2.5}} + \sqrt{H^8 S^3 A^3 (M-1)\iota^2} \right).
$$

Due to

$$
\sqrt{MH^8 S^3 A^3 \iota^4} \geq \sqrt{MH^{6.5} S^3 A^3 \iota^{3.5}},
$$
$$
\sqrt{H^8 S^3 A^3 (M-1)\iota^2} \leq \iota^2\sqrt{MS^3 A^3 H^8}
$$

and

$$
\sqrt{H^6 S^2 A^2 \hat{T}^{0.5}\iota^{2.5}} \leq H^{4.5} S^{1.5} A^{1.5}\iota^{1.5} + \sqrt{\hat{T}SAH^3\iota^2} \leq H^{4.5} S^{1.5} A^{1.5}\iota^2 + \sqrt{\hat{T}SAH^3\iota^2},
$$

we have

$$
\sum_{k,m,j,h} \beta_{t_h^{m,k,j}}(x_h^{m,k,j}, a_h^{m,k,j}, h)
$$

$$
\leq O\left( HSA(M-1)\sqrt{H^3\iota} + \iota^2\sqrt{H^9 S^3 A^3} + \iota^2\sqrt{MS^3 A^3 H^8} + \sqrt{H^3 SA\hat{T}\iota^2} \right).
$$

Inserting it into Equation (49), we have

$$
\text{Regret}(T) \leq \sum_{k=1}^{K} \delta_1^k
$$

$$
= O\left( MH^2 SA + H^4 SA(M-1) + HSA(M-1)\sqrt{H^3\iota} \right.
$$

$$
\left. + \iota^2\sqrt{H^9 S^3 A^3} + \iota^2\sqrt{MS^3 A^3 H^8} + \sqrt{H^3 SA\hat{T}\iota^2} \right).
$$

Finally, for the probability of the intersection of all the events, if we use $p/c_p'$ to replace $p$, we complete the proof.

## F   NUMERICAL EXPERIMENTS

In this section, we conduct experiments in a synthetic environment to validate the theoretical performances of FedQ-Hoeffding, FedQ-Beinstein, and compare with their single-user counterparts UCB-H and UCB-B (Jin et al., 2018), respectively.

**Synthetic Environment.** We generate a synthetic environment to evaluate the proposed algorithms. We set the number of states $S$ to be 3, the number of actions $A$ for each state to be 2, and the episode length $H$ to be 5. The reward $r_h(s, a)$ for each state-action pair and each step is generated independently and uniformly at random from $[0, 1]$. We also generate the transition kernel $P_h(\cdot \mid s, a)$ from an $S$-dimensional simplex independently and uniformly at random for each state-action pair and each step. Such procedure guarantees that the synthetic environment is a proper tabular MDP.

Under the given MDP, we set $M = 10$ and $T/H = 3 \times 10^4$ for FedQ-Hoeffding, FedQ-Beinstein, and $T/H = 3 \times 10^5$, $M = 1$ for UCB-H and UCB-B. Thus, the total number of episodes is $3 \times 10^5$ for all four algorithms. We choose $c = \iota = 1$ for all algorithms. For each episode, we randomly choose the initial state uniformly from $S$ states. We collect 10 sample paths under all algorithms under the same MDP environment, and plot $\text{Regret}(T)/\sqrt{MT}$ versus $MT/H$ in Figure 1. The

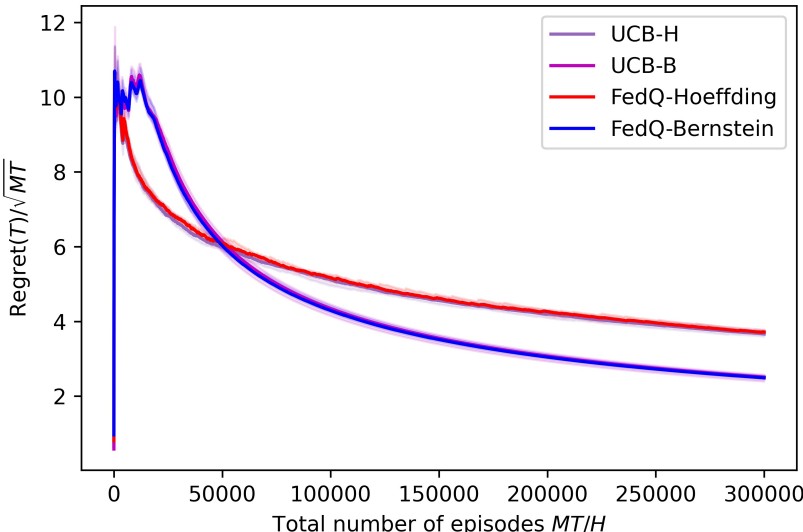

Figure 1: Regret comparison.

solid line represents the median of the 10 sample paths, while the shaded area shows the 10th and 90th percentiles. As we can see, both FedQ-Hoeffding and FedQ-Beinstein stay very close to their single-agent counterpart, indicating that FedQ-Hoeffding achieves linear speedup with respect to the number of clients $M$, as predicted by Theorem 4.1 and Theorem 5.1. Besides, as time progresses, FedQ-Beinstein achieves lower regret than FedQ-Hoeffding, which is consistent with the theoretical results as well.

We also track the number of communication rounds throughout the learning process under FedQ-Hoeffding and FedQ-Bernstein, and plot the median profiles as well as the 10th and 90th percentiles in Figure 2. Both curves exhibit sublinear growth, corroborating the theoretical result in Theorem 4.2. Besides, the total number of communication rounds under FedQ-Bernstein becomes lower than that under FedQ-Hoeffding as $T$ becomes sufficiently large. This is because after the more active early-stage exploration of FedQ-Bernstein, it reaches a more stable policy, under which the synchronization triggered by $(x, a, h)$s that are less likely to be visited under the optimal policy rarely happens.

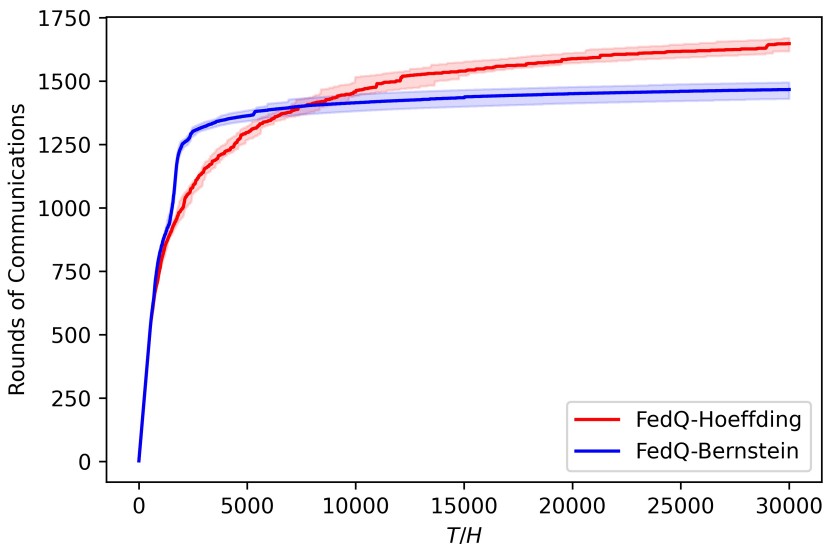

Figure 2: Total number of communication rounds as a function of $T/H$.

