# OpenReview forum: "Federated Q-Learning: Linear Regret Speedup with Low Communication Cost"
_ICLR.cc/2024/Conference — ICLR 2024 poster_

### Official Review · Reviewer_dPut · 2023-10-26

**Soundness:** 3 good
**Presentation:** 3 good
**Contribution:** 2 fair
**Rating:** 5
**Confidence:** 3

**Summary:**

This paper studies the federated RL and proposes two model-free algorithms---FedQ-Hoeffding and FedQ-Bernstein---that achieve regret speedup and low communication.

**Strengths:**

Overall, this paper has a fair contribution. It proposes two federated RL algorithms with a regret speed up and logarithmic communication.

**Weaknesses:**

It would be great if the authors could provide some empirical validations for their algorithm. I understand this is a theoretical work, but it is always helpful to corroborate the theoretical results with some experiments.

**Questions:**

- Why define $\tilde{C}$ as it is used only once (above Eq. (2)) in the main paper?
- Case 1 at Page 5, it should be $\eqqcolon i_0$ instead of $\coloneqq i_0$.

---

> ### Author Response · Authors · 2023-11-18
> **Responses to Reviewer dPut**
>
> We thank the reviewer for the careful reading and thoughtful comments. We address the reviewer's questions in the following and have revised the paper accordingly. The changes are marked in blue in our revision. We hope the responses below and changes in the paper address the reviewer's concerns.
>
> **Weakness:** It would be great if the authors could provide some empirical validations for their algorithm. I understand this is a theoretical work, but it is always helpful to corroborate the theoretical results with some experiments.
>
> **Response:** Thank you for the suggestion. We have performed some experiments and updated our draft to include the experimental results of FedQ-Hoeffding and the comparison with the single-agent Q-learning algorithm UCB-Hoeffding in Appendix F. The source code has been uploaded as a supplementary file. The results clearly exhibit the linear speedup of the learning regret and logarithmic communication cost, which corroborate our theoretical results. We will perform more experiments to thoroughly validate the performances of FedQ-Hoeffding and FedQ-Bernstein, and include the results in the next version of this paper.
>
> **Question 1:** Why define $\tilde{C}$ as it is used only once (above Eq. (2)) in the main paper?
>
> **Response:** Thank you for the careful reading.  $\tilde{C}$ has been used frequently in the proofs in the appendix. We have followed your suggestion to only keep this notation in the statements of the theorems in the main paper.
>
> **Question 2:** Case 1 at Page 5, it should be $=:i _0$ instead of $:=i _0$.
>
> **Response:** Thank you for your careful reading. We have revised it accordingly in the updated version of this paper.
>
>
>
> -----
>
> We thank the reviewer again for the helpful comments and suggestions for our work. If our response resolves your concerns to a satisfactory level, we kindly ask the reviewer to consider raising the rating of our work. Certainly, we are more than happy to address any further questions that you may have.

---

> > ### Comment · Reviewer_dPut · 2023-11-19
> >
> > The reviewer thanks the author for their response. The reviewer will take these into consideration in the later discussion with AC.

---

> > > ### Comment · Reviewer_dPut · 2023-11-19
> > >
> > > The reviewer thinks the empirical simulation is unsatisfactory. For one thing, the mentioned baseline algorithms in Table 1 are not compared, and the author does not explain why not compare with them. Especially, while one key improvement in this paper is on low communication, the numerical results cannot support this advantage by only reporting the performance of the algorithm designed in this paper. For another thing, the authors may also consider comparing FedQ-Hoeffding and FedQ-Bernstein.

---

> ### Author Response · Authors · 2023-11-19
>
> We thank the reviewer for the prompt reply. As acknowledged by the reviewer, this is a theoretical work, and we followed the reviewer's suggestion to conduct some experiments within the short rebuttal window in order to verify the linear speedup and logarithmic communication cost shown in our theoretical results. We have worked tirelessly to implement UCB-H and FedQ-Hoeffding within the short time window and obtained those experimental results, which indeed corroborate our theoretical results as we expected.
>
> We agree with the reviewer that the experimental results are not comprehensive at this stage. This is because those works listed in Table 1 are all theoretical works, and they do not provide any experimental results or source code. It is thus impossible for us to implement those algorithms and perform the comparison given the short deadline we have. We are currently working on the implementation of FedQ-Beinstein, and we will include more comprehensive comparison with other works (if possible) in the next version of this work.

---

> > ### Author Response · Authors · 2023-11-20
> > **Updated numerical experiments that include FedQ-Beinstein**
> >
> > We would like to let the reviewer know that we have successfully implemented FedQ-Beinstein and UCB-B. The new experimental results have been included in the latest version of this paper. We are striving to implement other algorithms from related works in order to have some comparison between those works and our algorithms. We will update the draft when such results become available.

---

### Official Review · Reviewer_8uCJ · 2023-10-29

**Soundness:** 3 good
**Presentation:** 3 good
**Contribution:** 2 fair
**Rating:** 5
**Confidence:** 3

**Summary:**

This work considers a federated Q-learning for tabular episodic MDP, where multiple agents collaboratively explore the environment and learn an optimal Q-value with the aid of a central server. They proposed two federated Q-learning algorithms (FedQ-Hoeffding, FedQ-Bernstein) with event-triggered policy switching and synchronization. The algorithms provably achieve linear regret speedup while requiring communication cost logarithmically scaling with the total number of samples (T).

**Strengths:**

- They proposed a federated Q-learning algorithm with event-triggered synchronization, which  guarantees logarithmic communication cost in terms of T.
- They provided a finite-time regret analysis on the federated Q-learning algorithm with policy switching and proved linear regret speedup.

**Weaknesses:**

* Although the algorithm considers a setting that agents can collaboratively explore by changing their policies, the algorithm requires all agents to use the same fixed policy during local iterations, which seems to be quite restrictive. It would be nice if you could elaborate on the necessity of these restrictions.
* Although the paper claims that the event-triggered synchronization method is a key to reducing communication costs, the order of communication costs they showed in this paper seems to be larger than the one shown in [1], which uses just a fixed communication period. The communication cost shown in [1] not only logarithmically scales with T, but also is more efficient in terms of other factors (M: number of agents, $(1-\gamma)^{-1} (\approx H)$: length of horizon). I understand that direct comparison might be difficult given the settings are different, but I’m still not convinced that the communication cost shown in this paper is especially low. It would be nice if you could provide more detailed comparisons with recent literature to help better understand on the communication efficiency.

[1]: Jiin Woo, Gauri Joshi, and Yuejie Chi. The blessing of heterogeneity in federated q-learning: Linear speedup and beyond. In International Conference on Machine Learning, pp. 37157–37216, 2023.

**Questions:**

* The previous federated Q-learning literature [1] already showed that communication cost logarithmically scaling with T is achievable without using event-triggered synchronization (with fixed communication period). Is there any reason to introduce event-triggered synchronization especially in this setting?
* The algorithm seems to fix the Q-values and behavior policies to be the same for all agents during local iterations. However, a setting that agents can flexibly change their policy based on their local observations before the next synchronization seems more natural to me, especially in the federated setting. Would it hurt the performance if they can change their policies and Q-estimates locally during local updates? I wonder if letting agents to change their policies can introduce some diversity in their exploration, which might be an advantage in learning.

---

> ### Author Response · Authors · 2023-11-18
> **Responses to Reviewer 8uCJ (part one)**
>
> We thank the reviewer for the careful reading and thoughtful comments. We address the reviewer's questions in the following and have revised the paper accordingly. The changes are marked in blue in our revision. We hope the responses below and changes in the paper address the reviewer's concerns. Due to character limits of openreview, our reply is splitted into three parts.
>
>
> **Weakness 1:** Although the algorithm considers a setting that agents can collaboratively explore by changing their policies, the algorithm requires all agents to use the same fixed policy during local iterations, which seems to be quite restrictive. It would be nice if you could elaborate on the necessity of these restrictions.
>
> **Response:** The main reason that our algorithm adopts the same policy for all agents is two-fold: *First, such a fixed policy design facilitates tractable global information aggregation in each round*. In federated RL, the distribution of the collected samples in each episode depends on the executed policy. Different exploration policy leads to different distribution of the data samples. If we allow local clients to adopt different policies, or allow the exploration policies to update locally in each episode, the data samples will no longer be IID across clients or episodes in the same round, leading to significant challenges when aggregating them for an analytically tractable global estimate.
>
> *Second, the fixed policy design naturally fits the homogeneous MDP setting and the low regret and communication cost requirements.* Given that the MDP environments encountered by the local clients are homogeneous, in order to achieve low learning regret across all clients, it is expected that the policies adopted by the clients will eventually converge to the same optimal policy. Meanwhile, in order to save the communication cost, it is desirable to keep the local policy fixed, such that the required information exchange is minimized.
>
> We note that such a fixed exploration policy design is widely adopted in the study of federated RL and federated bandits, such as [3]-[6].
>
> [3] Yiding Chen, Xuezhou Zhang, Kaiqing Zhang, Mengdi Wang, and Xiaojin Zhu. Byzantine-robust online and offline distributed reinforcement learning. In International Conference on Artificial Intelligence and Statistics, pp. 3230–3269. PMLR, 2023.
>
> [4] Flint Xiaofeng Fan, Yining Ma, Zhongxiang Dai, Wei Jing, Cheston Tan, and Bryan Kian Hsiang Low. Fault-tolerant federated reinforcement learning with theoretical guarantee. Advances in Neural Information Processing Systems, 34:1007–1021, 2021.
>
> [5] Chengshuai Shi and Cong Shen. Federated multi-armed bandits. In Proceedings of the AAAI Conference on Artificial Intelligence, volume 35, pp. 9603–9611, 2021.
>
> [6] Ruiquan Huang, Weiqiang Wu, Jing Yang, and Cong Shen. Federated linear contextual bandits. Advances in neural information processing systems 34 (2021): 27057-27068.

---

> ### Author Response · Authors · 2023-11-18
> **Responses to Reviewer 8uCJ (part two)**
>
> **Weakness 2:** Communication cost comparison with Woo et al (2023).
>
> **Response:** As pointed out by the reviewer, Woo et al (2023) considers a very different setting (infinite-horizon MDP with time-invariant transition kernels) with different learning objective (near-optimal Q-estimate at the central server). It also assumes the accessibility to a simulator for each agent in the synchronous setting, which generates a new sample
> for *every state-action pair* independently at every iteration. Thus, it does not require any exploration policy, which is in stark contrast to our problem and cannot be compared fairly.
>
> For the asynchronous setting, since the objective is to learn a near-optimal Q-estimate *at the central server*, the local clients adopt *fixed exploration policies* through the learning process to collect trajectories only.
> As a result, it is able to achieve a communication round upper bound in $O(\frac{ M}{\mu(1-\gamma)}\log T)$ with a fixed communication period design. Here $\mu = \mu _{min}$ with equal weight assignment, and $\mu = \mu _{avg}$ with unequal weight assignment, where $\mu _{min} = \min _{x,a,m} \mu^m(x,a)$ and $\mu _{avg} = \min _{x,a}\frac{1}{M}\sum _{m=1}^M\mu^m(x,a)$, and $\mu^m(x,a)$ is the visitation frequency for $(x,a)$ under the exploration policy at agent $m$.
>
> We note that the total number of communication rounds in both our work and Woo et al (2023) scales linearly in $M$. As pointed out in our response to **Question 1 of reviewer tdCn**, the dependency on $M$ under our algorithm can be further reduced or even removed without affecting the regret if we relax the synchronization requirement.
>
> In terms of the dependency $S,A,H$, we first note that $\mu _{min} \leq \mu _{avg} \leq \frac{1}{SA}$. Besides, if we use the approximation $(1-\gamma)^{-1} \approx H$ as suggested by the reviewer, the bound in Woo et al (2023) becomes $O(MHSA\log T)$. Compared with our bound $O(MH^3SA\log T)$, we note that the only difference lies in the order of $H$. We emphasize that in the episodic MDP we consider, we have to handle *$H$ different transition kernels* for the $H$ steps in each episode. Roughly speaking, learning such MDPs is at least $H$ times harder than stationary infinite-horizon MDPs. Thus, it is no surprising that our dependency on $H$ has a higher order. Whether we can reduce the dependency on $H$ from $H^3$ to $H$ by assuming a constant transition kernel in each step requires some significant modification of the algorithm design and analysis, and we leave it to our future work.
>
>
> **Question 1:** The previous federated Q-learning literature Woo et al (2023) already showed that communication cost logarithmically scaling with $T$ is achievable without using event-triggered synchronization (with fixed communication period). Is there any reason to introduce event-triggered synchronization especially in this setting?
>
> **Response:** As pointed out in our response to **Weakness 2** above, Woo et al (2023) consider a very different setting (infinite-horizon MDP with time-invariant transition kernels) with different learning objective (near-optimal Q-estimate at the central server), which enables them to achieve logarithmic communication cost with fixed communication period. However, in our setting, we consider episodic MDPs with different transition kernels $P _h$ in each step $h$, and we aim to achieve low learning regret across all clients. Thus, in contrast to the fixed policy adopted by the clients throughout the learning process in Woo et al (2023), in our setting, it is necessary for the clients to adaptively change their local exploration policy and eventually converge to the optimal policy. On the other hand, in order to reduce the communication cost, it is also desirable to have the local exploration policy stays fixed for a period of time. It thus requires a sophisticated design to achieve low regret and low communication cost at the same time, **which in general cannot be achieved by a fixed communication period design**.
>
> Motivated by the event-triggered policy switching in the single-agent setting, which has been shown to achieve low regret with logarithmic policy switching cost, we adopt the event-triggered synchronization in our setting. As we have explained in our response to **Question 1 of reviewer tdCn**, such event-triggered synchronization essentially guarantees **exponentially increasing round lengths**, which is critical to guarantee the logarithmic communication cost. Besides, when combining this technique with the new equal weight assignment in global aggregation, we are able to bound the estimation error in each round using the novel round-wise approximation technique, and obtain low learning regret at the same time.

---

> > ### Author Response · Authors · 2023-11-18
> > **Responses to Reviewer 8uCJ (part three)**
> >
> > **Question 2:** The algorithm seems to fix the Q-values and behavior policies to be the same for all agents during local iterations. However, a setting that agents can flexibly change their policy based on their local observations before the next synchronization seems more natural to me, especially in the federated setting. Would it hurt the performance if they can change their policies and Q-estimates locally during local updates? I wonder if letting agents to change their policies can introduce some diversity in their exploration, which might be an advantage in learning.
> >
> > **Response:** Please refer to our response to **Weakness 1** regarding why a fixed exploration policy is desirable in our setting.
> >
> > -----
> >
> > We thank the reviewer again for the helpful comments and suggestions for our work. If our response resolves your concerns to a satisfactory level, we kindly ask the reviewer to consider raising the rating of our work. Certainly, we are more than happy to address any further questions that you may have.

---

> > > ### Comment · Reviewer_8uCJ · 2023-11-21
> > >
> > > Thanks to the authors for the detailed responses. I have several follow-up questions for further clarification.
> > >
> > > 1. Now I understand that the fixed policy design is necessary for generating i.i.d. data and the tractability of global estimates in analysis. However, I am still confused if the fixed policy design choice is fundamentally necessary in practice or if this is a simplification for analysis. From the "exploration" perspective, making all agents choose different actions rather than a fixed common action seems more efficient, especially when round lengths are very long. Could you elaborate on some simple intuitive examples or empirical demonstrations where adaptively updating local policy and Q-values using newly obtained local observation before synchronization can harm convergence?
> > >
> > > 2. I am still not convinced that the communication efficiency shown in this work is especially novel. As the authors explained, the communication cost shown in this paper is comparable to, or worse than, the one suggested in Woo et al (2023). Although there exist differences in analyzing episodic MDP and infinite MDP, I am not convinced how the difference causes some special challenges in the "communication" perspective. Could you provide some intuitive examples or empirical demonstrations that explain the communication challenges stemming from finite-horizon or transition kernels for $h$, other than general differences or challenges previously discussed in the single-agent setting?

---

> > > > ### Author Response · Authors · 2023-11-22
> > > > **Responses to new comments from  Reviewer 8uCJ (part one)**
> > > >
> > > > We thank the reviewer for the valuable feedback! We address the reviewer's questions as follows.
> > > >
> > > >  **Q1:** Now I understand that the fixed policy design is necessary for generating i.i.d. data and the tractability of global estimates in analysis. However, I am still confused if the fixed policy design choice is fundamentally necessary in practice or if this is a simplification for analysis. From the "exploration" perspective, making all agents choose different actions rather than a fixed common action seems more efficient, especially when round lengths are very long. Could you elaborate on some simple intuitive examples or empirical demonstrations where adaptively updating local policy and Q-values using newly obtained local observation before synchronization can harm convergence?
> > > >
> > > > **A1:** Thank you for the feedback. At this stage, we cannot claim that the fixed policy design is fundamentally necessary. It may be possible that updating local policy and Q-values using newly obtained local observation before synchronization can achieve comparable performances with our fixed policy design. In principle, this method cannot do worse than fixed policy because it includes the fixed policy as a special case. The main difficulty comes from the analysis, as elaborated in our previous response.
> > > >
> > > > In addition, as we have shown in the experimental results in Appendix F in our revised paper, both FedQ-Hoeffding and FedQ-Beinstein achieve **almost perfect** linear regret speedup compared with their single-agent counterparts. We would like to emphasize that the single-agent algorithms adaptively update policies in each episode. Thus, the experimental results show that, with the fixed local policy design, **our federated RL algorithms essentially achieve the best we could hope for even if we allow sequential data collection and adaptive policy updating**. Whether such a flexible policy updating design will lead to better exploration in other federated RL algorithms, such as model-based approaches remains unknown, and we leave it as our future work.
> > > >
> > > > **Q2:** I am still not convinced that the communication efficiency shown in this work is especially novel. As the authors explained, the communication cost shown in this paper is comparable to, or worse than, the one suggested in Woo et al (2023). Although there exist differences in analyzing episodic MDP and infinite MDP, I am not convinced how the difference causes some special challenges in the "communication" perspective. Could you provide some intuitive examples or empirical demonstrations that explain the communication challenges stemming from finite-horizon or transition kernels for $h$, other than general differences or challenges previously discussed in the single-agent setting?
> > > >
> > > > **A2:** Thank you for your question. In the following, we will first take a closer look at the time-invariant infinite MDPs studied in [1] and the rational behind their communication scheme design. Next, we will compare the finite-horizon MDPs studied in our work with [1], and elaborate how the differences in problem settings affect the corresponding communication scheme design.
> > > >
> > > > There are two major factors that affect the algorithm design in [1]: (i) The infinite-horizon MDPs with a time-invariant transition kernel. (ii) The learning objective to identify a near-optimal Q-estimate at the central server at the end of time-horizon. Because of (ii), the exploration policies adopted by the local agents do not need to converge to the optimal policy; Rather, as long as the state-action space is sufficiently covered by the data samples collected by the local agents, the central server can output a near-optimal Q-estimate.
> > > > Because of this, under the asynchronous setting, each agent adopts a *fixed exploration policy* throughout the learning process. The fixed exploration policy, together with (i), ensures that after a burn-in cost ($T _0$ and $\tilde{T} _0$ in Theorem 4.2 and Theorem 4.3, respectively),   the distribution of the state-action pairs in each time step at each local agent becomes very close to the corresponding stationary distribution under the uniform ergodicity assumption (Assumption 4.1). Thus, the corresponding **estimation error of the local and global Q-estimates in each round can be controlled by the stationary distribution and the length of the round**. It thus becomes very natural to adopt a fixed communication round design, as varying the round length does not have any benefit but complicating the analysis.
> > > >
> > > > (please check additional contents in the second part.)

---

> > > > > ### Author Response · Authors · 2023-11-22
> > > > > **Responses to new comments from Reviewer 8uCJ (part two)**
> > > > >
> > > > > Now, let us examine the problem setting of our work. Compared with [1], our problem setting has two major differences: (a) Episodic MDPs with step-varying transition kernels $P _h$. (b) The learning objective to achieve low learning regret across all clients. Because of (b), the local agents need to adopt exploration policies that converge to the optimal policy. In other words, we cannot stick with the same exploration policy for any client throughout the learning process. This is in stark contrast to factor (ii) in the discussion above. As a result, it becomes difficult to use the stationary distribution to control the estimation error in each round: Achieving a stationary distribution requires to follow a policy for a large number of episodes (burn-in cost). Factor (a) that the considered episodic MDP has step-varying transition kernels makes the burn-in cost even higher. Even if we choose to stick with a fixed exploration policy in a round (as our algorithms do), it will incur **a burn-in cost in each round** before the agent starts to follow the stationary distribution in that round, which cannot be amortized in time. Thus, it becomes prohibitive to adopt a similar fixed communication round design from [1] in our setting. Instead, we adopt an event-triggered synchronization scheme, which avoids the stationary distribution based analysis and the corresponding burn-in cost in each round. Such event-triggered scheme enables us to have a finer control on the number of visits of $(x,a,h)$ in each round compared with the stationary distribution-based analysis, leading to **tighter regret bounds**. Meanwhile, such synchronization scheme ensures exponentially increasing round lengths (see our response to Question 1 of reviewer tdCn), leading to **logarithmic communication cost**.
> > > > >
> > > > > As a side point, we also note the communication cost in [1] highly depends on the parameters of the MDP as well as the exploration policy, which for some cases may become very large.  As an example, consider a simple MDP with $S = 2, A = 1$ and the time-invariant transition kernel $P(0|0,0) = P(0|1,0) = SA(1-\gamma)^k<1$. Then, $\mu _{min} = \mu _{avg} = {SA(1-\gamma)^k}$ and the communication cost becomes $O(MH^kSA\log T)$ if we adopt the convention that $\frac{1}{1-\gamma}\approx H$. This could be much higher than our communication cost $O(MH^3SA\log T)$ if $k\gg 3$, despite that it considers a homogeneous setting while we consider step-varying transition kernels.
> > > > >
> > > > > In summary, even though [1] achieves logarithmic communication cost with the fixed communication design, it hinges on the infinite-horizon homogeneous setting and different learning objective. Our problems differ from [1] in both aspects. Those differences, especially our low regret learning objective, require a more sophisticated communication round control. Our event-triggered synchronization successfully achieves the low-regret learning objective with logarithmic communication cost, which cannot be achieved with a fixed communication round design as in [1].
> > > > >
> > > > > [1] Jiin Woo, Gauri Joshi, and Yuejie Chi. The blessing of heterogeneity in federated Q-learning: Linear speedup and beyond. In International Conference on Machine Learning, pp. 37157–37216, 2023.
> > > > >
> > > > > ------
> > > > >
> > > > > We thank the reviewer again for the feedback, and hope that our responses resolve your questions. We are more than happy to address any further questions that you may have. If our response resolves your concerns to a satisfactory level, we kindly request the reviewer to consider raising the rating of our work.

---

### Official Review · Reviewer_62fC · 2023-10-30

**Soundness:** 3 good
**Presentation:** 3 good
**Contribution:** 3 good
**Rating:** 6
**Confidence:** 3

**Summary:**

The paper studies federated reinforcement learning for the tabular episodic Markov Decision Process (MDP). In the model, there are $M$ agents that play in a given tabular episodic MDP and aim to minimize total regret. More formally, a tabular episodic MDP consists of $J$ episodes, where in each episode, the agents are given an adversarial picked initial state and everyone will keep picking actions till an absorbing state is reached. Each time an agent picks an action, it receives a reward that can help it update the strategy. An agent's regret for one episode is defined to be the difference between the rewards obtained by the optimal strategy and its strategy, and the goal is to minimize the sum of regrets for each agent in each episode.

The authors first propose a federated Q-Learning algorithm and show that with a communication cost of $O(M^2H^4S^2A\log(T/M))$, the algorithm obtains a regret of $\tilde{O}(\sqrt{H^4SAMT})$, where $H$ is the number of steps per episode, $T$ is the total number of steps, $S$ is the number of states in MDP, $A$ is the number of actions and $M$ is the number of agents. Further, using a higher upper confidence bound, the regret can be improved to $\tilde{O}(\sqrt{H^3SAMT})$ under the same communication cost.

**Strengths:**

- Federated reinforcement learning is a very interesting topic and the paper makes theoretical contributions in this direction. They prove that there exists a federated and model-free algorithm that achieves linear regret speedup (compared with the single-agent setting) with a relatively low communication cost.

- The paper is well-stated. In addition to describing the algorithm, some intuitions behind the algorithm design are provided in the paper.

**Weaknesses:**

- The main weakness is that the experimental evaluation is missing. The paper would be strengthened if an experimental section were added.

**Questions:**

(1) Are there any simple experimental figures for the proposed algorithms?

---

> ### Author Response · Authors · 2023-11-18
> **Responses to Reviewer 62fC**
>
> We thank the reviewer for the careful reading and thoughtful comments. We address the reviewer's questions in the following and have revised the paper accordingly. The changes are marked in blue in our revision. We hope the responses below and changes in the paper address the reviewer's concerns.
>
> **Q1:** The main weakness is that the experimental evaluation is missing. The paper would be strengthened if an experimental section were added.
>
>
> **A1:** Thank you for the suggestion. We have performed some experiments and updated our draft to include the experimental results of FedQ-Hoeffding and the comparison with the single-agent Q-learning algorithm UCB-Hoeffding in Appendix F. The source code has been uploaded as a supplementary file. The results clearly exhibit the linear speedup of the learning regret and logarithmic communication cost, which corroborate our theoretical results. We will perform more experiments to thoroughly validate the performances of FedQ-Hoeffding and FedQ-Bernstein, and include the results in the next version of this paper.
>
>
>
> -----
>
> We thank the reviewer again for the helpful comments and suggestions for our work. If our response resolves your concerns to a satisfactory level, we kindly ask the reviewer to consider raising the rating of our work. Certainly, we are more than happy to address any further questions that you may have.

---

> > ### Author Response · Authors · 2023-11-20
> > **Updated numerical experiments that included FedQ-Bernstein**
> >
> > We would like to let the reviewer know that we have successfully implemented FedQ-Beinstein and UCB-B. The new experimental results have been included in the latest version of this paper. We are striving to implement other algorithms from related works in order to have some comparison between those works and our algorithms. We will update the draft when such results become available.

---

> > > ### Author Response · Authors · 2023-11-22
> > >
> > > Dear Reviewer,
> > >
> > > As the author-reviewer discussion period will end soon, we will appreciate it if you could check our response to your review comments. This way, if you have further questions and comments, we can still reply before the author-reviewer discussion period ends. If our response resolves your concerns, we kindly ask you to consider raising the rating of our work. Thank you very much for your time and efforts!

---

> > > > ### Comment · Reviewer_62fC · 2023-11-22
> > > >
> > > > The reviewer thanks the author for their response. Given the new experimental results, the paper looks more complete now.

---

### Official Review · Reviewer_tdCn · 2023-10-30

**Soundness:** 3 good
**Presentation:** 3 good
**Contribution:** 2 fair
**Rating:** 6
**Confidence:** 4

**Summary:**

The algorithm proposes a federated-style Q-learning for tabular MDPs and shows the algorithm achieves linear-speedup in terms of regret, only requiring $\log(T)$ communication rounds. Two types of uncertainty bonuses, Hoeffding and Berstein, are considered.

**Strengths:**

- The paper's theoretical analysis is comprehensive, comprising of both Hoeffding-style and Berstein-style bonuses.
- The paper does not analyze the UCB-style algorithms typically considered in distributed/federated RL/bandit, and instead focuses on the harder to analyze Q-learning style algorithms.

**Weaknesses:**

1. The paper slightly over-claims its results. The regret speedup is nearly linear, but not exactly linear, due to the overhead terms that are linear in $M$, the number of machines, that appears in both Thm 4.1 and 5.1. Immediately after Thm 4.1, the paper also states that the algorithm enjoys a linear speedup in terms of $M$ in the general FL setting, despite the presence of the overhead terms.
2. While the typical martingale-style concentration analysis in the single agent RL setting cannot be directly applied, bounding each local term's "drift" from some "averaged parameter update path" is a commonly used technique in federated learning.
3. Compared with contemporary or prior works on federated RL, the paper focuses on a more "vanilla" setting, where updates are allowed to be adversarial or asynchronous, the setting considered in this paper is a fairly simplified, bare bones version of federated RL.

Minor Comments
1. The term $\beta$ in eq (3) and (4) should be (at least) indexed to reflect the fact that it changes over time. Preferably, it should also indicate that the term changes with $(x, a, h)$. Otherwise, as of now, it appears that a constant is added to the Q-function estimate at every single round and is misleading.
2. It might be useful to rename paragraph "New Weight Assignment Approach" on page 7 to "Equal Weight Assignment Approach", or something similar. Currently, the phrase "equal weight assignment" is claimed to be a major contribution, but from the manuscript it is not immediately obvious what this procedure means mathematically.
3. Please see the questions below on the tightness of the technical results.
4. Due to the overall similarity between RL and bandits, some literature survey on federated/distributed bandit would be a welcomed addition to Appendix A.

**Questions:**

1. It is a bit surprising that the number of communication rounds is linear in $M$. Can the authors intuitively explain why this must be the case, or if this term could be removed by more involved technical analysis?
2. Can the authors provide either some lower bound or some additional justification for the overhead terms?
3. For theorem 4.1, why can we ignore the overhead terms that are on the order of $O(HSAM\sqrt{H^3\iota} + H^4SAM)$ in the general FL setting?

---

> ### Author Response · Authors · 2023-11-18
> **Responses to Reviewer tdCn (part one)**
>
> We thank the reviewer for the careful reading and thoughtful comments. We address the reviewer's questions in the following and have revised the paper accordingly. The changes are marked in blue in our revision. We hope the responses below and the changes in the paper address the reviewer's concerns. Due to character limits of openreview, our reply is splitted into five parts.
>
> **Weakness 1**: The paper slightly over-claims its results. The regret speedup is nearly linear, but not exactly linear, due to the overhead terms that are linear in $M$, the number of machines, that appears in both Theorem 4.1 and 5.1. Immediately after Theorem 4.1, the paper also states that the algorithm enjoys a linear speedup in terms of $M$ in the general FL setting, despite the presence of the overhead terms.
>
> **Response**:  Thank you for your comment. In Theorem 4.1, we claim that with high probability, the cumulative regret over all clients can be upper bounded as
>     $$\mbox{Regret}(T)\leq O\left(\sqrt{H^4\iota MTSA} + M\mbox{poly}(H,S,A)\sqrt{\iota}\right),$$
> where $H, S, A$ are episode length, total number of states, and total number of actions, respectively, $T$ is the time horizon, and $\iota$ is a $\log$ factor.
>
> Correspondingly, the per-client regret under our algorithm scales in
> $$ O\left(\sqrt{\frac{H^4SAT\iota}{M}} + \mbox{poly}(H,S,A)\sqrt{\iota}\right),$$ in which **the overhead term no longer depends on $M$**.
>
> Thus, when we consider the regret speedup, we ignore the overhead and just focus on the impact of $M$ on the dominating term $\sqrt{\frac{H^4SAT\iota}{M}} $. This is why we claim "linear" regret speedup when $T$ is sufficiently large.
>
> We note that "linear speedup" is a commonly adopted terminology to describe the impact of multiple agents on various performance metrics in the federated setting, even though similar overhead terms exist and linear speedup is only observed in the dominant term. For example, Theorem 2 of [1] shows the sample complexity for FedAsynQ-EqAvg required to find an $\varepsilon$-optimal Q function scales in  $ \tilde{O}\left(\frac{1}{M\varepsilon^2} + M\right)$, and they claim it achieves a linear speedup in the sample complexity with respect to $M$, even though an overhead term $M$ exists.  Similar examples can be found in Theorem 4.1 of [2].
>
> [1] Jiin Woo, Gauri Joshi, and Yuejie Chi. The blessing of heterogeneity in federated Q-learning: Linear speedup and beyond. In International Conference on Machine Learning, pp. 37157–37216, 2023.
>
> [2] Kumar Kshitij Patel, Lingxiao Wang, Aadirupa Saha, and Nathan Srebro. Federated online and bandit convex optimization. In International Conference on Machine Learning, pp. 27439–27460, 2023.

---

> > ### Author Response · Authors · 2023-11-18
> > **Responses to Reviewer tdCn (part two)**
> >
> > **Weakness 2**: While the typical martingale-style concentration analysis in the single agent RL setting cannot be directly applied, bounding each local term's "drift" from some ``averaged parameter update path" is a commonly used technique in federated learning.
> >
> > **Response**: We agree with with reviewer that bounding each local term's "drift" from some "averaged parameter update path" is a commonly used technique in federated learning (FL). However, such bounding techniques in FL rely on certain assumptions (e.g., smoothness, convexity, bounded variation in gradients, uniform data sampling), which do not exist in federated RL. In particular, due to the online exploration nature of RL, the observations are collected as actions are taken sequentially at local agents. Due to the inherent randomness in the environment, even if the same policy is taken at all local agents, it may result in very different trajectories. Thus, it is hard to obtain an easy-to-track "averaged parameter update path" in federated RL, or a tight bound on the local terms' drifts from such averaged parameter update path.
> >
> > Our technique to bound the global estimation error in $Q$ for each round is significantly different from the "drift"-based analysis in FL. In particular, as we outlined in the Proof Sketch of Theorem 4.1, the critical step in our analysis is to bound $\left|\sum _{i=1}^{t^k} \tilde{\theta} _{t^k}^iX _i\right|$ where $X _i = V _{h+1}^\star(x _{h+1,i}) - \mathbb{E}\left[V _{h+1}^\star(x _{h+1})|(x _h,a _h) = (x,a)\right].$ Since $\tilde{\theta} _{t^k}^i=\sum _{i\in\mathcal{I}^{k'}} \theta _{t^k}^i / n _h^{k'}(x,a)$ if $N _h^k(x,a)\geq i _0$ (See Sec. 3.1, Case 2), which requires non-causal information $n _h^{k'}(x,a)$ to determine the weight for the $i$-th visit in the current round, $\tilde{\theta} _{t^k}^i,i=1,2...$ is a non-martingale sequence and the analysis of $\left|\sum _{i=1}^{t^k} \tilde{\theta} _{t^k}^iX _i\right|$ thus becomes challenging.
> >
> > To overcome this challenge, in the proof for Lemma C.3, we introduce a novel technique for **round-wise approximation and concentration**. In particular, we decompose the summation as
> > $$\sum _{i=1}^{t^k}\tilde{\theta} _{t^k}^iX _i=\sum _{i=1}^{t^k} \theta _{t^k}^iX _i +\sum _{i=1}^{t^k}(\tilde{\theta} _{t^k}^i - \theta _{t^k}^i)X _i,$$
> > and bound each term separately. While the first term preserves the martingale structure and can be bounded using Azuma-Hoeffding inequality, bounding the second term is still not straightforward.
> >
> > If we try to **bound each local term's drift**, we can only show that
> > $$\left|\sum _{i=1}^{t^k} (\tilde{\theta} _{t^k}^i - \theta _{t^k}^i)X _i\right|\leq H\sum _{i=1}^{t^k}\left|\tilde{\theta} _{t^k}^i - \theta _{t^k}^i\right| \leq O(1)$$
> > under our current algorithm design. This is because $\tilde{\theta} _{t^k}^i/\theta _{t^k}^i\in [\exp\left(-O(1/H)\right),\exp\left(O(1/H)\right)]$ as shown in (d) of Lemma B.3 and $\sum _{i = 1}^{t^k}\theta _{t^k}^i = 1$ due to the definition of $\theta _t^i$. If we stick to this technique, we have to further reduce the sample size generated in one round in order to have a tighter bound on $\tilde{\theta} _{t^k}^i/\theta _{t^k}^i$ and reduce the "drift" $\left|\tilde{\theta} _{t^k}^i - \theta _{t^k}^i\right|$. This would in turn increase the communication cost to $O(\sqrt{T})$, which is much worse compared to our current communication cost $O(\log T)$.
> >
> > To overcome this difficulty, instead of bounding the local drift in each time step, we choose to **group the "drift" terms based on the corresponding rounds** as
> >  $$\left|\sum _{i=1}^{t^k} (\tilde{\theta} _{t^k}^i - \theta _{t^k}^i)X_i\right|\leq \sum _{k'\in [k]}\left|\sum _{i\in\mathcal{I}^{k'}}(\tilde{\theta} _{t^k}^i - \theta _{t^k}^i)X _i\right|$$
> > and then bound the total deviation in each round $\left|\sum _{i\in \mathcal{I}^{k'}}(\tilde{\theta} _{t^k}^i - \theta _{t^k}^i)X _i\right|$ separately.
> >
> > In particular, within round $k'$, $X _i,i\in [N _h^{k'}(x,a)+1,N _h^{k'+1}(x,a)]$ is a martingale and $\tilde{\theta} _{t^k}^i-\theta _{t^k}^i=\alpha^c(t^{k'}+1,t^k)(\tilde{\theta} _{t^{k'}}^i - \theta _{t^{k'}}^i)$ thanks to the specially crafted weight assignment during global aggregation. Note that $\tilde{\theta} _{t^{k'}}^i-\theta _{t^{k'}}^i$ only depends on the value of $t^{k'}$, and does not depend on the realization after round $k'$, which enables us to obtain **a tighter bound compared with bounding each local drifts separately** without increasing the communication cost as follows:
> >  $$\left|\sum _{i=1}^{t^k} (\tilde{\theta} _{t^k}^i - \theta _{t^k}^i)X _i\right|\leq \sum _{k'\in [k]}\alpha^c(t^{k'}+1,t^k)\left|\sum _{i\in \mathcal{I}^{k'}}(\tilde{\theta} _{t^{k'}}^i - \theta _{t^{k'}}^i)X _i\right|\leq \tilde{O}(\sqrt{H^2/t^k}).$$
> > The detailed analysis can be found between Eqn (17) and Eqn (19) in Appendix C.2. To the best of our knowledge, this technique is new and may be useful to handle the randomness of visits in other federated RL algorithms.

---

> > > ### Author Response · Authors · 2023-11-18
> > > **Responses to Reviewer tdCn (part three)**
> > >
> > > **Weakness 3**: Compared with contemporary or prior works on federated RL, the paper focuses on a more "vanilla" setting, where updates are allowed to be adversarial or asynchronous, the setting considered in this paper is a fairly simplified, bare bones version of federated RL.
> > >
> > > **Response**: First, as pointed out in Appendix C.1, our algorithms naturally accommodate the **asynchronous setting** where each agent may generate different number of episodes during each round, which may be caused by different processing speed or latency in communications, and our theoretical analysis in the appendix applies to this general setting as well.
> > >
> > > We would like to emphasize that our paper is the first federated model-free RL algorithm that enjoys both (near) linear regret speedup and low communication cost. Even though it does not cover settings such as adversarial MDP, we believe that the results presented in this work fill important research gaps in the existing literature, and serve as a first step to the exploration of model-free federated RL in other more complicated settings.
> > >
> > > **Minor Comment 1**: The $\beta$ term in eq (3) and (4) should be (at least) indexed to reflect the fact that it changes over time. Preferably, it should also indicate that the term changes with $(x,a,h)$
> > > . Otherwise, as of now, it appears that a constant is added to the Q-function estimate at every single round and is misleading.
> > >
> > > **Response**: Thank you for your suggestion. We have revised our paper accordingly to make the notation clearer.
> > >
> > > **Minor Comment 2**: It might be useful to rename the paragraph "New Weight Assignment Approach" on page 7 to "Equal Weight Assignment Approach", or something similar. Currently, the phrase "equal weight assignment" is claimed to be a major contribution, but from the manuscript, it is not immediately obvious what this procedure means mathematically.
> > >
> > > **Response**: Thank you for your suggestion. We have revised our paper accordingly to highlight the novel weight assignment design and clarify the procedure that refers to the Q-value aggregation procedure at the central server.
> > >
> > > **Minor Comment 3**: Due to the overall similarity between RL and bandits, some literature survey on federated/distributed bandit would be a welcomed addition to Appendix A.
> > >
> > > **Response**: Thank you for the suggestion. We have included related work on federated/distributed bandits in the literature survey in Appendix A.

---

> ### Author Response · Authors · 2023-11-18
> **Responses to Reviewer tdCn (part four)**
>
> **Question 1**: It is a bit surprising that the number of communication rounds is linear in $M$. Can the authors intuitively explain why this must be the case, or if this term could be removed by more involved technical analysis?
>
> **Response**:
>  The current linear dependency of the number of communication rounds on $M$ is due to the event-triggered synchronization design in our algorithm. Specifically, if in round $k$, the visitation to any $(x,a,h)$ tuple at any agent $m$ equals $\max\left(1,\left\lfloor\frac{N _h^k(x,a)}{MH(H+1)}\right\rfloor\right)$, the current round ends and communications happen.
> Intuitively, as more agents are involved in this procedure, such an event happens more frequently, triggering more communication rounds.
>
> Mathematically,  under this synchronization protocol, for each round $k$, we can only claim that there exists one $(x,a,h,m)$ tuple such that the equality holds:
> $$n _h^{m,k}(x,a)= \max\left(1,\left\lfloor\frac{N _h^k(x,a)}{MH(H+1)}\right\rfloor\right).$$
> For the $(x,a,h,m)$ that meets this equality, after the early sequential updating stage, $n_h^{m,k}(x,a)$ is at least $\frac{N _h^k(x,a)}{MH(H+1)}$, which implies that $N _h^{k+1}(x,a)$ is at least $(1+\frac{1}{MH(H+1)}) N _h^{k}(x,a)$, i.e., increasing exponentially from a round to the next round.
>
> On the other hand, by the pigeonhole principle, if in total there are $K$ communication rounds over time $T$, there must exist one $(x,a,h)$ tuple such that the above equality holds for at least $\left\lceil\frac{K}{HSA}\right\rceil$ rounds. Combining with the exponential increase in the visiting number $n_h^{m,k}(x,a)$ during those rounds, we have that
>  $$MT\geq \Omega\left(\left[1+\Omega\left(\frac{1}{MH(H+1)}\right)\right]^{\left\lceil K/(HSA)\right\rceil}\right).$$
>  We note that $M$ appears in the denominator of the base on the RHS but is absent from the exponent, leading to the linear dependency on $M$ in the communication cost.
>
>  We point out that such a linear dependency on $M$ in the communication round upper bound commonly exists in federated RL algorithms, such as [1], [3].
>
>
> **The linear dependency in $M$ can be improved by removing the synchronization design.** Specifically, we can let each agent $m$ continue its current exploration round until the above equation is met for an $(x,a,h)$ tuple locally. Thus, out of $K$ rounds, the equality must hold for at least $KM$ times. By the pigeonhole principle, there exists one $(x,a,h)$ such that the above equality holds for at least $\left\lceil\frac{KM}{HSA}\right\rceil$ times. Intuitively, since now the length of each round is determined locally, the total number of communication rounds does not depend on $M$. A detailed analysis can be performed to show that the communication round scales in $O(\log T)$. We note that the implicit price for the improved dependency in $M$ is that the clients who finish their exploration in the current round earlier need to wait until all clients are done before proceeding to the next round.
>
>
> [1] Jiin Woo, Gauri Joshi, and Yuejie Chi. The blessing of heterogeneity in federated Q-learning: Linear speedup and beyond. In International Conference on Machine Learning, pp. 37157–37216, 2023.
>
>
> [3] Yiding Chen, Xuezhou Zhang, Kaiqing Zhang, Mengdi Wang, and Xiaojin Zhu. Byzantine-robust online and offline distributed reinforcement learning. In International Conference on Artificial Intelligence and Statistics, pp. 3230–3269. PMLR, 2023.

---

> ### Author Response · Authors · 2023-11-18
> **Responses to Reviewer tdCn (part five)**
>
> **Question 2**: Can the authors provide either some lower bound or some additional justification for the overhead terms?
>
> **Response**: Thank you for the question. We explain why those overhead terms linear in $M$ appear in the regret upper bound as follows.
>
> In our paper, $N _h^k(x,a)$ is the total visiting number of $(x,a,h)$ before starting round $k$ and $n _h^{m,k}(x,a)$ is the visiting number of $(x,a,h)$ of agent $m$ in round $k$. Our design for both the Hoeffding-type and the Bernstein-type algorithms guarantees that
> $$n _h^{m,k}(x,a)\leq \max \left(1,\left\lfloor\frac{N _h^k(x,a)}{MH(H+1)}\right\rfloor\right).$$
> During the early stage of the algorithms when $N _h^k(x,a)$ is small, we have $n _h^{m,k}(x,a)\leq 1$. Therefore, the total number of visits during round $k$ satisfies that
> $$n _h^k(x,a) = \sum _{m \in [M]} n _h^{m,k}(x,a)\leq M .$$
> Only when $N _h^k(x,a)$ becomes sufficiently large, the algorithms enter the second stage and the statistics start to kick in.
>
> In other words, the overhead terms is contributed by the $O(M)$ samples collected in the first stage, i.e., the burn-in cost. Such a burn-in cost is arguably inevitable in the federated setting. Intuitively, the learning agent needs to collect at least one sample for each $(x,a,h)$ tuple for a proper initialization. In the federated setting where all local clients collect their samples *in parallel*, it is hard to coordinate their exploration to reduce the dependency on $M$. We note that such linear overhead terms exist in [1] above as well.
>
>
> **Question 3**: For theorem 4.1, why can we ignore the overhead terms that are on the order of  in the general FL setting?
>
> **Response**: Please refer to our response to **Weakness 1**.
>
> -----
>
> We thank the reviewer again for the helpful comments and suggestions for our work. If our response resolves your concerns to a satisfactory level, we kindly ask the reviewer to consider raising the rating of our work. Certainly, we are more than happy to address any further questions that you may have.

---

> ### Comment · Reviewer_tdCn · 2023-11-19
>
> I am not convinced by the comment on the "linear speedup".
>
> - The overhead no longer depends on $M$ because an average is taken over all the machines. On a per machine basis, that means there is at least a part of the regret that does not benefit from the distributed setup whatsoever.
>
> - Moreover, in the discussion for Theorem 4.1 in [2], an example provided here, the authors of [2] explicitly mentions that we see linear speedup only for certain problem settings, but not overall. The referenced work clearly discusses when linear speedup is possible and when it is not by deriving a condition for $d$ under which the overhead term is dominated. Similar discussions are missing in the submission.
>
> - For [1], they claim linear speedup because in Theorem 1, the number of rounds required $T$ itself is shown to be $1/K$. In other words, they have achieved linear speedup because there are no additional terms that are not improved by having more machines, which is not the case for the regret bound in this paper. In later theorems, [1] also discusses the implications of $T_0$ and $\tilde{T}$ in the form of burn-in costs. For Theorems 2 and 3, [1] explicitly said that the burn-in costs need to be amortized over time in order to achieve some notion of linear speedup. Their burn-in costs are also slightly more appealing, as they are instance dependent and are not directly affected by $|S|$, $|A|$, and $H$.
>
> I think the paper would benefit greatly if it could incorporate these discussions, including
> 1. The reasons behind the burn-in cost and a brief discussion on when is the regret bound dominated by the term that is $O(\sqrt{M})$ rather than the terms that are $O(M)$.
> 2. Challenges caused by the RL setting that are not found in federated convex optimization.

---

> ### Author Response · Authors · 2023-11-19
> **Responses to Reviewer tdCn on new comments**
>
> We thank the reviewer for carefully checking our responses and providing prompt feedback and insightful suggestions. We address the reviewer's comments in the following and have revised the paper accordingly. We hope the responses below and the changes in the paper address the reviewer's concerns. We would be more than happy to address any further concerns that you may have.
>
> **Q1:** Condition for linear regret speedup in our work.
>
> **A1:** We thank the reviewer for the helpful suggestion about the conditions under which the linear regret speedup holds for the dominating term.  Rigorously speaking, the linear regret speedup can be guaranteed when the time steps $T$ is sufficiently large. Specifically, in Theorem 4.1, we show that with high probability,
> $$\mbox{Regret}(T)\leq \tilde{O}(1)\times O(\sqrt{H^4MTSA} + M\mbox{poly}(H,S,A)).$$
> Thus, when $T\geq \Omega(M\mbox{poly}(H,S,A)),$
> with high probability, FedQ-Hoeffding enjoys a linear speedup as follows
> $$\mbox{Reget}(T)\leq \tilde{O}(\sqrt{H^4MTSA}).$$
>
> Similar arguments can be provided for FedQ-Bernstein. When $T\geq \tilde{\Omega}(M\mbox{poly}(H,S,A))$ where $\tilde{\Omega}$ hides a log factor that takes the form $\log^2 (MT\mbox{poly}(H,S,A))$), the term $\tilde{O}(\sqrt{H^3SAMT})$ in Theorem 5.1 becomes the dominating term, thus a linear regret speedup can be achieved.
>
> We have added those conditions in the remarks after the theorems to make our statements more rigorous. We have also made according changes throughout the paper to clarify that the claimed linear speedup only holds when $T$ is sufficiently large.
>
> **Q2:** Discussions on the burn-in costs.
>
> **A2:** Thank you for your valuable suggestion! We have followed your advice to incorporate a brief discussion on the burn-in cost in the remark after Theorem 4.1.
>
> We would like to clarify that the instance-dependent burn-in cost in [1] also depends on $H,S,A$ in the asynchronous setting, and thus is of the same flavor as in our work.
>
> For asynchronous setting with equal weights (Theorem 2 in [1]), the burn in cost is
>     $$\tilde{O}\left(\frac{t _{min}^{max}\max (\frac{1}{1-\gamma}, M)}{\mu _{min}^2(1-\gamma)}\right).$$
> For asynchronous setting with unequal weights (Theorem 3 in [1]), it is
>     $$\tilde{O}\left(\frac{\max(t _{min}^{max},\frac{1}{1-\gamma}, M)}{\mu _{avg}(1-\gamma)}\right).$$
> Here $\mu _{min} = \min _{x,a,m} \mu^m(x,a)$ and $\mu _{avg} = \min _{x,a}\frac{1}{M}\sum _{m=1}^M\mu^m(x,a)$ in which $\mu^m(x,a)$ is the stationary visitation frequency for $(x,a)$ under the exploration policy at agent $m$, and constant $t _{min}^{max}\geq 1$ is the required time for reaching the stationary frequency.
>
> We first note that $\mu _{min} \leq \mu _{avg} \leq \frac{1}{SA}$. Besides, if we use the approximation $(1-\gamma)^{-1} \approx H$ to convert the results from infinite-horizon MDPs to those under episodic MDPs, both of the bounds now explicitly depend on $M,S,A,H$.
>
> **Q3:** Discussion on the challenges caused by the RL setting that are not found in federated convex optimization.
>
> **A3:** Thank you for this great suggestion. We have included a brief discussion on the unique challenges in federated RL that are not found in FL in the Proof Sketch of Theorem 4.1. Specifically, we add the following paragragh:
>
> *We would like to emphasize that the techniques required to bound those non-martingale differences are fundamentally different from the commonly used techniques in federated learning (FL), which usually construct an "averaged parameter update path'' and then bound each local term's "drift'' from it. This is because such bounding techniques in FL rely on certain assumptions that do not exist in federated RL. Due to the inherent randomness in the environment, even if the same policy is taken at all local agents, it may result in very different trajectories. Thus, it is hard to obtain an easy-to-track "averaged parameter update path'' in federated RL, or a tight bound on the local terms' drifts from such averaged parameter update path.  We overcome this challenge by relating $\{\tilde{\theta} _{t^k}^i\} _i$ with $\{{\theta} _{t^k}^i\} _i$. Instead of bounding the local drift $\tilde{\theta} _{t^k}^i-{\theta} _{t^k}^i$ in each time step, we choose to group the ``drift'' terms based on the corresponding rounds and then leverage the round-wise equal weight assignment adopted in our algorithm to obtain a tight bound.*
>
> ----
>
> We thank the reviewer again for your prompt reply and invaluable suggestions, which have helped us greatly to enhance the rigor and presentation of our work. If you have any further comments, questions or suggestions, we will be more than happy to address them. Meanwhile, if our responses have addressed your concerns satisfactorily, we kindly request that you re-evaluate our work based on the revision and raise the rating of our work.

---

> > ### Author Response · Authors · 2023-11-22
> >
> > Dear Reviewer,
> >
> > As the author-reviewer discussion period will end soon, we will appreciate it if you could check our response to your review comments. This way, if you have further questions and comments, we can still reply before the author-reviewer discussion period ends. If our response resolves your concerns, we kindly ask you to consider raising the rating of our work. Thank you very much for your time and efforts!

---

### Meta-Review · Area_Chair_zAfo · 2023-12-08

**Metareview:**

This paper considers federated reinforcement learning for tabular episodic Markov Decision Processes (MDP) where, under the coordination of a central server, multiple agents collaboratively explore the environment and learn an optimal policy without sharing their raw data. Regret bounds with reduced communication costs are studied.

Strength:
This is the first work showing that linear regret speedup and logarithmic communication cost can be achieved by model-free algorithms in federated reinforcement learning. The work has good contribution, and improves the state of the art in the literature.

Weaknesses:
The main concerns pointed is that infinite horizon discounted cost setup also has logarithmic communication cost. When comparing terms, there is an additional multiple of H. Given that approaches for logarithmic communication cost has been studied in different federated RL setups (not episodic model-free), there are some concerns on novelty.

**Justification For Why Not Higher Score:**

There are concerns on over-claiming the linear speedup, and the novelty in the logarithmic communication cost.

**Justification For Why Not Lower Score:**

This is the first work showing that linear regret speedup and logarithmic communication cost can be achieved by model-free algorithms in federated reinforcement learning.

---

### Decision · Program_Chairs · 2024-01-16

Accept (poster)